# $\mathcal{H}$-Consistency Bounds: Characterization and Extensions

**Anqi Mao**
Courant Institute
New York, NY 10012
aqmao@cims.nyu.edu

**Mehryar Mohri**
Google Research & CIMS
New York, NY 10011
mohri@google.com

**Yutao Zhong**
Courant Institute
New York, NY 10012
yutao@cims.nyu.edu

## Abstract

A series of recent publications by Awasthi, Mao, Mohri, and Zhong [2022b] have introduced the key notion of $\mathcal{H}$-*consistency bounds* for surrogate loss functions. These are upper bounds on the zero-one estimation error of any predictor in a hypothesis set, expressed in terms of its surrogate loss estimation error. They are both non-asymptotic and hypothesis set-specific and thus stronger and more informative than Bayes-consistency. However, determining if they hold and deriving these bounds have required a specific proof and analysis for each surrogate loss. Can we derive more general tools and characterizations? This paper provides both a general characterization and an extension of $\mathcal{H}$-consistency bounds for multi-class classification. We present new and tight $\mathcal{H}$-consistency bounds for both the family of constrained losses and that of comp-sum losses, which covers the familiar cross-entropy, or logistic loss applied to the outputs of a neural network. We further extend our analysis beyond the completeness assumptions adopted in previous studies and cover more realistic bounded hypothesis sets. Our characterizations are based on error transformations, which are explicitly defined for each formulation. We illustrate the application of our general results through several special examples. A by-product of our analysis is the observation that a recently derived multi-class $\mathcal{H}$-consistency bound for cross-entropy reduces to an excess bound and is not significant. Instead, we prove a much stronger and more significant guarantee.

## 1 Introduction

Bayes-consistency is an important property of surrogate loss functions. It requires that minimizing the surrogate excess error over the family of all measurable functions leads to the minimization of the target error loss in the limit [Steinwart, 2007]. This property applies to a broad family of convex margin-based losses in binary classification [Zhang, 2004a, Bartlett et al., 2006], as well as some extensions in multi-class classification [Tewari and Bartlett, 2007]. However, Bayes-consistency does not apply to the hypothesis sets commonly used for learning, such as the family of linear models or that of neural networks, which of course do not include all measurable functions. Furthermore, it is also only an asymptotic property and does not supply any convergence guarantee.

To address these limitations, a series of recent publications by Awasthi, Mao, Mohri, and Zhong [2022b] introduced the key notion of $\mathcal{H}$-*consistency bounds* for surrogate loss functions. These are upper bounds on the zero-one estimation error of any predictor in a hypothesis set, expressed in terms of its surrogate loss estimation error. They are both non-asymptotic and hypothesis set-specific and thus stronger and more informative than Bayes-consistency. However, determining the validity of these bounds and deriving them have required a specific proof and analysis for each surrogate loss. Can we derive more general tools and characterizations for $\mathcal{H}$-consistency bounds?

37th Conference on Neural Information Processing Systems (NeurIPS 2023).

This paper provides both a general characterization and an extension of $\mathcal{H}$-consistency bounds for multi-class classification. Previous approaches to deriving these bounds required the development of new proofs for each specific case. In contrast, we introduce the general concept of an *error transformation function* that serves as a very general tool for deriving such guarantees with tightness guarantees. We show that deriving an $\mathcal{H}$-consistency bound for comp-sum losses and constrained losses for both complete and bounded hypothesis sets can be reduced to the calculation of their corresponding error transformation function. Our general tools and tight bounds show several remarkable advantages: first, they improve existing bounds for complete hypothesis sets previously proven in [Awasthi et al., 2022b]; second, they encompass all previously comp-sum and constrained losses studied thus far as well as many new ones [Awasthi et al., 2022a, Mao et al., 2023h]; third, they extend beyond the completeness assumption adopted in previous work; fourth, they provide novel guarantees for bounded hypothesis sets; and, finally, they help prove a much stronger and more significant guarantee for logistic loss with linear hypothesis set than [Zheng et al., 2023].

**Previous work.** Here, we briefly discuss recent studies of $\mathcal{H}$-consistency bounds by Awasthi et al. [2022a,b], Mao et al. [2023h] and Zheng et al. [2023]. Awasthi et al. [2022a] introduced and studied $\mathcal{H}$-consistency bounds in binary classification. They provided a series of *tight* $\mathcal{H}$-consistency bounds for *bounded* hypothesis set of linear models and one-hidden-layer neural networks. The subsequent study [Awasthi et al., 2022b] further generalized the framework to multi-class classification and presented an extensive study of $\mathcal{H}$-consistency bounds for diverse multi-class surrogate losses, including negative results for *max losses* [Crammer and Singer, 2001] and positive results for *sum losses* [Weston and Watkins, 1998], and *constrained losses* [Lee et al., 2004]. However, the hypothesis sets examined in their analysis were assumed to be complete, which rules out the bounded hypothesis sets typically used in practice. Moreover, the final bounds derived from [Awasthi et al., 2022b] are based on ad hoc methods and may not be tight. [Mao et al., 2023h] complemented this previous work by studying a wide family of *comp-sum losses* in the multi-class classification, which generalizes the *sum-losses* and includes as special cases the logistic loss [Verhulst, 1838, 1845, Berkson, 1944, 1951], the *generalized cross-entropy loss* [Zhang and Sabuncu, 2018], and the *mean absolute error loss* [Ghosh et al., 2017]. Here too, the completeness assumption on the hypothesis sets was adopted and their $\mathcal{H}$-consistency bounds do not apply to common bounded hypothesis sets in practice. Recently, Zheng et al. [2023] proved $\mathcal{H}$-consistency bounds for multi-class logistic loss with bounded linear hypothesis sets. However, their bounds require a crucial distributional assumption, under which the minimizability gaps coincide with the approximation errors. Thus, their bounds can be recovered as excess error bounds, which are less significant.

Other related work on $\mathcal{H}$-consistency bounds includes $\mathcal{H}$-consistency bounds for pairwise ranking [Mao, Mohri, and Zhong, 2023d,e]; theoretically grounded surrogate losses and algorithms for learning with abstention supported by $\mathcal{H}$-consistency bounds, including the study of score-based abstention [Mao, Mohri, and Zhong, 2023f], predictor-rejector abstention [Mao, Mohri, and Zhong, 2023c] and learning to abstain with a fixed predictor with application in decontextualization [Mohri, Andor, Choi, Collins, Mao, and Zhong, 2023]; principled approaches for learning to defer with multiple experts that benefit from strong $\mathcal{H}$-consistency bounds, including the single-stage scenario [Mao, Mohri, and Zhong, 2023b] and a two-stage scenario [Mao, Mohri, Mohri, and Zhong, 2023a]; $\mathcal{H}$-consistency theory and algorithms for adversarial robustness [Awasthi et al., 2021a,b, 2023a, Mao et al., 2023h, Awasthi et al., 2023b]; and efficient algorithms and loss functions for structured prediction with stronger $\mathcal{H}$-consistency guarantees [Mao et al., 2023g].

**Structure of this paper.** We present new and tight $\mathcal{H}$-consistency bounds for both the family of comp-sum losses (Section 4.1) and that of constrained losses (Section 5.1), which cover the familiar cross-entropy, or logistic loss applied to the outputs of a neural network. We further extend our analysis beyond the completeness assumptions adopted in previous studies and cover more realistic bounded hypothesis sets (Section 4.2 and 5.2). Our characterizations are based on error transformations, which are explicitly defined for each formulation. We illustrate the application of our general results through several special examples. A by-product of our analysis is the observation that a recently derived multi-class $\mathcal{H}$-consistency bound for cross-entropy reduces to an excess bound independent of the hypothesis set. Instead, we prove a much stronger and more significant guarantee (Section 4.2).

We give a comprehensive discussion of related work in Appendix A. We start with some basic definitions and notation in Section 2.

## 2 Preliminaries

We denote by $\mathcal{X}$ the input space, by $\mathcal{Y}$ the output space, and by $\mathcal{D}$ a distribution over $\mathcal{X} \times \mathcal{Y}$. We consider the standard scenario of multi-class classification, where $\mathcal{Y} = \{1, \ldots, n\}$. Given a hypothesis set $\mathcal{H}$ of functions mapping $\mathcal{X} \times \mathcal{Y}$ to $\mathbb{R}$, the multi-class classification problem consists of finding a hypothesis $h \in \mathcal{H}$ with small generalization error $\mathcal{R}_{\ell_{0-1}}(h)$, defined by $\mathcal{R}_{\ell_{0-1}}(h) = \mathbb{E}_{(x,y)\sim\mathcal{D}}[\ell_{0-1}(h, x, y)]$, where $\ell_{0-1}(h, x, y) = \mathbb{1}_{\mathsf{h}(x)\neq y}$ is the multi-class zero-one loss with $\mathsf{h}(x) = \mathrm{argmax}_{y\in\mathcal{Y}} h(x, y)$ the prediction of $h$ for the input point $x$. We also denote by $\mathsf{H}(x)$ the set of all predictions associated to input $x$ generated by functions in $\mathcal{H}$, that is, $\mathsf{H}(x) = \{\mathsf{h}(x) : h \in \mathcal{H}\}$.

We will analyze the guarantees of surrogate multi-class losses in terms of the zero-one loss. We denote by $\ell$ a surrogate loss and by $\mathcal{R}_\ell(h)$ its generalization error, $\mathcal{R}_\ell(h) = \mathbb{E}_{(x,y)\sim\mathcal{D}}[\ell(h, x, y)]$. For a loss function $\ell$, we define the best-in-class generalization error within a hypothesis set $\mathcal{H}$ as $\mathcal{R}_\ell^*(\mathcal{H}) = \inf_{h\in\mathcal{H}} \mathcal{R}_\ell(h)$, and refer to $\mathcal{R}_\ell(h) - \mathcal{R}_\ell^*(\mathcal{H})$ as the *estimation error*. We will study the key notion of $\mathcal{H}$-*consistency bounds* [Awasthi et al., 2022a,b], which are upper bounds on the zero-one estimation error of any predictor in a hypothesis set, expressed in terms of its surrogate loss estimation error, for some real-valued function $f$ that is non-decreasing:

$$\forall h \in \mathcal{H}, \ \mathcal{R}_{\ell_{0-1}}(h) - \mathcal{R}_{\ell_{0-1}}^*(\mathcal{H}) \leq f(\mathcal{R}_\ell(h) - \mathcal{R}_\ell^*(\mathcal{H})).$$

These bounds imply that the zero-one estimation error is at most $f(\epsilon)$ whenever the surrogate loss estimation error is bounded by $\epsilon$. Thus, the learning guarantees provided by $\mathcal{H}$-consistency bounds are both non-asymptotic and hypothesis set-specific. The function $f$ appearing in these bounds is expressed in terms of a *minimizability gap*, which is a quantity measuring the difference of best-in-class error $\mathcal{R}_\ell^*(\mathcal{H})$ and the expected *best-in-class conditional error* $\mathbb{E}_x[\mathcal{C}_\ell^*(\mathcal{H}, x)]$: $\mathcal{M}_\ell(\mathcal{H}) = \mathcal{R}_\ell^*(\mathcal{H}) - \mathbb{E}_X[\mathcal{C}_\ell^*(\mathcal{H}, x)]$, where $\mathcal{C}_\ell(h, x) = \mathbb{E}_{y|x}[\ell(h, x, y)]$ and $\mathcal{C}_\ell^*(\mathcal{H}, x) = \inf_{h\in\mathcal{H}} \mathcal{C}_\ell(h, x)$ are the *conditional error* and *best-in-class conditional error* respectively. We further write $\Delta\mathcal{C}_{\ell,\mathcal{H}} = \mathcal{C}_\ell(h, x) - \mathcal{C}_\ell^*(\mathcal{H}, x)$ to denote the *conditional regret*. Note that that the minimizability gap is an inherent quantity depending on a hypothesis set $\mathcal{H}$ and the loss function $\ell$.

By Lemma 1, the minimizability gap for the zero-one loss, $\mathcal{M}_{\ell_{0-1}}(\mathcal{H})$, coincides with its approximation error $\mathcal{A}_{\ell_{0-1}}(\mathcal{H}) = \mathcal{R}_{\ell_{0-1}}^*(\mathcal{H}) - \mathcal{R}_{\ell_{0-1}}^*(\mathcal{H}_{\mathrm{all}})$ when the set of all possible predictions generated by $\mathcal{H}$ covers the label space $\mathcal{Y}$. This holds for typical hypothesis sets used in practice. However, for a surrogate loss $\ell$, the minimizability gap $\mathcal{M}_\ell(\mathcal{H})$ is always upper bounded by and in general finer than its approximation error $\mathcal{A}_\ell(\mathcal{H}) = \mathcal{R}_\ell^*(\mathcal{H}) - \mathcal{R}_\ell^*(\mathcal{H}_{\mathrm{all}})$ since $\mathcal{M}_\ell(\mathcal{H}) = \mathcal{A}_\ell(\mathcal{H}) - I_\ell(\mathcal{H})$, where $\mathcal{H}_{\mathrm{all}}$ is the family of all measurable functions and $I_\ell(\mathcal{H}) = \mathbb{E}_x[\mathcal{C}_\ell^*(\mathcal{H}, x) - \mathcal{C}_\ell^*(\mathcal{H}_{\mathrm{all}}, x)]$ (see Appendix B for a more detailed discussion). Thus, an $\mathcal{H}$-consistency bound, expressed as follows for some increasing function $\Gamma$:

$$\mathcal{R}_{\ell_{0-1}}(h) - \mathcal{R}_{\ell_{0-1}}^*(\mathcal{H}) + \mathcal{M}_{\ell_{0-1}}(\mathcal{H}) \leq \Gamma(\mathcal{R}_\ell(h) - \mathcal{R}_\ell^*(\mathcal{H}) + \mathcal{M}_\ell(\mathcal{H})), \tag{1}$$

is more favorable than an excess error bound expressed in terms of approximation errors $\mathcal{R}_{\ell_{0-1}}(h) - \mathcal{R}_{\ell_{0-1}}^*(\mathcal{H}) + \mathcal{A}_{\ell_{0-1}}(\mathcal{H}) \leq \Gamma(\mathcal{R}_\ell(h) - \mathcal{R}_\ell^*(\mathcal{H}) + \mathcal{A}_\ell(\mathcal{H}))$. Here, $\Gamma$ is typically linear or the square-root function modulo constants. When $\mathcal{H} = \mathcal{H}_{\mathrm{all}}$, the family of all measurable functions, an $\mathcal{H}$-consistency bound coincides with the excess error bound and implies Bayes-consistency by taking the limit. It is therefore a stronger guarantee than an excess error bound and Bayes-consistency.

The minimizability gap is always non-negative, since the infimum of the expectation is greater than or equal to the expectation of infimum. Furthermore, as shown in Appendix B, when $\mathcal{H}$ is the family of all measurable functions or when the Bayes-error coincides with the best-in-class error, that is, $\mathcal{R}_\ell^*(\mathcal{H}) = \mathcal{R}_\ell^*(\mathcal{H}_{\mathrm{all}})$, the minimizability gap vanishes. In such cases, (1) implies the $\mathcal{H}$-consistency of a surrogate loss $\ell$ with respect to the zero-one loss $\ell_{0-1}$:

$$\mathcal{R}_\ell(h_n) - \mathcal{R}_\ell^*(\mathcal{H}) \xrightarrow{n\to+\infty} 0 \implies \mathcal{R}_{\ell_{0-1}}(h_n) - \mathcal{R}_{\ell_{0-1}}^*(\mathcal{H}) \xrightarrow{n\to+\infty} 0.$$

In the next sections, we will provide both a general characterization and an extension of $\mathcal{H}$-consistency bounds for multi-class classification. Before proceeding, we first introduce a useful lemma from [Awasthi et al., 2022b] which characterizes the conditional regret of zero-one loss explicitly. We denote by $p(x) = (p(x, 1), \ldots, p(x, n))$ as the conditional distribution of $y$ given $x$.

**Lemma 1.** *For zero-one loss $\ell_{0-1}$, the best-in-class conditional error and the conditional regret for $\ell_{0-1}$ can be expressed as follows: for any $x \in \mathcal{X}$, we have*

$$\mathcal{C}_{\ell_{0-1}}^*(\mathcal{H}, x) = 1 - \max_{y\in\mathsf{H}(x)} p(x, y) \quad and \quad \Delta\mathcal{C}_{\ell_{0-1},\mathcal{H}}(h, x) = \max_{y\in\mathsf{H}(x)} p(x, y) - p(x, \mathsf{h}(x)).$$

# 3 Comparison with previous work

Here, we briefly discuss previous studies of $\mathcal{H}$-consistency bounds [Awasthi et al., 2022a,b, Zheng et al., 2023, Mao et al., 2023h] in standard binary or multi-class classification and compare their results with those we present.

Awasthi et al. [2022a] studied $\mathcal{H}$-consistency bounds in binary classification. They provided a series of *tight* $\mathcal{H}$-consistency bounds for the *bounded* hypothesis set of linear models $\mathcal{H}_{\text{lin}}^{\text{bi}}$ and one-hidden-layer neural networks $\mathcal{H}_{\text{NN}}^{\text{bi}}$, defined as follows:

$$\mathcal{H}_{\text{lin}}^{\text{bi}} = \left\{ x \mapsto w \cdot x + b \mid \|w\| \leq W, |b| \leq B \right\}$$

$$\mathcal{H}_{\text{NN}}^{\text{bi}} = \left\{ x \mapsto \sum_{j=1}^{n} u_j (w_j \cdot x + b)_+ \mid \|u\|_1 \leq \Lambda, \|w_j\| \leq W, |b| \leq B \right\},$$

where $B$, $W$, and $\Lambda$ are positive constants and where $(\cdot)_+ = \max(\cdot, 0)$. We will show that our bounds recover these binary classification $\mathcal{H}$-consistency bounds.

The scenario of multi-class classification is more challenging and more crucial in applications. Recent work by Awasthi et al. [2022b] showed that *max losses* [Crammer and Singer, 2001], defined as $\ell^{\max}(h, x, y) = \max_{y' \neq y} \Phi(h(x, y) - h(x, y'))$ for some convex and non-increasing function $\Phi$, cannot admit meaningful $\mathcal{H}$-consistency bounds, unless the distribution is deterministic. They also presented a series of $\mathcal{H}$-consistency bounds for *sum losses* [Weston and Watkins, 1998] and *constrained losses* [Lee et al., 2004] for *symmetric* and *complete* hypothesis sets, that is such that:

$$\mathcal{H} = \{h : \mathcal{X} \times \mathcal{Y} \to \mathbb{R} : h(\cdot, y) \in \mathcal{F}, \forall y \in \mathcal{Y}\} \qquad \text{(symmetry)}$$

$$\forall x \in \mathcal{X}, \{f(x) : f \in \mathcal{F}\} = \mathbb{R}, \qquad \text{(completeness)}$$

for some family $\mathcal{F}$ of functions mapping from $\mathcal{X}$ to $\mathbb{R}$. The completeness assumption rules out the bounded hypothesis sets typically used in practice such as $\mathcal{H}_{\text{lin}}$. Moreover, the final bounds derived from [Awasthi et al., 2022b] are based on ad hoc proofs and may not be tight. In contrast, we will study both the complete and bounded hypothesis sets, and provide a very general tool to derive $\mathcal{H}$-consistency bounds. Our bounds are tighter than those of Awasthi et al. [2022b] given for complete hypothesis sets and extend beyond the completeness assumption.

[Mao et al., 2023h] complemented the work of [Awasthi et al., 2022b] by studying a wide family of *comp-sum losses* in multi-class classification, which generalized the *sum-losses* and included as special cases the logistic loss [Verhulst, 1838, 1845, Berkson, 1944, 1951], the *generalized cross-entropy loss* [Zhang and Sabuncu, 2018], and the *mean absolute error loss* [Ghosh et al., 2017]. Here too, the completeness assumption was adopted, thus their $\mathcal{H}$-consistency bounds do not apply to common bounded hypothesis sets used in practice. We illustrate the application of our general results through a broader set of surrogate losses than [Mao et al., 2023h] and significantly generalize the bounds of [Mao et al., 2023h] to bounded hypothesis sets.

Recently, Zheng et al. [2023] proved $\mathcal{H}$-consistency bounds for logistic loss with linear hypothesis sets in the multi-class classification: $\mathcal{H}_{\text{lin}} = \{x \mapsto w_y \cdot x + b_y \mid \|w_y\| \leq W, |b_y| \leq B, y \in \mathcal{Y}\}$. However, their bounds require a crucial distributional assumption under which, the minimizability gaps $\mathcal{M}_{\ell_{0-1}}(\mathcal{H}_{\text{lin}})$ and $\mathcal{M}_{\ell_{\log}}(\mathcal{H}_{\text{lin}})$ coincide with the approximation errors $\mathcal{R}_{\ell_{0-1}}(\mathcal{H}_{\text{lin}}) - \mathcal{R}_{\ell_{0-1}}^*(\mathcal{H}_{\text{all}})$ and $\mathcal{R}_{\ell_{\log}}(\mathcal{H}_{\text{lin}}) - \mathcal{R}_{\ell_{\log}}^*(\mathcal{H}_{\text{all}})$ respectively (see the note before [Zheng et al., 2023, Appendix F]). Thus, their bounds can be recovered as excess error bounds $\mathcal{R}_{\ell_{0-1}}(h) - \mathcal{R}_{\ell_{0-1}}^*(\mathcal{H}_{\text{all}}) \leq \sqrt{2}\left(\mathcal{R}_{\ell_{\log}}(h) - \mathcal{R}_{\ell_{\log}}^*(\mathcal{H}_{\text{all}})\right)^{\frac{1}{2}}$, which are less significant. In contrast, our $\mathcal{H}_{\text{lin}}$-consistency bound are much finer and take into account the role of the parameter $B$ and that of the number of labels $n$. Thus, we provide stronger and more significant guarantees for logistic loss with linear hypothesis set than [Zheng et al., 2023].

In summary, our general tools offer the remarkable advantages of deriving tight bounds, which improve upon the existing bounds of Awasthi et al. [2022b] given for complete hypothesis sets, cover the comp-sum and constrained losses considered in [Awasthi et al., 2022a, Mao et al., 2023h] as well as new ones, extend beyond the completeness assumption with novel guarantees valid for bounded hypothesis sets, and are much stronger and more significant guarantees for logistic loss with linear hypothesis sets than those of Zheng et al. [2023].

# 4 Comp-sum losses

In this section, we present a general characterization of $\mathcal{H}$-consistency bounds for *comp-sum losses*, a family of loss functions including the *logistic loss* [Verhulst, 1838, 1845, Berkson, 1944, 1951], the *sum exponential loss* [Weston and Watkins, 1998, Awasthi et al., 2022b], the *generalized cross entropy loss* [Zhang and Sabuncu, 2018], the *mean absolute error loss* [Ghosh et al., 2017], and many other loss functions used in applications.

This is a family of loss functions defined via the composition of a non-negative and non-decreasing function $\Psi$ with the sum exponential losses (see [Mao et al., 2023h]):

$$\forall h \in \mathcal{H}, \forall (x,y) \times \mathcal{X} \times \mathcal{Y}, \quad \ell^{\mathrm{comp}}(h,x,y) = \Psi\left(\sum_{y' \neq \mathcal{Y}} e^{h(x,y') - h(x,y)}\right). \tag{2}$$

This expression can be equivalently written as $\ell^{\mathrm{comp}}(h,x,y) = \Phi\left(\frac{e^{h(x,y)}}{\sum_{y' \in \mathcal{Y}} e^{h(x,y')}}\right)$, where $\Phi: u \mapsto \Psi\left(\frac{1-u}{u}\right)$ is a non-increasing auxiliary function from $[0,1]$ to $\mathbb{R}_+ \cup \{+\infty\}$. As an example, the logistic loss corresponds to the choice $\Phi: u \mapsto -\log(u)$ and the sum exponential loss to $\Phi: u \mapsto \frac{1-u}{u}$.

## 4.1 $\mathcal{H}$-consistency bounds

In previous work, deriving $\mathcal{H}$-consistency bounds has required giving new proofs for each instance. The following result provides a very general tool for deriving such bounds with tightness guarantees. We introduce an *error transformation function* and show that deriving an $\mathcal{H}$-consistency bound for comp-sum losses can be reduced to the calculation of this function.

**Theorem 2** ($\mathcal{H}$**-consistency bound for comp-sum losses**)**.** *Assume that $\mathcal{H}$ is symmetric and complete and that $\mathcal{T}^{\mathrm{comp}}$ is convex. Then, the following inequality holds for any hypothesis $h \in \mathcal{H}$ and any distribution*

$$\mathcal{T}^{\mathrm{comp}}\big(\mathcal{R}_{\ell_{0-1}}(h) - \mathcal{R}^*_{\ell_{0-1}}(\mathcal{H}) + \mathcal{M}_{\ell_{0-1}}(\mathcal{H})\big) \leq \mathcal{R}_{\ell^{\mathrm{comp}}}(h) - \mathcal{R}^*_{\ell^{\mathrm{comp}}}(\mathcal{H}) + \mathcal{M}_{\ell^{\mathrm{comp}}}(\mathcal{H}), \tag{3}$$

*with $\mathcal{T}^{\mathrm{comp}}$ an $\mathcal{H}$-estimation error transformation for comp-sum losses defined for all $t \in [0,1]$ by*

$\mathcal{T}^{\mathrm{comp}}(t) =$

$$\begin{cases} \inf_{\tau \in [0, \frac{1}{2}]} \sup_{\mu \in [-\tau, 1-\tau]} \left\{ \frac{1+t}{2}\big[\Phi(\tau) - \Phi(1-\tau-\mu)\big] + \frac{1-t}{2}\big[\Phi(1-\tau) - \Phi(\tau+\mu)\big] \right\} & n = 2 \\ \inf_{P \in [\frac{1}{n-1} \vee t, 1]} \inf_{\substack{\tau_1 \geq \max(\tau_2, 1/n) \\ \tau_1 + \tau_2 \leq 1, \tau_2 \geq 0}} \sup_{\mu \in [-\tau_2, \tau_1]} \left\{ \frac{P+t}{2}\big[\Phi(\tau_2) - \Phi(\tau_1 - \mu)\big] + \frac{P-t}{2}\big[\Phi(\tau_1) - \Phi(\tau_2 + \mu)\big] \right\} & n > 2. \end{cases}$$

*Furthermore, for any $t \in [0,1]$, there exist a distribution $\mathcal{D}$ and a hypothesis $h \in \mathcal{H}$ such that $\mathcal{R}_{\ell_{0-1}}(h) - \mathcal{R}^*_{\ell_{0-1}}(\mathcal{H}) + \mathcal{M}_{\ell_{0-1}}(\mathcal{H}) = t$ and $\mathcal{R}_{\ell^{\mathrm{comp}}}(h) - \mathcal{R}^*_{\ell^{\mathrm{comp}}}(\mathcal{H}) + \mathcal{M}_{\ell^{\mathrm{comp}}}(\mathcal{H}) = \mathcal{T}^{\mathrm{comp}}(t)$.*

Thus, Theorem 2 shows that, when $\mathcal{T}^{\mathrm{comp}}$ is convex, to make these guarantees explicit, all that is needed is to calculate $\mathcal{T}^{\mathrm{comp}}$. Moreover, the last statement shows the *tightness* of the guarantees derived using this function. The constraints in $\mathcal{T}^{\mathrm{comp}}$ are due to the forms that the conditional probability vector and scoring functions take. These forms become more flexible for $n > 2$, leading to intricate constraints. Note that our $\mathcal{H}$-consistency bounds are distribution-independent and we cannot claim tightness across all distributions.

The general expression of $\mathcal{T}^{\mathrm{comp}}$ in Theorem 2 is complex, but it can be considerably simplified under some broad assumptions, as shown by the following result.

**Theorem 3** (**characterization of** $\mathcal{T}^{\mathrm{comp}}$)**.** *Assume that $\Phi$ is convex, differentiable at $\frac{1}{2}$ and $\Phi'\left(\frac{1}{2}\right) < 0$. Then, $\mathcal{T}^{\mathrm{comp}}$ can be expressed as follows:*

$$\mathcal{T}^{\mathrm{comp}}(t) = \begin{cases} \Phi\left(\frac{1}{2}\right) - \inf_{\mu \in [-\frac{1}{2}, \frac{1}{2}]} \left\{ \frac{1-t}{2}\Phi\left(\frac{1}{2} + \mu\right) + \frac{1+t}{2}\Phi\left(\frac{1}{2} - \mu\right) \right\} & n = 2 \\ \inf_{\tau \in [\frac{1}{n}, \frac{1}{2}]} \left\{ \Phi(\tau) - \inf_{\mu \in [-\tau, \tau]} \left\{ \frac{1+t}{2}\Phi(\tau - \mu) + \frac{1-t}{2}\Phi(\tau + \mu) \right\} \right\} & n > 2. \end{cases}$$

The proof of this result as well as that of other theorems in this section are given in Appendix C.

**Examples.** We now illustrate the application of our theory through several examples. To do so, we compute the $\mathcal{H}$-estimation error transformation $\mathcal{T}^{\mathrm{comp}}$ for comp-sum losses and present the results in

Table 1: $\mathcal{H}$-estimation error transformation for common comp-sum losses.

| Auxiliary function $\Phi$ | $-\log(t)$ | $\frac{1}{t}-1$ | $\frac{1}{q}(1-t^q), q \in (0,1)$ | $1-t$ | $(1-t)^2$ |
|---|---|---|---|---|---|
| Transformation $\mathcal{T}^{\mathrm{comp}}$ | $\frac{1+t}{2}\log(1+t) + \frac{1-t}{2}\log(1-t)$ | $1-\sqrt{1-t^2}$ | $\frac{1}{qn^q}\left(\frac{(1+t)^{\frac{1}{1-q}} + (1-t)^{\frac{1}{1-q}}}{2}\right)^{1-q} - \frac{1}{qn^q}$ | $\frac{t}{n}$ | $\frac{t^2}{4}$ |

Table 1. Remarkably, by applying Theorem 2, we are able to obtain the same $\mathcal{H}$-consistency bounds for comp-sum losses with $\Phi(t) = -\log(t)$, $\frac{1}{t} - 1$, $\frac{1}{q}(1-t^q)$, $q \in (0,1)$ and $1-t$ as those derived using ad hoc methods in [Mao et al., 2023h], and a novel tight $\mathcal{H}$-consistency bound for the new comp-sum loss $\ell_{\mathrm{sq}} = \left[1 - \frac{e^{h(x,y)}}{\sum_{y' \in \mathcal{Y}} e^{h(x,y')}}\right]^2$ with $\Phi(t) = (1-t)^2$ in Theorem 4.

The calculation of $\mathcal{T}^{\mathrm{comp}}$ for all entries of Table 1 is detailed in Appendix C.3. To illustrate the effectiveness of our general tools, here, we show how the error transformation function can be straightforwardly calculated in the case of the new surrogate loss $\ell_{\mathrm{sq}}$.

**Theorem 4** ($\mathcal{H}$-**consistency bound for a new comp-sum loss**). *Assume that $\mathcal{H}$ is symmetric and complete. Then, for all $h \in \mathcal{H}$ and any distribution, the following tight bound holds.*

$$\mathcal{R}_{\ell_{0-1}}(h) - \mathcal{R}^*_{\ell_{0-1}}(\mathcal{H}) \le 2\left(\mathcal{R}_{\ell_{\mathrm{sq}}}(h) - \mathcal{R}^*_{\ell_{\mathrm{sq}}}(\mathcal{H}) + \mathcal{M}_{\ell_{\mathrm{sq}}}(\mathcal{H})\right)^{\frac{1}{2}} - \mathcal{M}_{\ell_{0-1}}(\mathcal{H}).$$

*Proof.* For $n = 2$, plugging in $\Phi(t) = (1-t)^2$ in Theorem 3, gives

$$\mathcal{T}^{\mathrm{comp}} = \frac{1}{4} - \inf_{\mu \in \left[-\frac{1}{2}, \frac{1}{2}\right]} \left\{ \frac{1-t}{2}\left(\frac{1}{2} - \mu\right)^2 + \frac{1+t}{2}\left(\frac{1}{2} + \mu\right)^2 \right\} = \frac{1}{4} - \frac{1-t^2}{4} = \frac{t^2}{4}.$$

Similarly, for $n > 2$, plugging in $\Phi(t) = (1-t)^2$ in Theorem 3 yields

$$
\begin{aligned}
\mathcal{T}^{\mathrm{comp}} &= \inf_{\tau \in \left[\frac{1}{n}, \frac{1}{2}\right]} \left\{ (1-\tau)^2 - \inf_{\mu \in [-\tau, \tau]} \left\{ \frac{1+t}{2}(1-\tau+\mu)^2 + \frac{1-t}{2}(1-\tau-\mu)^2 \right\} \right\} \\
&= \inf_{\tau \in \left[\frac{1}{n}, \frac{1}{2}\right]} \left\{ (1-\tau)^2 - (1-\tau)^2(1-t^2) \right\} \qquad \text{(minimum achieved at } \mu = t(\tau - 1)) \\
&= \frac{t^2}{4}. \qquad\qquad\qquad\qquad\qquad\qquad\qquad\qquad \text{(minimum achieved at } \tau = \tfrac{1}{2})
\end{aligned}
$$

By Theorem 2, the bound obtained is tight, which completes the proof. $\qquad\square$

## 4.2 Extension to non-complete/bounded hypothesis sets: comp-sum losses

As pointed out earlier, the hypothesis sets typically used in practice are bounded. Let $\mathcal{F}$ be a family of real-valued functions $f$ with $|f(x)| \le \Lambda(x)$ for all $x \in \mathcal{X}$ and such that all values in $[-\Lambda(x), +\Lambda(x)]$ can be reached, where $\Lambda(x) > 0$ is a fixed function on $\mathcal{X}$. We will study hypothesis sets $\overline{\mathcal{H}}$ in which each scoring function is bounded:

$$\overline{\mathcal{H}} = \{h : \mathcal{X} \times \mathcal{Y} \to \mathbb{R} \mid h(\cdot, y) \in \mathcal{F}, \forall y \in \mathcal{Y}\}. \tag{4}$$

This holds for most hypothesis sets used in practice. The symmetric and complete hypothesis sets studied in previous work correspond to the special case of $\overline{\mathcal{H}}$ where $\Lambda(x) = +\infty$ for all $x \in \mathcal{X}$. The hypothesis set of linear models $\mathcal{H}_{\mathrm{lin}}$, defined by

$$\mathcal{H}_{\mathrm{lin}} = \left\{ (x, y) \mapsto w_y \cdot x + b_y \mid \|w_y\| \le W, |b_y| \le B, y \in \mathcal{Y} \right\},$$

is also a special instance of $\overline{\mathcal{H}}$ where $\Lambda(x) = W\|x\| + B$. Let us emphasize that previous studies did not establish any $\mathcal{H}$-consistency bound for these general hypothesis sets, $\overline{\mathcal{H}}$.

**Theorem 5** ($\overline{\mathcal{H}}$-**consistency bound for comp-sum losses**). *Assume that $\overline{\mathcal{T}}^{\mathrm{comp}}$ is convex. Then, the following inequality holds for any hypothesis $h \in \overline{\mathcal{H}}$ and any distribution:*

$$\overline{\mathcal{T}}^{\mathrm{comp}}\left(\mathcal{R}_{\ell_{0-1}}(h) - \mathcal{R}^*_{\ell_{0-1}}(\overline{\mathcal{H}}) + \mathcal{M}_{\ell_{0-1}}(\overline{\mathcal{H}})\right) \le \mathcal{R}_{\ell^{\mathrm{comp}}}(h) - \mathcal{R}^*_{\ell^{\mathrm{comp}}}(\overline{\mathcal{H}}) + \mathcal{M}_{\ell^{\mathrm{comp}}}(\overline{\mathcal{H}})$$

with $\overline{\mathcal{T}}^{\mathrm{comp}}$ *the* $\overline{\mathcal{H}}$*-estimation error transformation for comp-sum losses defined for all* $t \in [0,1]$ *by* $\overline{\mathcal{T}}^{\mathrm{comp}}(t) =$

$$
\begin{cases}
\displaystyle \inf_{\tau \in [0,\frac{1}{2}]} \sup_{\mu \in [s_{\min}-\tau, 1-\tau-s_{\min}]} \left\{ \frac{1+t}{2}[\Phi(\tau) - \Phi(1-\tau-\mu)] + \frac{1-t}{2}[\Phi(1-\tau) - \Phi(\tau+\mu)] \right\} & n = 2 \\
\displaystyle \inf_{P \in [\frac{1}{n-1} \vee t, 1]} \inf_{\substack{S_{\min} \le \tau_2 \le \tau_1 \le S_{\max} \\ \tau_1 + \tau_2 \le 1}} \sup_{\mu \in C} \left\{ \frac{P+t}{2}[\Phi(\tau_2) - \Phi(\tau_1 - \mu)] + \frac{P-t}{2}[\Phi(\tau_1) - \Phi(\tau_2 + \mu)] \right\} & n > 2,
\end{cases}
$$

*where* $C = [\max\{s_{\min} - \tau_2, \tau_1 - s_{\max}\}, \min\{s_{\max} - \tau_2, \tau_1 - s_{\min}\}]$, $s_{\max} = \frac{1}{1+(n-1)e^{-2\inf_x \Lambda(x)}}$ *and* $s_{\min} = \frac{1}{1+(n-1)e^{2\inf_x \Lambda(x)}}$. *Furthermore, for any* $t \in [0,1]$*, there exist a distribution* $\mathcal{D}$ *and* $h \in \mathcal{H}$ *such that* $\mathcal{R}_{\ell_{0-1}}(h) - \mathcal{R}^*_{\ell_{0-1}}(\mathcal{H}) + \mathcal{M}_{\ell_{0-1}}(\mathcal{H}) = t$ *and* $\mathcal{R}_{\ell^{\mathrm{comp}}}(h) - \mathcal{R}^*_{\ell^{\mathrm{comp}}}(\mathcal{H}) + \mathcal{M}_{\ell^{\mathrm{comp}}}(\mathcal{H}) = \mathcal{T}^{\mathrm{comp}}(t)$.

This theorem significantly broadens the applicability of our framework as it encompasses bounded hypothesis sets. The last statement of the theorem further shows the tightness of the $\mathcal{H}$-consistency bounds derived using this error transformation function. We now illustrate the application of our theory through several examples.

**A. Example: logistic loss.** We first consider the multinomial logistic loss, that is $\ell^{\mathrm{comp}}$ with $\Phi(u) = -\log(u)$, for which we give the following guarantee.

**Theorem 6** ($\overline{\mathcal{H}}$**-consistency bounds for logistic loss**). *For any* $h \in \overline{\mathcal{H}}$ *and any distribution, we have*

$$
\mathcal{R}_{\ell_{0-1}}(h) - \mathcal{R}^*_{\ell_{0-1}}(\overline{\mathcal{H}}) + \mathcal{M}_{\ell_{0-1}}(\overline{\mathcal{H}}) \le \Psi^{-1}\Big( \mathcal{R}_{\ell_{\log}}(h) - \mathcal{R}^*_{\ell_{\log}}(\overline{\mathcal{H}}) + \mathcal{M}_{\ell_{\log}}(\overline{\mathcal{H}}) \Big),
$$

*where* $\ell_{\log} = -\log\Big( \frac{e^{h(x,y)}}{\sum_{y' \in \mathcal{Y}} e^{h(x,y')}} \Big)$ *and* $\Psi(t) = \begin{cases} \frac{1+t}{2}\log(1+t) + \frac{1-t}{2}\log(1-t) & t \le \frac{s_{\max}-s_{\min}}{s_{\min}+s_{\max}} \\ \frac{t}{2}\log\Big(\frac{s_{\max}}{s_{\min}}\Big) + \log\Big(\frac{2\sqrt{s_{\max}s_{\min}}}{s_{\max}+s_{\min}}\Big) & \text{otherwise.} \end{cases}$

The proof of Theorem 6 is given in Appendix E.2. With the help of some simple calculations, we can derive a simpler upper bound:

$$
\Psi^{-1}(t) \le \Gamma(t) = \begin{cases} \sqrt{2t} & t \le \frac{(s_{\max}-s_{\min})^2}{2(s_{\min}+s_{\max})^2} \\ \frac{2(s_{\min}+s_{\max})}{s_{\max}-s_{\min}}t & \text{otherwise.} \end{cases}
$$

When the relative difference between $s_{\min}$ and $s_{\max}$ is small, the coefficient of the linear term in $\Gamma$ explodes. On the other hand, making that difference large essentially turns $\Gamma$ into a square-root function for all values. In general, $\Lambda$ is not infinite since a regularization is used, which controls both the complexity of the hypothesis set and the magnitude of the scores.

**Comparison with [Mao et al., 2023h].** For the symmetric and complete hypothesis sets $\mathcal{H}$ considered in [Mao et al., 2023h], $\Lambda(x) = +\infty$, $s_{\max} = 1$, $s_{\min} = 0$, $\Psi(t) = \frac{1+t}{2}\log(1+t) + \frac{1-t}{2}\log(1-t)$ and $\Gamma(t) = \sqrt{2t}$. By Theorem 6, this yields an $\mathcal{H}$-consistency bound for the logistic loss.

**Corollary 7** ($\mathcal{H}$**-consistency bounds for logistic loss**). *Assume that* $\mathcal{H}$ *is symmetric and complete. Then, for any* $h \in \mathcal{H}$ *and any distribution, we have*

$$
\mathcal{R}_{\ell_{0-1}}(h) - \mathcal{R}^*_{\ell_{0-1}}(\mathcal{H}) \le \Psi^{-1}\Big( \mathcal{R}_{\ell_{\log}}(h) - \mathcal{R}^*_{\ell_{\log}}(\mathcal{H}) + \mathcal{M}_{\ell_{\log}}(\mathcal{H}) \Big) - \mathcal{M}_{\ell_{0-1}}(\mathcal{H})
$$

*where* $\Psi(t) = \frac{1+t}{2}\log(1+t) + \frac{1-t}{2}\log(1-t)$ *and* $\Psi^{-1}(t) \le \sqrt{2t}$.

Corollary 7 recovers the $\mathcal{H}$-consistency bounds of Mao et al. [2023h].

**Comparison with [Awasthi et al., 2022a] and [Zheng et al., 2023].** For the linear models $\mathcal{H}_{\mathrm{lin}} = \big\{ (x,y) \mapsto w_y \cdot x + b_y \mid \|w_y\| \le W, |b_y| \le B \big\}$, we have $\Lambda(x) = W\|x\| + B$. By Theorem 6, we obtain $\mathcal{H}_{\mathrm{lin}}$-consistency bounds for logistic loss.

**Corollary 8** ($\mathcal{H}_{\mathrm{lin}}$**-consistency bounds for logistic loss**). *For any* $h \in \mathcal{H}_{\mathrm{lin}}$ *and any distribution,*

$$
\mathcal{R}_{\ell_{0-1}}(h) - \mathcal{R}^*_{\ell_{0-1}}(\mathcal{H}_{\mathrm{lin}}) \le \Psi^{-1}\Big( \mathcal{R}_{\ell_{\log}}(h) - \mathcal{R}^*_{\ell_{\log}}(\mathcal{H}_{\mathrm{lin}}) + \mathcal{M}_{\ell_{\log}}(\mathcal{H}_{\mathrm{lin}}) \Big) - \mathcal{M}_{\ell_{0-1}}(\mathcal{H}_{\mathrm{lin}})
$$

*where* $\Psi(t) = \begin{cases} \frac{1+t}{2}\log(1+t) + \frac{1-t}{2}\log(1-t) & t \le \frac{(n-1)(e^{2B}-e^{-2B})}{2+(n-1)(e^{2B}+e^{-2B})} \\ \frac{t}{2}\log\Big(\frac{1+(n-1)e^{2B}}{1+(n-1)e^{-2B}}\Big) + \log\Big(\frac{2\sqrt{(1+(n-1)e^{2B})(1+(n-1)e^{-2B})}}{2+(n-1)(e^{2B}+e^{-2B})}\Big) & \text{otherwise.} \end{cases}$

For $n = 2$, we have $\Psi(t) = \begin{cases} \frac{t+1}{2}\log(t+1) + \frac{1-t}{2}\log(1-t) & t \leq \frac{e^{2B}-1}{e^{2B}+1} \\ \frac{t}{2}\log\left(\frac{1+e^{2B}}{1+e^{-2B}}\right) + \log\left(\frac{2\sqrt{(1+e^{2B})(1+e^{-2B})}}{2+e^{2B}+e^{-2B}}\right) & \text{otherwise,} \end{cases}$ which coincides with the $\mathcal{H}_{\text{lin}}$-estimation error transformation in [Awasthi et al., 2022a]. Thus, Corollary 8 includes as a special case the $\mathcal{H}_{\text{lin}}$-consistency bounds given by Awasthi et al. [2022a] for binary classification.

Our bounds of Corollary 8 improves upon the multi-class $\mathcal{H}_{\text{lin}}$-consistency bounds of recent work [Zheng et al., 2023, Theorem 3.3] in the following ways: i) their bound holds only for restricted distributions while our bound holds for any distribution; ii) their bound holds only for restricted values of the estimation error $\mathcal{R}_{\ell_{\log}}(h) - \mathcal{R}^*_{\ell_{\log}}(\mathcal{H}_{\text{lin}})$ while ours holds for any value in $\mathbb{R}$ and more precisely admits a piecewise functional form; iii) under their distributional assumption, the minimizability gaps $\mathcal{M}_{\ell_{0-1}}(\mathcal{H}_{\text{lin}})$ and $\mathcal{M}_{\ell_{\log}}(\mathcal{H}_{\text{lin}})$ coincide with the approximation errors $\mathcal{R}_{\ell_{0-1}}(\mathcal{H}_{\text{lin}}) - \mathcal{R}^*_{\ell_{0-1}}(\mathcal{H}_{\text{all}})$ and $\mathcal{R}_{\ell_{\log}}(\mathcal{H}_{\text{lin}}) - \mathcal{R}^*_{\ell_{\log}}(\mathcal{H}_{\text{all}})$ respectively (see the note before [Zheng et al., 2023, Appendix F]). Thus, their bounds can be recovered as an excess error bound $\mathcal{R}_{\ell_{0-1}}(h) - \mathcal{R}^*_{\ell_{0-1}}(\mathcal{H}_{\text{all}}) \leq \sqrt{2}\left[\mathcal{R}_{\ell_{\log}}(h) - \mathcal{R}^*_{\ell_{\log}}(\mathcal{H}_{\text{all}})\right]^{\frac{1}{2}}$, which is not specific to the hypothesis set $\mathcal{H}$ and thus not as significant. In contrast, our $\mathcal{H}_{\text{lin}}$-consistency bound is finer and takes into account the role of the parameter $B$ as well as the number of labels $n$; iv) [Zheng et al., 2023, Theorem 3.3] only offers approximate bounds that are not tight; in contrast, by Theorem 5, our bound is tight.

Note that our $\mathcal{H}$-consistency bound in Theorem 6 are not limited to specific hypothesis set forms. They are directly applicable to various types of hypothesis sets including neural networks. For example, the same derivation can be extended to one-hidden-layer neural networks studied in [Awasthi et al., 2022a] and their multi-class generalization by calculating and substituting the corresponding $\Lambda(x)$. As a result, we can obtain novel and tight $\mathcal{H}$-consistency bounds for bounded neural network hypothesis sets in multi-class classification, which highlights the versatility of our general tools.

**B. Example: sum exponential loss**. We then consider the sum exponential loss, that is $\ell^{\text{comp}}$ with $\Phi(u) = \frac{1-u}{u}$. By computing the error transformation in Theorem 5, we obtain the following result.

**Theorem 9 ($\overline{\mathcal{H}}$-consistency bounds for sum exponential loss).** *For any $h \in \mathcal{H}$ and any distribution,*

$$\mathcal{R}_{\ell_{0-1}}(h) - \mathcal{R}^*_{\ell_{0-1}}(\overline{\mathcal{H}}) + \mathcal{M}_{\ell_{0-1}}(\overline{\mathcal{H}}) \leq \Psi^{-1}\left(\mathcal{R}_{\ell_{\exp}}(h) - \mathcal{R}^*_{\ell_{\exp}}(\overline{\mathcal{H}}) + \mathcal{M}_{\ell_{\exp}}(\overline{\mathcal{H}})\right)$$

*where $\ell_{\exp} = \sum_{y' \neq y} e^{h(x,y')-h(x,y)}$ and $\Psi(t) = \begin{cases} 1 - \sqrt{1-t^2} & t \leq \frac{s_{\max}^2 - s_{\min}^2}{s_{\min}^2 + s_{\max}^2} \\ \frac{s_{\max}-s_{\min}}{2s_{\max}s_{\min}}t - \frac{(s_{\max}-s_{\min})^2}{2s_{\max}s_{\min}(s_{\max}+s_{\min})} & \text{otherwise.} \end{cases}$*

The proof of Theorem 9 is given in Appendix E.3. Observe that $1 - \sqrt{1-t^2} \geq t^2/2$. By Theorem 9, making $s_{\min}$ close to zero, that is making $\Lambda$ close to infinite for any $x \in \mathcal{X}$, essentially turns $\Psi$ into a square function for all values. In general, $\Lambda$ is not infinite since a regularization is used in practice, which controls both the complexity of the hypothesis set and the magnitude of the scores.

**C. Example: generalized cross-entropy loss and mean absolute error loss**. Due to space limitations, we present the results for these loss functions in Appendix E.

## 5 Constrained losses

In this section, we present a general characterization of $\mathcal{H}$-consistency bounds for *constrained loss*, that is loss functions defined via a constraint, as in [Lee et al., 2004]:

$$\ell^{\text{cstnd}}(h, x, y) = \sum_{y' \neq y} \Phi(-h(x, y')) \tag{5}$$

with the constraint that $\sum_{y \in \mathcal{Y}} h(x, y) = 0$ for a non-negative and non-increasing auxiliary function $\Phi$.

### 5.1 $\mathcal{H}$-consistency bounds

As in the previous section, we prove a result that supplies a very general tool, an *error transformation function* for deriving $\mathcal{H}$-consistency bounds for constrained losses. When $\mathcal{T}^{\text{cstnd}}$ is convex, to make these guarantees explicit, we only need to calculate $\mathcal{T}^{\text{cstnd}}$.

Table 2: $\mathcal{H}$-estimation error transformation for common constrained losses.

| Auxiliary function $\Phi$ | $\Phi_{\exp}(t) = e^{-t}$ | $\Phi_{\text{hinge}}(t) = \max\{0, 1-t\}$ | $\Phi_{\text{sq-hinge}}(t) = (1-t)^2 \mathbb{1}_{t \leq 1}$ | $\Phi_{\text{sq}} = (1-t)^2$ |
|---|---|---|---|---|
| Transformation $\mathcal{T}^{\text{cstnd}}$ | $\mathcal{T}^{\text{cstnd}}(t) = 2 - \sqrt{4 - t^2}$ | $\mathcal{T}^{\text{cstnd}}(t) = t$ | $\mathcal{T}^{\text{cstnd}}(t) = \frac{t^2}{2}$ | $\mathcal{T}^{\text{cstnd}}(t) = \frac{t^2}{2}$ |

**Theorem 10** ($\mathcal{H}$-**consistency bound for constrained losses**). *Assume that $\mathcal{H}$ is symmetric and complete. Assume that $\mathcal{T}^{\text{cstnd}}$ is convex. Then, for any hypothesis $h \in \mathcal{H}$ and any distribution,*

$$\mathcal{T}^{\text{cstnd}}\big(\mathcal{R}_{\ell_{0-1}}(h) - \mathcal{R}^*_{\ell_{0-1}}(\mathcal{H}) + \mathcal{M}_{\ell_{0-1}}(\mathcal{H})\big) \leq \mathcal{R}_{\ell^{\text{cstnd}}}(h) - \mathcal{R}^*_{\ell^{\text{cstnd}}}(\mathcal{H}) + \mathcal{M}_{\ell^{\text{cstnd}}}(\mathcal{H})$$

*with $\mathcal{H}$-estimation error transformation for constrained losses defined on $t \in [0,1]$ by $\mathcal{T}^{\text{cstnd}}(t) =$*

$$\begin{cases} \inf_{\tau \geq 0} \sup_{\mu \in \mathbb{R}} \left\{ \frac{1-t}{2}\big[\Phi(\tau) - \Phi(-\tau + \mu)\big] + \frac{1+t}{2}\big[\Phi(-\tau) - \Phi(\tau - \mu)\big] \right\} & n = 2 \\ \inf_{P \in [\frac{1}{n-1}, 1]} \inf_{\tau_1 \geq \max\{\tau_2, 0\}} \sup_{\mu \in \mathbb{R}} \left\{ \frac{2-P-t}{2}\big[\Phi(-\tau_2) - \Phi(-\tau_1 + \mu)\big] + \frac{2-P+t}{2}\big[\Phi(-\tau_1) - \Phi(-\tau_2 - \mu)\big] \right\} & n > 2. \end{cases}$$

*Furthermore, for any $t \in [0,1]$, there exist a distribution $\mathcal{D}$ and a hypothesis $h \in \mathcal{H}$ such that $\mathcal{R}_{\ell_{0-1}}(h) - \mathcal{R}^*_{\ell_{0-1}}(\mathcal{H}) + \mathcal{M}_{\ell_{0-1}}(\mathcal{H}) = t$ and $\mathcal{R}_{\ell^{\text{cstnd}}}(h) - \mathcal{R}^*_{\ell^{\text{cstnd}}}(\mathcal{H}) + \mathcal{M}_{\ell^{\text{cstnd}}}(\mathcal{H}) = \mathcal{T}^{\text{cstnd}}(t)$.*

Here too, the theorem guarantees the tightness of the bound. This general expression of $\mathcal{T}^{\text{cstnd}}$ can be considerably simplified under some broad assumptions, as shown by the following result.

**Theorem 11** (**characterization of** $\mathcal{T}^{\text{cstnd}}$). *Assume that $\Phi$ is convex, differentiable at zero and $\Phi'(0) < 0$. Then, $\mathcal{T}^{\text{cstnd}}$ can be expressed as follows:*

$$\mathcal{T}^{\text{cstnd}}(t) = \begin{cases} \Phi(0) - \inf_{\mu \in \mathbb{R}} \left\{ \frac{1-t}{2}\Phi(\mu) + \frac{1+t}{2}\Phi(-\mu) \right\} & n = 2 \\ \inf_{\tau \geq 0} \left\{ \left(2 - \frac{1}{n-1}\right)\Phi(-\tau) - \inf_{\mu \in \mathbb{R}} \left\{ \frac{2-t-\frac{1}{n-1}}{2}\Phi(-\tau + \mu) + \frac{2+t-\frac{1}{n-1}}{2}\Phi(-\tau - \mu) \right\} \right\} & n > 2 \end{cases}$$

$$\geq \begin{cases} \Phi(0) - \inf_{\mu \in \mathbb{R}} \left\{ \frac{1-t}{2}\Phi(\mu) + \frac{1+t}{2}\Phi(-\mu) \right\} & n = 2 \\ \inf_{\tau \geq 0} \left\{ 2\Phi(-\tau) - \inf_{\mu \in \mathbb{R}} \left\{ \frac{2-t}{2}\Phi(-\tau + \mu) + \frac{2+t}{2}\Phi(-\tau - \mu) \right\} \right\} & n > 2. \end{cases}$$

The proof of all the results in this section are given in Appendix D.

**Examples.** We now compute the $\mathcal{H}$-estimation error transformation for constrained losses and present the results in Table 2. Here, we present the simplified $\mathcal{T}^{\text{cstnd}}$ by using the lower bound in Theorem 11. Remarkably, by applying Theorem 10, we are able to obtain tighter $\mathcal{H}$-consistency bounds for constrained losses with $\Phi = \Phi_{\text{hinge}}, \Phi_{\text{sq-hinge}}, \Phi_{\exp}$ than those derived using ad hoc methods in [Awasthi et al., 2022b], and a novel $\mathcal{H}$-consistency bound for the new constrained loss $\ell^{\text{cstnd}}(h, x, y) = \sum_{y' \neq y} (1 + h(x, y'))^2$ with $\Phi(t) = (1-t)^2$.

### 5.2 Extension to non-complete or bounded hypothesis sets

As in the case of comp-sum losses, we extend our results beyond the completeness assumption adopted in previous work and establish $\overline{\mathcal{H}}$-consistency bounds for bounded hypothesis sets. This significantly broadens the applicability of our framework.

**Theorem 12** ($\overline{\mathcal{H}}$-**consistency bound for constrained losses**). *Assume that $\overline{\mathcal{T}}^{\text{cstnd}}$ is convex. Then, the following inequality holds for any hypothesis $h \in \overline{\mathcal{H}}$ and any distribution:*

$$\overline{\mathcal{T}}^{\text{cstnd}}\big(\mathcal{R}_{\ell_{0-1}}(h) - \mathcal{R}^*_{\ell_{0-1}}(\overline{\mathcal{H}}) + \mathcal{M}_{\ell_{0-1}}(\overline{\mathcal{H}})\big) \leq \mathcal{R}_{\ell^{\text{cstnd}}}(h) - \mathcal{R}^*_{\ell^{\text{cstnd}}}(\overline{\mathcal{H}}) + \mathcal{M}_{\ell^{\text{cstnd}}}(\overline{\mathcal{H}}). \quad (6)$$

*with $\overline{\mathcal{T}}^{\text{cstnd}}$ the $\overline{\mathcal{H}}$-estimation error transformation for constrained losses defined for all $t \in [0,1]$ by $\overline{\mathcal{T}}^{\text{cstnd}}(t) =$*

$$\begin{cases} \inf_{\tau \geq 0} \sup_{\mu \in [\tau - \Lambda_{\min}, \tau + \Lambda_{\min}]} \left\{ \frac{1-t}{2}\big[\Phi(\tau) - \Phi(-\tau + \mu)\big] + \frac{1+t}{2}\big[\Phi(-\tau) - \Phi(\tau - \mu)\big] \right\} & n = 2 \\ \inf_{P \in [\frac{1}{n-1}, 1]} \inf_{\tau_1 \geq \max\{\tau_2, 0\}} \sup_{\mu \in C} \left\{ \frac{2-P-t}{2}\big[\Phi(-\tau_2) - \Phi(-\tau_1 + \mu)\big] + \frac{2-P+t}{2}\big[\Phi(-\tau_1) - \Phi(-\tau_2 - \mu)\big] \right\} & n > 2, \end{cases}$$

*where $C = [\max\{\tau_1, -\tau_2\} - \Lambda_{\min}, \min\{\tau_1, -\tau_2\} + \Lambda_{\min}]$ and $\Lambda_{\min} = \inf_{x \in \mathcal{X}} \Lambda(x)$. Furthermore, for any $t \in [0,1]$, there exist a distribution $\mathcal{D}$ and a hypothesis $h \in \mathcal{H}$ such that $\mathcal{R}_{\ell_{0-1}}(h) - \mathcal{R}^*_{\ell_{0-1}}(\mathcal{H}) + \mathcal{M}_{\ell_{0-1}}(\mathcal{H}) = t$ and $\mathcal{R}_{\ell^{\text{cstnd}}}(h) - \mathcal{R}^*_{\ell^{\text{cstnd}}}(\mathcal{H}) + \mathcal{M}_{\ell^{\text{cstnd}}}(\mathcal{H}) = \mathcal{T}^{\text{cstnd}}(t)$.*

The proof is presented in Appendix F.1. Next, we illustrate the application of our theory through an example of constrained exponential losses, that is $\ell^{\text{cstnd}}$ with $\Phi(t) = e^{-t}$. By using the error transformation in Theorem 12, we obtain new $\overline{\mathcal{H}}$-consistency bounds in Theorem 13 (see Appendix F.2 for the proof) for bounded hypothesis sets $\overline{\mathcal{H}}$.

**Theorem 13** ($\overline{\mathcal{H}}$**-consistency bounds for constrained exponential loss**)**.** *Let* $\Phi(t) = e^{-t}$. *For any* $h \in \overline{\mathcal{H}}$ *and any distribution,*

$$\mathcal{R}_{\ell_{0-1}}(h) - \mathcal{R}^*_{\ell_{0-1}}(\overline{\mathcal{H}}) + \mathcal{M}_{\ell_{0-1}}(\overline{\mathcal{H}}) \le \Psi^{-1}\big(\mathcal{R}_{\ell^{\text{cstnd}}}(h) - \mathcal{R}^*_{\ell^{\text{cstnd}}}(\overline{\mathcal{H}}) + \mathcal{M}_{\ell^{\text{cstnd}}}(\overline{\mathcal{H}})\big)$$

*where* $\Psi(t) = \begin{cases} 1 - \sqrt{1-t^2} & t \le \frac{e^{2\Lambda_{\min}}-1}{e^{2\Lambda_{\min}}+1} \\ \frac{t}{2}\big(e^{\Lambda_{\min}} - e^{-\Lambda_{\min}}\big) + \frac{2-e^{\Lambda_{\min}}-e^{-\Lambda_{\min}}}{2} & \text{otherwise.} \end{cases}$

Awasthi et al. [2022b] proves $\mathcal{H}$-consistency bounds for constrained exponential losses when $\mathcal{H}$ is symmetric and complete. Theorem 13 significantly generalizes those results to the non-complete hypothesis sets. Different from the complete case, the functional form of our new bounds has two pieces which corresponds to the linear and the square root convergence respectively, modulo the constants. Furthermore, the coefficient of the linear piece depends on the the magnitude of $\Lambda_{\min}$. When $\Lambda_{\min}$ is small, the coefficient of the linear term in $\Psi^{-1}$ explodes. On the other hand, making $\Lambda_{\min}$ large essentially turns $\Psi^{-1}$ into a square-root function.

## 6 Discussion

Here, we further elaborate on the practical value of our tools and $\mathcal{H}$-consistency bounds. Our contributions include a more general and convenient mathematical tool for proving $\mathcal{H}$-consistency bounds, along with tighter bounds that enable a better comparison of surrogate loss functions and extensions beyond previous completeness assumptions. As mentioned by [Awasthi et al., 2022b], given a hypothesis set $\mathcal{H}$, $\mathcal{H}$-consistency bounds can be used to compare different surrogate loss functions and select the most favorable one, which depends on the functional form of the $\mathcal{H}$-consistency bound; the smoothness of the surrogate loss and its optimization properties; approximation properties of the surrogate loss function controlled by minimizability gaps; and the dependency on the number of classes in the multiplicative constant. Consequently, a tighter $\mathcal{H}$-consistency bound provides a more accurate comparison, as a loose bound might not adequately capture the full advantage of using one surrogate loss. In contrast, Bayes-consistency does not take into account the hypothesis set and is an asymptotic property, thereby failing to guide the comparison of different surrogate losses.

Another application of our $\mathcal{H}$-consistency bounds involves deriving generalization bounds for surrogate loss minimizers [Mao et al., 2023h], expressed in terms of the same quantities previously discussed. Therefore, when dealing with finite samples, a tighter $\mathcal{H}$-consistency bound could also result in a corresponding tighter generalization bound. Moreover, our novel results extend beyond previous completeness assumptions, offering guarantees applicable to bounded hypothesis sets commonly used with regularization. This enhancement provides meaningful learning guarantees. Technically, our error transformation function serves as a very general tool for deriving $\mathcal{H}$-consistency bounds with tightness guarantees. These functions are defined within each class of loss functions including comp-sum losses and constrained losses, and their formulation depends on the structure of the individual loss function class, the range of the hypothesis set and the number of classes. To derive explicit bounds, all that is needed is to calculate these error transformation functions. Under some broad assumptions on the auxiliary function within a loss function, these error transformation functions can be further distilled into more simplified forms, making them straightforward to compute.

## 7 Conclusion

We presented a general characterization and extension of $\mathcal{H}$-consistency bounds for multi-class classification. We introduced new tools for deriving such bounds with tightness guarantees and illustrated their benefits through several applications and examples. Our proposed method is a significant advance in the theory of $\mathcal{H}$-consistency bounds for multi-class classification. It can provide a general and powerful tool for deriving tight bounds for a wide variety of other loss functions and hypothesis sets. We believe that our work will open up new avenues of research in the field of multi-class classification consistency.

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

# Contents of Appendix

# A   Related work

The notions of Bayes-consistency (also known as consistency) and calibration have been well studied not only with respect to the binary zero-one loss [Zhang, 2004a, Bartlett et al., 2006, Steinwart, 2007, Mohri et al., 2018], but also with respect to the multi-class zero-one loss [Zhang, 2004b, Tewari and Bartlett, 2007], the general multi-class losses [Ramaswamy and Agarwal, 2012, Narasimhan et al., 2015, Ramaswamy and Agarwal, 2016], the multi-class SVMs [Chen and Sun, 2006, Chen and Xiang, 2006, Liu, 2007, Dogan et al., 2016, Wang and Scott, 2020], the gamma-phi losses [Wang and Scott, 2023], the multi-label losses [Gao and Zhou, 2011, Dembczynski et al., 2012, Zhang et al., 2020], the losses with a reject option [Ramaswamy et al., 2015, Cortes et al., 2016a,b, 2023], the ranking losses [Ravikumar et al., 2011, Ramaswamy et al., 2013, Gao and Zhou, 2015, Uematsu and Lee, 2017], the cost sensitive losses [Pires et al., 2013, Pires and Szepesvári, 2016], the structured losses [Ciliberto et al., 2016, Osokin et al., 2017, Blondel, 2019], the polyhedral losses [Frongillo and Waggoner, 2021, Finocchiaro et al., 2022], the Top-$k$ classification losses [Thilagar et al., 2022], the proper losses [Agarwal and Agarwal, 2015, Williamson et al., 2016] and the losses of ordinal regression [Pedregosa et al., 2017].

Bayes-consistency only holds for the full family of measurable functions, which of course is distinct from the more restricted hypothesis set used by a learning algorithm. Therefore, a hypothesis set-dependent notion of $\mathcal{H}$-consistency has been proposed by Long and Servedio [2013] in the realizable setting, used by Zhang and Agarwal [2020] for linear models, and generalized by Kuznetsov et al. [2014] to the structured prediction case. Long and Servedio [2013] showed that there exists a case where a Bayes-consistent loss is not $\mathcal{H}$-consistent while inconsistent losses can be $\mathcal{H}$-consistent. Zhang and Agarwal [2020] further investigated the phenomenon in [Long and Servedio, 2013] and showed that the situation of losses that are not $\mathcal{H}$-consistent with linear models can be remedied by carefully choosing a larger piecewise linear hypothesis set. Kuznetsov et al. [2014] proved positive results for the $\mathcal{H}$-consistency of several multi-class ensemble algorithms, as an extension of $\mathcal{H}$-consistency results in [Long and Servedio, 2013].

Recently, Awasthi et al. [2022a,b], Mao et al. [2023h], Zheng et al. [2023] presented a series of results providing $\mathcal{H}$-consistency bounds. These are upper bounds on the zero-one estimation error of any predictor in a hypothesis set, expressed in terms of its surrogate loss estimation error. They are more informative guarantees than similar excess error bounds derived in the literature, which correspond to the special case where $\mathcal{H}$ is the family of all measurable functions [Zhang, 2004a, Bartlett et al., 2006, Mohri et al., 2018]. Awasthi et al. [2022a] studied $\mathcal{H}$-consistency bounds in binary classification. They provided a series of *tight* $\mathcal{H}$-consistency bounds for *bounded* hypothesis set of linear models and one-hidden-layer neural networks. The subsequent study [Awasthi et al., 2022b] further generalized the framework to multi-class classification, where they presented a extensive study of $\mathcal{H}$-consistency bounds for diverse multi-class surrogate losses including negative results for *max losses* [Crammer and Singer, 2001] and positive results for *sum losses* [Weston and Watkins, 1998], and *constrained losses* [Lee et al., 2004]. However, the hypothesis sets adopted there were assumed to be complete, which rules out the bounded hypothesis sets typically used in practice. Moreover, the final bounds derived from [Awasthi et al., 2022b] are based on ad hoc methods and may not be tight. [Mao et al., 2023h] complemented the previous work by studying a wide family of *comp-sum losses* in the multi-class classification, which generalized the *sum-losses* and included as special cases the logistic loss [Verhulst, 1838, 1845, Berkson, 1944, 1951], the *generalized cross-entropy loss* [Zhang and Sabuncu, 2018], and the *mean absolute error loss* [Ghosh et al., 2017]. Here too, the completeness assumption on the hypothesis sets was adopted and their $\mathcal{H}$-consistency bounds do not apply to common bounded hypothesis sets in practice. Zheng et al. [2023] proved $\mathcal{H}$-consistency bounds for multi-class logistic loss with bounded linear hypothesis sets. However, their bounds require a crucial distributional assumption under which, the minimizability gaps coincide with the approximation errors. Thus, their bounds can be recovered as excess error bounds, which are less significant.

This paper provides both a general characterization and an extension of $\mathcal{H}$-consistency bounds for multi-class classification. Our general tools and tight bounds show several remarkable advantages: first, they improve existing bounds for complete hypothesis sets previously proven in [Awasthi et al., 2022b], second, they encompass all previously comp-sum and constrained losses studied thus far as well as many new ones [Awasthi et al., 2022a, Mao et al., 2023h], third, they extend beyond the completeness assumption adopted in previous work, fourth, they give novel guarantees for bounded

hypothesis sets, and finally they help prove a much stronger and more significant guarantee for logistic loss with linear hypothesis set than [Zheng et al., 2023].

Other related work on $\mathcal{H}$-consistency bounds includes: $\mathcal{H}$-consistency bounds for pairwise ranking [Mao et al., 2023d,e]; theoretically grounded surrogate losses and algorithms for learning with abstention supported by $\mathcal{H}$-consistency bounds, including the study of score-based abstention [Mao et al., 2023f], predictor-rejector abstention [Mao et al., 2023c] and learning to abstain with a fixed predictor with application in decontextualization [Mohri et al., 2023]; principled approaches for learning to defer with multiple experts that benefit from strong $\mathcal{H}$-consistency bounds, including the single-stage scenario [Mao et al., 2023b] and a two-stage scenario [Mao et al., 2023a]; $\mathcal{H}$-consistency theory and algorithms for adversarial robustness [Awasthi et al., 2021a,b, 2023a, Mao et al., 2023h, Awasthi et al., 2023b]; and efficient algorithms and loss functions for structured prediction with stronger $\mathcal{H}$-consistency guarantees [Mao et al., 2023g].

## B   Minimizability gap

This is a brief discussion of minimizability gaps and their properties. By definition, for any loss function $\ell$, the minimizability gap is defined by

$$\mathcal{M}_\ell(\mathcal{H}) = \inf_{h \in \mathcal{H}} \left\{ \mathbb{E}_{(x,y) \sim \mathcal{D}} [\ell(h,x,y)] \right\} - \mathbb{E}_x \left[ \inf_{h \in \mathcal{H}} \mathbb{E}_{y|x} [\ell(h,x,y)] \right] = \mathcal{R}_\ell^*(\mathcal{H}) - \mathbb{E}_x [\mathcal{C}_\ell^*(\mathcal{H},x)].$$

### B.1   Zero minimizability

**Lemma 14.** *Let $\ell$ be a surrogate loss such that for $(x,y) \in \mathcal{X} \times \mathcal{Y}$ and any measurable function $h \in \mathcal{H}_{\text{all}}$, the loss $\ell(h,x,y)$ only depends on $h(x)$ and $y$ (thus we can write $\ell(h,x,y) = \overline{\ell}(h(x),y)$ for some function $\overline{\ell}$). Then, the minimizabiliy gap vanishes: $\mathcal{M}_\ell(\mathcal{H}_{\text{all}}) = 0$.*

*Proof.* Fix $\epsilon > 0$. Then, by definition of the infimum, for any $x \in \mathcal{X}$, there exists $h_x \in \mathcal{H}_{\text{all}}$ such that

$$\mathbb{E}_{y|x} [\ell(h_x,x,y)] \leq \inf_{h \in \mathcal{H}_{\text{all}}} \mathbb{E}_{y|x} [\ell(h,x,y)] + \epsilon.$$

Now, define the function $h$ by $h(x) = h_x(x)$, for all $x \in \mathcal{X}$. $h$ can be shown to be measurable, for example, when $\mathcal{X}$ admits a countable dense subset. Then,

$$\begin{aligned}
\mathbb{E}_{(x,y) \sim \mathcal{D}} [\ell(h,x,y)] = \mathbb{E}_{(x,y) \sim \mathcal{D}} [\overline{\ell}(h(x),y)] &= \mathbb{E}_{(x,y) \sim \mathcal{D}} [\overline{\ell}(h_x(x),y)] \\
&= \mathbb{E}_{(x,y) \sim \mathcal{D}} [\ell(h_x,x,y)] \\
&\leq \mathbb{E}_x \left[ \inf_{h \in \mathcal{H}_{\text{all}}} \mathbb{E}_{y|x} [\ell(h,x,y)] + \epsilon \right] \\
&= \mathbb{E}_x [\mathcal{C}_\ell^*(\mathcal{H}_{\text{all}},x)] + \epsilon.
\end{aligned}$$

Thus, we have

$$\inf_{h \in \mathcal{H}_{\text{all}}} \mathbb{E}_{(x,y) \sim \mathcal{D}} [\ell(h,x,y)] \leq \mathbb{E}_x [\mathcal{C}_\ell^*(\mathcal{H}_{\text{all}},x)] + \epsilon.$$

Since the inequality holds for any $\epsilon > 0$, we have $\mathcal{R}_\ell^*(\mathcal{H}_{\text{all}}) = \inf_{h \in \mathcal{H}_{\text{all}}} \mathbb{E}_{(x,y) \sim \mathcal{D}} [\ell(h,x,y)] \leq \mathbb{E}_x [\mathcal{C}_\ell^*(\mathcal{H}_{\text{all}},x)]$. This implies equality since the inequality $\mathcal{R}_\ell^*(\mathcal{H}) \geq \mathbb{E}_x [\mathcal{C}_\ell^*(\mathcal{H},x)]$ holds for any $\mathcal{H}$. $\square$

### B.2   Relationship with approximation error

Let $\mathcal{A}_\ell$ denote the approximation error of a loss function $\ell$ and a hypothesis set $\mathcal{H}$: $\mathcal{A}_\ell(\mathcal{H}) = \mathcal{R}_\ell^*(\mathcal{H}) - \mathcal{R}_\ell^*(\mathcal{H}_{\text{all}})$. We will denote by $I_\ell(\mathcal{H})$ the difference of pointwise infima $I_\ell(\mathcal{H}) = \mathbb{E}_x [\mathcal{C}_\ell^*(\mathcal{H},x) - \mathcal{C}_\ell^*(\mathcal{H}_{\text{all}},x)]$, which is non-negative. The minimizability gap can be decomposed as

follows in terms of the approximation error and the difference of pointwise infima:

$$
\begin{aligned}
\mathcal{M}_\ell(\mathcal{H}) &= \mathcal{R}_\ell^*(\mathcal{H}) - \mathbb{E}_x\left[\mathcal{C}_\ell^*(\mathcal{H}, x)\right] \\
&= \mathcal{R}_\ell^*(\mathcal{H}) - \mathcal{R}_\ell^*(\mathcal{H}_{\text{all}}) + \mathcal{R}_\ell^*(\mathcal{H}_{\text{all}}) - \mathbb{E}_x\left[\mathcal{C}_\ell^*(\mathcal{H}, x)\right] \\
&= \mathcal{A}_\ell(\mathcal{H}) + \mathcal{R}_\ell^*(\mathcal{H}_{\text{all}}) - \mathbb{E}_x\left[\mathcal{C}_\ell^*(\mathcal{H}, x)\right] \\
&= \mathcal{A}_\ell(\mathcal{H}) + \mathbb{E}_x\left[\mathcal{C}_\ell^*(\mathcal{H}_{\text{all}}, x) - \mathcal{C}_\ell^*(\mathcal{H}, x)\right] \qquad \text{(By Lemma 14)} \\
&= \mathcal{A}_\ell(\mathcal{H}) - I_\ell(\mathcal{H}).
\end{aligned}
$$

The decomposition immediately implies the following result.

**Lemma 15.** *Let $\ell$ be a surrogate loss such that for $(x, y) \in \mathcal{X} \times \mathcal{Y}$ and any measurable function $h \in \mathcal{H}_{\text{all}}$, the loss $\ell(h, x, y)$ only depends on $h(x)$ and $y$ (thus we can write $\ell(h, x, y) = \bar{\ell}(h(x), y)$ for some function $\bar{\ell}$). Then, for any loss function $\ell$ and hypothesis set $\mathcal{H}$, we have: $\mathcal{M}_\ell(\mathcal{H}) \leq \mathcal{A}_\ell(\mathcal{H})$.*

By Lemma 1, when $\ell$ is the zero-one loss, $I_\ell(\mathcal{H}) = 0$ when the hypothesis set generates labels that cover all possible outcomes for each input. However, for a surrogate loss function, $I_\ell(\mathcal{H})$ is non-negative, and is generally non-zero.

Take the example of binary classification and denote the conditional distribution as $\eta(x) = D(Y = 1 | X = x)$. Let $\mathcal{H}$ be a family of functions $h$ with $|h(x)| \leq \Lambda$ for all $x \in \mathcal{X}$ and such that all values in $[-\Lambda, +\Lambda]$ can be reached. Consider for example the exponential-based margin loss: $\ell(h, x, y) = e^{-yh(x)}$. Then, $\mathcal{C}_\ell(h, x) = \eta(x)e^{-h(x)} + (1 - \eta(x))e^{h(x)}$. Upon observing this, it becomes apparent that the infimum over all measurable functions can be expressed in the following way, for all $x$:

$$
\mathcal{C}_\ell^*(\mathcal{H}_{\text{all}}, x) = 2\sqrt{\eta(x)(1 - \eta(x))},
$$

while the infimum over $\mathcal{H}$, $\mathcal{C}_\ell^*(\mathcal{H}, x)$, depends on $\Lambda$ and can be expressed as

$$
\mathcal{C}_\ell^*(\mathcal{H}, x) = \begin{cases} \max\{\eta(x), 1 - \eta(x)\}e^{-\Lambda} + \min\{\eta(x), 1 - \eta(x)\}e^{\Lambda} & \Lambda < \frac{1}{2}\left|\log\frac{\eta(x)}{1-\eta(x)}\right| \\ 2\sqrt{\eta(x)(1 - \eta(x))} & \text{otherwise.} \end{cases}
$$

Thus, in the deterministic scenario,

$$
I_\ell(\mathcal{H}) = \mathbb{E}_x\left[\mathcal{C}_\ell^*(\mathcal{H}, x) - \mathcal{C}_\ell^*(\mathcal{H}_{\text{all}}, x)\right] = e^{-\Lambda}.
$$

### B.3   Significance of $\mathcal{H}$-consistency bounds

As shown in the previous section, for target loss $\ell_{0-1}$, the minimizability gap coincides with the approximation error $\mathcal{M}_{\ell_{0-1}}(\mathcal{H}) = \mathcal{A}_{\ell_{0-1}}(\mathcal{H})$ when the hypothesis set generates labels that cover all possible outcomes for each input. However, for a surrogate loss $\ell$, the minimizability gap is generally strictly less than the approximation error $\mathcal{M}_\ell(\mathcal{H}) < \mathcal{A}_\ell(\mathcal{H})$. Thus, an $\mathcal{H}$-consistency bound, expressed as follows for some increasing function $\Gamma$:

$$
\mathcal{R}_{\ell_{0-1}}(h) - \mathcal{R}_{\ell_{0-1}}^*(\mathcal{H}) + \mathcal{M}_{\ell_{0-1}}(\mathcal{H}) \leq \Gamma(\mathcal{R}_\ell(h) - \mathcal{R}_\ell^*(\mathcal{H}) + \mathcal{M}_\ell(\mathcal{H})).
$$

is more favorable than an excess error bound expressed in terms of approximation errors:

$$
\mathcal{R}_{\ell_{0-1}}(h) - \mathcal{R}_{\ell_{0-1}}^*(\mathcal{H}) + \mathcal{A}_{\ell_{0-1}}(\mathcal{H}) \leq \Gamma(\mathcal{R}_\ell(h) - \mathcal{R}_\ell^*(\mathcal{H}) + \mathcal{A}_\ell(\mathcal{H})).
$$

Here, $\Gamma$ is typically linear or the square-root function modulo constants. When $\mathcal{H} = \mathcal{H}_{\text{all}}$, the family of all measurable functions, by Lemma 14, the $\mathcal{H}$-consistency bound coincides with the excess error bound and implies Bayes-consistency by taking the limit. It is therefore a stronger guarantee than an excess error bound and Bayes-consistency.

## C   Proofs for comp-sum losses

Let $y_{\max} = \operatorname{argmax}_{y \in \mathcal{Y}} p(x, y)$ and $\mathsf{h}(x) = \operatorname{argmax}_{y \in \mathcal{Y}} h(x, y)$, where we choose the label with the highest index under the natural ordering of labels as the tie-breaking strategy.

## C.1 Proof of $\mathcal{H}$-consistency bounds with $\mathcal{T}^{\mathrm{comp}}$ (Theorem 2)

**Theorem 2** (**$\mathcal{H}$-consistency bound for comp-sum losses**). *Assume that $\mathcal{H}$ is symmetric and complete and that $\mathcal{T}^{\mathrm{comp}}$ is convex. Then, the following inequality holds for any hypothesis $h \in \mathcal{H}$ and any distribution*

$$\mathcal{T}^{\mathrm{comp}}\left(\mathcal{R}_{\ell_{0-1}}(h) - \mathcal{R}^*_{\ell_{0-1}}(\mathcal{H}) + \mathcal{M}_{\ell_{0-1}}(\mathcal{H})\right) \le \mathcal{R}_{\ell^{\mathrm{comp}}}(h) - \mathcal{R}^*_{\ell^{\mathrm{comp}}}(\mathcal{H}) + \mathcal{M}_{\ell^{\mathrm{comp}}}(\mathcal{H}), \qquad (3)$$

*with $\mathcal{T}^{\mathrm{comp}}$ an $\mathcal{H}$-estimation error transformation for comp-sum losses defined for all $t \in [0,1]$ by*

$$\mathcal{T}^{\mathrm{comp}}(t) =$$
$$\begin{cases} \displaystyle\inf_{\tau\in[0,\frac{1}{2}]} \sup_{\mu\in[-\tau,1-\tau]} \left\{\frac{1+t}{2}[\Phi(\tau) - \Phi(1-\tau-\mu)] + \frac{1-t}{2}[\Phi(1-\tau) - \Phi(\tau+\mu)]\right\} & n = 2 \\ \displaystyle\inf_{P\in[\frac{1}{n-1}\vee t, 1]} \inf_{\substack{\tau_1\ge\max(\tau_2,1/n) \\ \tau_1+\tau_2\le 1, \tau_2\ge 0}} \sup_{\mu\in[-\tau_2,\tau_1]} \left\{\frac{P+t}{2}[\Phi(\tau_2) - \Phi(\tau_1-\mu)] + \frac{P-t}{2}[\Phi(\tau_1) - \Phi(\tau_2+\mu)]\right\} & n > 2. \end{cases}$$

*Furthermore, for any $t \in [0,1]$, there exist a distribution $\mathcal{D}$ and a hypothesis $h \in \mathcal{H}$ such that $\mathcal{R}_{\ell_{0-1}}(h) - \mathcal{R}^*_{\ell_{0-1}}(\mathcal{H}) + \mathcal{M}_{\ell_{0-1}}(\mathcal{H}) = t$ and $\mathcal{R}_{\ell^{\mathrm{comp}}}(h) - \mathcal{R}^*_{\ell^{\mathrm{comp}}}(\mathcal{H}) + \mathcal{M}_{\ell^{\mathrm{comp}}}(\mathcal{H}) = \mathcal{T}^{\mathrm{comp}}(t)$.*

*Proof.* For the comp-sum loss $\ell^{\mathrm{comp}}$, the conditional $\ell^{\mathrm{comp}}$-risk can be expressed as follows:

$$\begin{aligned} \mathcal{C}_{\ell^{\mathrm{comp}}}(h, x) &= \sum_{y\in\mathcal{Y}} p(x,y)\ell^{\mathrm{comp}}(h, x, y) \\ &= \sum_{y\in\mathcal{Y}} p(x,y)\Phi\left(\frac{e^{h(x,y)}}{\sum_{y'\in\mathcal{Y}} e^{h(x,y')}}\right) \\ &= \sum_{y\in\mathcal{Y}} p(x,y)\Phi(S_h(x,y)) \\ &= p(x,y_{\max})\Phi(S_h(x,y_{\max})) + p(x,\mathsf{h}(x))\Phi(S_h(x,\mathsf{h}(x))) \\ &\quad + \sum_{y\notin\{y_{\max},\mathsf{h}(x)\}} p(x,y)\Phi(S_h(x,y)). \end{aligned}$$

where we let $S_h(x,y) = \frac{e^{h(x,y)}}{\sum_{y'\in\mathcal{Y}} e^{h(x,y')}}$ for any $y \in \mathcal{Y}$ with the constraint that $\sum_{y\in\mathcal{Y}} S_h(x,y) = 1$. For any $h \in \mathcal{H}$ such that $\mathsf{h}(x) \ne y_{\max}$ and $x \in \mathcal{X}$, we can always find a family of hypotheses $\{h_\mu\} \subset \mathcal{H}$ such that $S_{h,\mu}(x,\cdot) = \frac{e^{h_\mu(x,\cdot)}}{\sum_{y'\in\mathcal{Y}} e^{h_\mu(x,y')}}$ take the following values:

$$S_{h,\mu}(x,y) = \begin{cases} S_h(x,y) & \text{if } y \notin \{y_{\max}, \mathsf{h}(x)\} \\ S_h(x,y_{\max}) + \mu & \text{if } y = \mathsf{h}(x) \\ S_h(x,\mathsf{h}(x)) - \mu & \text{if } y = y_{\max}. \end{cases}$$

Note that $S_{h,\mu}$ satisfies the constraint:

$$\sum_{y\in\mathcal{Y}} S_{h,\mu}(x,y) = \sum_{y\in\mathcal{Y}} S_h(x,y) = 1.$$

Let $p_1 = p(x, y_{\max})$, $p_2 = p(x, \mathsf{h}(x))$, $\tau_1 = S_h(x, \mathsf{h}(x))$ and $\tau_2 = S_h(x, y_{\max})$ to simplify the notation. Then, by the definition of $S_{h,\mu}$, we have for any $h \in \mathcal{H}$ and $x \in \mathcal{X}$,

$$
\mathcal{C}_{\ell^{\mathrm{comp}}}(h, x) - \inf_{\mu \in [-\tau_2, \tau_1]} \mathcal{C}_{\ell^{\mathrm{comp}}}(h_\mu, x)
$$

$$
= \sup_{\mu \in [-\tau_2, \tau_1]} \left\{ p_1[\Phi(\tau_2) - \Phi(\tau_1 - \mu)] + p_2[\Phi(\tau_1) - \Phi(\tau_2 + \mu)] \right\}
$$

$$
= \sup_{\mu \in [-\tau_2, \tau_1]} \left\{ \frac{P + p_1 - p_2}{2}[\Phi(\tau_2) - \Phi(\tau_1 - \mu)] + \frac{P - p_1 + p_2}{2}[\Phi(\tau_1) - \Phi(\tau_2 + \mu)] \right\}
$$

$$
\left( P = p_1 + p_2 \in \left[ \tfrac{1}{n-1} \vee p_1 - p_2, 1 \right] \right)
$$

$$
\leq \inf_{P \in \left[ \frac{1}{n-1} \vee p_1 - p_2, 1 \right]} \inf_{\substack{\tau_1 \geq \max(\tau_2, 1/n) \\ \tau_1 + \tau_2 \leq 1, \tau_2 \geq 0}} \sup_{\mu \in [-\tau_2, \tau_1]} \left\{ \frac{P + p_1 - p_2}{2}[\Phi(\tau_2) - \Phi(\tau_1 - \mu)] \right.
$$

$$
\left. + \frac{P - p_1 + p_2}{2}[\Phi(\tau_1) - \Phi(\tau_2 + \mu)] \right\} \qquad (\tau_1 \geq \max(\tau_2, 1/n), \tau_1 + \tau_2 \leq 1, \tau_2 \geq 0)
$$

$$
= \mathcal{T}^{\mathrm{comp}}(p_1 - p_2)
$$

$$
= \mathcal{T}^{\mathrm{comp}}(\Delta \mathcal{C}_{\ell_{0-1}, \mathcal{H}}(h, x)), \qquad \text{(by Lemma 1)}
$$

where for $n = 2$, an additional constraint $\tau_1 + \tau_2 = 1$ is imposed and the expression of $\mathcal{T}^{\mathrm{comp}}$ is simplified. Since $\mathcal{T}^{\mathrm{comp}}$ is convex, by Jensen's inequality, we obtain for any hypothesis $h \in \mathcal{H}$ and any distribution,

$$
\mathcal{T}^{\mathrm{comp}}\big(\mathcal{R}_{\ell_{0-1}}(h) - \mathcal{R}^*_{\ell_{0-1}}(\mathcal{H}) + \mathcal{M}_{\ell_{0-1}}(\mathcal{H})\big)
$$

$$
= \mathcal{T}^{\mathrm{comp}}\Big( \mathbb{E}_X[\Delta \mathcal{C}_{\ell_{0-1}, \mathcal{H}}(h, x)] \Big)
$$

$$
\leq \mathbb{E}_X[\mathcal{T}^{\mathrm{comp}}(\Delta \mathcal{C}_{\ell_{0-1}, \mathcal{H}}(h, x))]
$$

$$
\leq \mathbb{E}_X[\Delta \mathcal{C}_{\ell^{\mathrm{comp}}, \mathcal{H}}(h, x)]
$$

$$
= \mathcal{R}_{\ell^{\mathrm{comp}}}(h) - \mathcal{R}^*_{\ell^{\mathrm{comp}}}(\mathcal{H}) + \mathcal{M}_{\ell^{\mathrm{comp}}}(\mathcal{H}).
$$

For the second part, we first consider $n = 2$. For any $t \in [0, 1]$, we consider the distribution that concentrates on a singleton $\{x\}$ and satisfies $p(x, 1) = \frac{1+t}{2}$, $p(x, 2) = \frac{1-t}{2}$. For any $\epsilon > 0$, by the definition of infimum, we can take $h \in \mathcal{H}$ such that $S_h(x, 1) = \tau_\epsilon \in \left[0, \frac{1}{2}\right]$ and satisfies

$$
\sup_{\mu \in [-\tau_\epsilon, 1-\tau_\epsilon]} \left\{ \frac{1+t}{2}[\Phi(\tau_\epsilon) - \Phi(1 - \tau_\epsilon - \mu)] + \frac{1-t}{2}[\Phi(1 - \tau_\epsilon) - \Phi(\tau_\epsilon + \mu)] \right\} < \mathcal{T}^{\mathrm{comp}}(t) + \epsilon.
$$

Then,

$$
\mathcal{R}_{\ell_{0-1}}(h) - \mathcal{R}^*_{\ell_{0-1}}(\mathcal{H}) + \mathcal{M}_{\ell_{0-1}}(\mathcal{H}) = \mathcal{R}_{\ell_{0-1}}(h) - \mathbb{E}_X[\mathcal{C}^*_{\ell_{0-1}}(\mathcal{H}, x)]
$$

$$
= \mathcal{C}_{\ell_{0-1}}(h, x) - \mathcal{C}^*_{\ell_{0-1}}(\mathcal{H}, x)
$$

$$
= t
$$

and

$$
\mathcal{T}^{\mathrm{comp}}(t) \leq \mathcal{R}_{\ell^{\mathrm{comp}}}(h) - \mathcal{R}^*_{\ell^{\mathrm{comp}}}(\mathcal{H}) + \mathcal{M}_{\ell^{\mathrm{comp}}}(\mathcal{H})
$$

$$
= \mathcal{R}_{\ell^{\mathrm{comp}}}(h) - \mathbb{E}_X[\mathcal{C}^*_{\ell^{\mathrm{comp}}}(\mathcal{H}, x)]
$$

$$
= \mathcal{C}_{\ell^{\mathrm{comp}}}(h, x) - \mathcal{C}^*_{\ell^{\mathrm{comp}}}(\mathcal{H}, x)
$$

$$
= \sup_{\mu \in [-\tau_\epsilon, 1-\tau_\epsilon]} \left\{ \frac{1+t}{2}[\Phi(\tau_\epsilon) - \Phi(1 - \tau_\epsilon - \mu)] + \frac{1-t}{2}[\Phi(1 - \tau_\epsilon) - \Phi(\tau_\epsilon + \mu)] \right\}
$$

$$
< \mathcal{T}^{\mathrm{comp}}(t) + \epsilon.
$$

By letting $\epsilon \to 0$, we prove the tightness for $n = 2$. The proof for $n > 2$ directly extends from the case when $n = 2$. Indeed, for any $t \in [0, 1]$, we consider the distribution that concentrates on a singleton $\{x\}$ and satisfies $p(x, 1) = \frac{1+t}{2}$, $p(x, 2) = \frac{1-t}{2}$, $p(x, y) = 0$, $3 \leq y \leq n$. For any $\epsilon > 0$, by the definition

of infimum, we can take $h \in \mathcal{H}$ such that $S_h(x,1) = \tau_{1,\epsilon}$, $S_h(x,2) = \tau_{2,\epsilon}$, $S_h(x,y) = 0$, $3 \le y \le n$ and satisfies $\tau_{1,\epsilon} + \tau_{2,\epsilon} = 1$, and

$$\inf_{P \in \left[\frac{1}{n-1} \vee t, 1\right]} \sup_{\mu \in [-\tau_{2,\epsilon}, \tau_{1,\epsilon}]} \left\{ \frac{P+t}{2} [\Phi(\tau_{2,\epsilon}) - \Phi(\tau_{1,\epsilon} - \mu)] + \frac{P-t}{2} [\Phi(\tau_{1,\epsilon}) - \Phi(\tau_{2,\epsilon} + \mu)] \right\}$$

$$= \sup_{\mu \in [-\tau_{2,\epsilon}, \tau_{1,\epsilon}]} \left\{ \frac{1+t}{2} [\Phi(\tau_{2,\epsilon}) - \Phi(\tau_{1,\epsilon} - \mu)] + \frac{1-t}{2} [\Phi(\tau_{1,\epsilon}) - \Phi(\tau_{2,\epsilon} + \mu)] \right\}$$

$$< \mathcal{T}^{\mathrm{comp}}(t) + \epsilon.$$

Then,

$$\mathcal{R}_{\ell_{0-1}}(h) - \mathcal{R}^*_{\ell_{0-1}}(\mathcal{H}) + \mathcal{M}_{\ell_{0-1}}(\mathcal{H}) = t$$

and

$$\mathcal{T}^{\mathrm{comp}}(t) < \mathcal{T}^{\mathrm{comp}}(t) + \epsilon.$$

By letting $\epsilon \to 0$, we prove the tightness for $n > 2$. $\qquad\square$

## C.2 Characterization of $\mathcal{T}^{\mathrm{comp}}$ (Theorem 3)

**Theorem 3 (characterization of $\mathcal{T}^{\mathrm{comp}}$).** *Assume that $\Phi$ is convex, differentiable at $\frac{1}{2}$ and $\Phi'\left(\frac{1}{2}\right) < 0$. Then, $\mathcal{T}^{\mathrm{comp}}$ can be expressed as follows:*

$$\mathcal{T}^{\mathrm{comp}}(t) = \begin{cases} \Phi\left(\frac{1}{2}\right) - \inf_{\mu \in \left[-\frac{1}{2}, \frac{1}{2}\right]} \left\{ \frac{1-t}{2} \Phi\left(\frac{1}{2} + \mu\right) + \frac{1+t}{2} \Phi\left(\frac{1}{2} - \mu\right) \right\} & n = 2 \\ \inf_{\tau \in \left[\frac{1}{n}, \frac{1}{2}\right]} \left\{ \Phi(\tau) - \inf_{\mu \in [-\tau, \tau]} \left\{ \frac{1+t}{2} \Phi(\tau - \mu) + \frac{1-t}{2} \Phi(\tau + \mu) \right\} \right\} & n > 2. \end{cases}$$

*Proof.* For $n = 2$, we have

$$\mathcal{T}^{\mathrm{comp}}(t) = \inf_{\tau \in [0, \frac{1}{2}]} \sup_{\mu \in [-\tau, 1-\tau]} \left\{ \frac{1+t}{2} [\Phi(\tau) - \Phi(1 - \tau - \mu)] + \frac{1-t}{2} [\Phi(1 - \tau) - \Phi(\tau + \mu)] \right\}$$

$$= \inf_{\tau \in [0, \frac{1}{2}]} \left( \frac{1+t}{2} \Phi(\tau) + \frac{1-t}{2} [\Phi(1 - \tau)] - \inf_{\mu \in [-\tau, 1-\tau]} \left\{ \frac{1+t}{2} \Phi(1 - \tau - \mu) + \frac{1-t}{2} \Phi(\tau + \mu) \right\} \right)$$

$$= \inf_{\tau \in [0, \frac{1}{2}]} \left( \frac{1+t}{2} \Phi(\tau) + \frac{1-t}{2} [\Phi(1 - \tau)] \right) - \inf_{\mu \in \left[-\frac{1}{2}, \frac{1}{2}\right]} \left\{ \frac{1-t}{2} \Phi\left(\frac{1}{2} + \mu\right) + \frac{1+t}{2} \Phi\left(\frac{1}{2} - \mu\right) \right\}$$

$$\ge \inf_{\tau \in [0, \frac{1}{2}]} \left( \Phi\left(\frac{1}{2}\right) + \Phi'\left(\frac{1}{2}\right) t \left(\tau - \frac{1}{2}\right) \right) - \inf_{\mu \in \left[-\frac{1}{2}, \frac{1}{2}\right]} \left\{ \frac{1-t}{2} \Phi\left(\frac{1}{2} + \mu\right) + \frac{1+t}{2} \Phi\left(\frac{1}{2} - \mu\right) \right\}$$

$$\text{($\Phi$ is convex)}$$

$$= \Phi\left(\frac{1}{2}\right) - \inf_{\mu \in \left[-\frac{1}{2}, \frac{1}{2}\right]} \left\{ \frac{1-t}{2} \Phi\left(\frac{1}{2} + \mu\right) + \frac{1+t}{2} \Phi\left(\frac{1}{2} - \mu\right) \right\} \qquad \text{($\Phi'\left(\frac{1}{2}\right) < 0$, $t\left(\tau - \frac{1}{2}\right) \le 0$)}$$

where the equality can be achieved by $\tau = \frac{1}{2}$.

For $n > 2$, we have

$$\mathcal{T}^{\mathrm{comp}}(t) = \inf_{P \in \left[\frac{1}{n-1}, 1\right]} \inf_{\substack{\tau_1 \ge \max(\tau_2, 1/n) \\ \tau_1 + \tau_2 \le 1, \tau_2 \ge 0}} \sup_{\mu \in [-\tau_2, \tau_1]} F(P, \tau_1, \tau_2, \mu)$$

where we let $F(P, \tau_1, \tau_2, \mu) = \frac{P+t}{2} [\Phi(\tau_2) - \Phi(\tau_1 - \mu)] + \frac{P-t}{2} [\Phi(\tau_1) - \Phi(\tau_2 + \mu)]$. For simplicity, we assume that $\Phi$ is differentiable. For general convex $\Phi$, we can proceed by using left and right derivatives, which are non-decreasing. Differentiate $F$ with respect to $\mu$, we have

$$\frac{\partial F}{\partial \mu} = \frac{P+t}{2} \Phi'(\tau_1 - \mu) + \frac{t - P}{2} \Phi'(\tau_2 + \mu).$$

Using the fact that $P \in \left[\frac{1}{n-1} \vee t, 1\right]$, $t \in [0, 1]$ and $\Phi'$ is non-decreasing, we obtain that $\frac{\partial F}{\partial \mu}$ is non-increasing. Furthermore, $\Phi'$ is non-decreasing and non-positive, $\Phi$ is non-negative, we obtain that

$\Phi'(+\infty) = 0$. This implies that $\frac{\partial F}{\partial \mu}(+\infty) \le 0$ and $\frac{\partial F}{\partial \mu}(-\infty) \ge 0$. Therefore, there exists $\mu_0 \in \mathbb{R}$ such that

$$\frac{\partial F}{\partial \mu}(\mu_0) = \frac{P+t}{2}\Phi'(\tau_1 - \mu_0) + \frac{t-P}{2}\Phi'(\tau_2 + \mu_0) = 0$$

By taking $\mu = \tau_1 - \tau_2$ and using the fact that $\tau_2 \le \frac{1}{2}$, $\Phi'\left(\frac{1}{2}\right) < 0$, we have

$$\frac{\partial F}{\partial \mu}(\tau_1 - \tau_2) = \frac{P+t}{2}\Phi'(\tau_2) + \frac{t-P}{2}\Phi'(\tau_1) < 0.$$

Thus, since $\frac{\partial F}{\partial \mu}$ is non-increasing, we obtain $\mu_0 < \tau_1 - \tau_2$. Differentiate $F$ with respect to $\tau_2$ at $\mu_0$, we have

$$\frac{\partial F}{\partial \tau_2} = \frac{P+t}{2}\Phi'(\tau_2) + \frac{t-P}{2}\Phi'(\tau_2 + \mu_0).$$

Since $\Phi'$ is non-decreasing, we obtain

$$\frac{\partial F}{\partial \tau_2} \le \frac{P+t}{2}\Phi'(\tau_2) + \frac{t-P}{2}\Phi'(\tau_2 + \tau_1 - \tau_2) = \frac{\partial F}{\partial \mu}(\tau_1 - \tau_2) < 0,$$

which implies that the infimum $\inf_{\tau_1 \ge \max\{\tau_2, \frac{1}{n}\}}$ is achieved when $\tau_2 = \tau_1$. Differentiate $F$ with respect to $P$ at $\mu_0$ and $\tau_1 = \tau_2$, by the convexity of $\Phi$, we obtain

$$\frac{\partial F}{\partial P} = \Phi(\tau_1) - \Phi(\tau_1 - \mu_0) + \Phi(\tau_1) - \Phi(\tau_1 + \mu_0) \le 0,$$

which implies that the infimum $\inf_{P \in \left[\frac{1}{n-1}, 1\right]}$ is achieved when $P = 1$. Above all, we obtain

$$\mathcal{T}^{\mathrm{comp}}(t) = \inf_{\tau \in \left[\frac{1}{n}, \frac{1}{2}\right]} \sup_{\mu \in [-\tau, \tau]} F(1, \tau, \tau, \mu)$$

$$= \inf_{\tau \in \left[\frac{1}{n}, \frac{1}{2}\right]} \left\{ \Phi(\tau) - \inf_{\mu \in [-\tau, \tau]} \left\{ \frac{1+t}{2}\Phi(\tau - \mu) + \frac{1-t}{2}\Phi(\tau + \mu) \right\} \right\}.$$

$\square$

## C.3 Computation of examples

**Example:** $\Phi(t) = -\log(t)$. For $n = 2$, plugging in $\Phi(t) = -\log(t)$ in Theorem 3, gives

$$\mathcal{T}^{\mathrm{comp}} = \log 2 - \inf_{\mu \in \left[-\frac{1}{2}, \frac{1}{2}\right]} \left\{ -\frac{1-t}{2}\log\left(\frac{1}{2} + \mu\right) - \frac{1+t}{2}\log\left(\frac{1}{2} - \mu\right) \right\}$$

$$= \frac{1+t}{2}\log(1+t) + \frac{1-t}{2}\log(1-t). \qquad \text{(minimum achieved at } \mu = -\frac{t}{2}\text{)}$$

Similarly, for $n > 2$, plugging in $\Phi(t) = -\log(t)$ in Theorem 3 yields

$$\mathcal{T}^{\mathrm{comp}} = \inf_{\tau \in \left[\frac{1}{n}, \frac{1}{2}\right]} \left\{ -\log\tau - \inf_{\mu \in [-\tau, \tau]} \left\{ -\frac{1-t}{2}\log(\tau + \mu) - \frac{1+t}{2}\log(\tau - \mu) \right\} \right\}$$

$$= \frac{1+t}{2}\log(1+t) + \frac{1-t}{2}\log(1-t). \qquad \text{(minimum achieved at } \mu = -\tau t\text{)}$$

**Example:** $\Phi(t) = \frac{1}{t} - 1$. For $n = 2$, plugging in $\Phi(t) = \frac{1}{t} - 1$ in Theorem 3, gives

$$\mathcal{T}^{\mathrm{comp}} = 2 - \inf_{\mu \in \left[-\frac{1}{2}, \frac{1}{2}\right]} \left\{ \frac{1-t}{2}\frac{1}{\frac{1}{2} + \mu} + \frac{1+t}{2}\frac{1}{\frac{1}{2} - \mu} \right\}$$

$$= 1 - \sqrt{1 - t^2}. \qquad \text{(minimum achieved at } \mu = \frac{(1-t)^{\frac{1}{2}} - (1+t)^{\frac{1}{2}}}{2\left((1+t)^{\frac{1}{2}} + (1-t)^{\frac{1}{2}}\right)}\text{)}$$

Similarly, for $n > 2$, plugging in $\Phi(t) = \frac{1}{t} - 1$ in Theorem 3 yields

$$\mathcal{T}^{\mathrm{comp}} = \inf_{\tau \in \left[\frac{1}{n}, \frac{1}{2}\right]} \left\{ \frac{1}{\tau} - \inf_{\mu \in [-\tau, \tau]} \left\{ \frac{1+t}{2}\frac{1}{\tau - \mu} + \frac{1+t}{2}\frac{1}{\tau + \mu} \right\} \right\}$$

$$= \inf_{\tau \in \left[\frac{1}{n}, \frac{1}{2}\right]} \frac{1}{2\tau}\left(1 - \sqrt{1 - t^2}\right) \qquad \text{(minimum achieved at } \mu = \frac{(1-t)^{\frac{1}{2}} - (1+t)^{\frac{1}{2}}}{(1+t)^{\frac{1}{2}} + (1-t)^{\frac{1}{2}}}\tau\text{)}$$

$$= 1 - \sqrt{1 - t^2}. \qquad \text{(minimum achieved at } \tau = \frac{1}{2}\text{)}$$

**Example:** $\Phi(t) = \frac{1}{q}(1 - t^q), q \in (0, 1)$. For $n = 2$, plugging in $\Phi(t) = \frac{1}{q}(1 - t^q)$ in Theorem 3, gives

$$\mathcal{J}^{\text{comp}} = -\frac{1}{q2^q} - \inf_{\mu \in [-\frac{1}{2}, \frac{1}{2}]} \left\{ -\frac{1-t}{2q}\left(\frac{1}{2} + \mu\right)^q - \frac{1+t}{2q}\left(\frac{1}{2} - \mu\right)^q \right\}$$

$$= \frac{1}{q2^q}\left(\frac{(1+t)^{\frac{1}{1-q}} + (1-t)^{\frac{1}{1-q}}}{2}\right)^{1-q} - \frac{1}{q2^q}.$$

$$\left(\text{minimum achieved at } \mu = \frac{(1-t)^{\frac{1}{1-q}} - (1+t)^{\frac{1}{1-q}}}{2\left((1+t)^{\frac{1}{1-q}} + (1-t)^{\frac{1}{1-q}}\right)}\right)$$

Similarly, for $n > 2$, plugging in $\Phi(t) = \frac{1}{q}(1 - t^q)$ in Theorem 3 yields

$$\mathcal{J}^{\text{comp}} = \inf_{\tau \in [\frac{1}{n}, \frac{1}{2}]} \left\{ -\frac{\tau^q}{q} - \inf_{\mu \in [-\tau, \tau]} \left\{ -\frac{1+t}{2q}(\tau - \mu)^q - \frac{1-t}{2q}(\tau + \mu)^q \right\} \right\}$$

$$= \inf_{\tau \in [\frac{1}{n}, \frac{1}{2}]} \frac{\tau^q}{q}\left(\frac{(1+t)^{\frac{1}{1-q}} + (1-t)^{\frac{1}{1-q}}}{2}\right)^{1-q} - \frac{\tau^q}{q}$$

$$\left(\text{minimum achieved at } \mu = \frac{(1-t)^{\frac{1}{1-q}} - (1+t)^{\frac{1}{1-q}}}{(1+t)^{\frac{1}{1-q}} + (1-t)^{\frac{1}{1-q}}}\tau\right)$$

$$= \frac{1}{qn^q}\left(\frac{(1+t)^{\frac{1}{1-q}} + (1-t)^{\frac{1}{1-q}}}{2}\right)^{1-q} - \frac{1}{qn^q}. \qquad \left(\text{minimum achieved at } \tau = \frac{1}{n}\right)$$

**Example:** $\Phi(t) = 1 - t$. For $n = 2$, plugging in $\Phi(t) = 1 - t$ in Theorem 3, gives

$$\mathcal{J}^{\text{comp}} = \frac{1}{2} - \inf_{\mu \in [-\frac{1}{2}, \frac{1}{2}]} \left\{ \frac{1-t}{2}\left(\frac{1}{2} - \mu\right) + \frac{1+t}{2}\left(\frac{1}{2} + \mu\right) \right\} = \frac{1}{2} - \frac{1-t}{2} = \frac{t}{2}.$$

Similarly, for $n > 2$, plugging in $\Phi(t) = 1 - t$ in Theorem 3 yields

$$\mathcal{J}^{\text{comp}} = \inf_{\tau \in [\frac{1}{n}, \frac{1}{2}]} \left\{ (1 - \tau) - \inf_{\mu \in [-\tau, \tau]} \left\{ \frac{1+t}{2}(1 - \tau + \mu) + \frac{1-t}{2}(1 - \tau - \mu) \right\} \right\}$$

$$= \inf_{\tau \in [\frac{1}{n}, \frac{1}{2}]} \tau t \qquad\qquad (\text{minimum achieved at } \mu = -\tau)$$

$$= \frac{t}{n}. \qquad\qquad \left(\text{minimum achieved at } \tau = \frac{1}{n}\right)$$

**Example:** $\Phi(t) = (1 - t)^2$. For $n = 2$, plugging in $\Phi(t) = (1 - t)^2$ in Theorem 3, gives

$$\mathcal{J}^{\text{comp}} = \frac{1}{4} - \inf_{\mu \in [-\frac{1}{2}, \frac{1}{2}]} \left\{ \frac{1-t}{2}\left(\frac{1}{2} - \mu\right)^2 + \frac{1+t}{2}\left(\frac{1}{2} + \mu\right)^2 \right\} = \frac{1}{4} - \frac{1-t^2}{4} = \frac{t^2}{4}.$$

Similarly, for $n > 2$, plugging in $\Phi(t) = (1 - t)^2$ in Theorem 3 yields

$$\mathcal{J}^{\text{comp}} = \inf_{\tau \in [\frac{1}{n}, \frac{1}{2}]} \left\{ (1 - \tau)^2 - \inf_{\mu \in [-\tau, \tau]} \left\{ \frac{1+t}{2}(1 - \tau + \mu)^2 + \frac{1-t}{2}(1 - \tau - \mu)^2 \right\} \right\}$$

$$= \inf_{\tau \in [\frac{1}{n}, \frac{1}{2}]} \left\{ (1 - \tau)^2 t^2 \right\} \qquad\qquad (\text{minimum achieved at } \mu = t(\tau - 1))$$

$$= \frac{t^2}{4}. \qquad\qquad \left(\text{minimum achieved at } \tau = \frac{1}{2}\right)$$

## D   Proofs for constrained losses

Let $y_{\max} = \operatorname{argmax}_{y \in \mathcal{Y}} p(x, y)$ and $\mathsf{h}(x) = \operatorname{argmax}_{y \in \mathcal{Y}} h(x, y)$, where we choose the label with the highest index under the natural ordering of labels as the tie-breaking strategy.

## D.1 Proof of $\mathcal{H}$-consistency bounds with $\mathcal{T}^{\mathrm{cstnd}}$ (Theorem 10)

**Theorem 10** ($\mathcal{H}$-**consistency bound for constrained losses**). *Assume that $\mathcal{H}$ is symmetric and complete. Assume that $\mathcal{T}^{\mathrm{cstnd}}$ is convex. Then, for any hypothesis $h \in \mathcal{H}$ and any distribution,*

$$\mathcal{T}^{\mathrm{cstnd}}\big(\mathcal{R}_{\ell_{0-1}}(h) - \mathcal{R}^*_{\ell_{0-1}}(\mathcal{H}) + \mathcal{M}_{\ell_{0-1}}(\mathcal{H})\big) \le \mathcal{R}_{\ell^{\mathrm{cstnd}}}(h) - \mathcal{R}^*_{\ell^{\mathrm{cstnd}}}(\mathcal{H}) + \mathcal{M}_{\ell^{\mathrm{cstnd}}}(\mathcal{H})$$

*with $\mathcal{H}$-estimation error transformation for constrained losses defined on $t \in [0,1]$ by $\mathcal{T}^{\mathrm{cstnd}}(t) =$*

$$\begin{cases} \inf\limits_{\tau \ge 0} \sup\limits_{\mu \in \mathbb{R}} \big\{ \frac{1-t}{2}[\Phi(\tau) - \Phi(-\tau + \mu)] + \frac{1+t}{2}[\Phi(-\tau) - \Phi(\tau - \mu)] \big\} & n = 2 \\ \inf\limits_{P \in [\frac{1}{n-1}, 1]} \inf\limits_{\tau_1 \ge \max\{\tau_2, 0\}} \sup\limits_{\mu \in \mathbb{R}} \big\{ \frac{2-P-t}{2}[\Phi(-\tau_2) - \Phi(-\tau_1 + \mu)] + \frac{2-P+t}{2}[\Phi(-\tau_1) - \Phi(-\tau_2 - \mu)] \big\} & n > 2. \end{cases}$$

*Furthermore, for any $t \in [0,1]$, there exist a distribution $\mathcal{D}$ and a hypothesis $h \in \mathcal{H}$ such that $\mathcal{R}_{\ell_{0-1}}(h) - \mathcal{R}^*_{\ell_{0-1}}(\mathcal{H}) + \mathcal{M}_{\ell_{0-1}}(\mathcal{H}) = t$ and $\mathcal{R}_{\ell^{\mathrm{cstnd}}}(h) - \mathcal{R}^*_{\ell^{\mathrm{cstnd}}}(\mathcal{H}) + \mathcal{M}_{\ell^{\mathrm{cstnd}}}(\mathcal{H}) = \mathcal{T}^{\mathrm{cstnd}}(t)$.*

*Proof.* For the constrained loss $\ell^{\mathrm{cstnd}}$, the conditional $\ell^{\mathrm{cstnd}}$-risk can be expressed as follows:

$$\begin{aligned} \mathcal{C}_{\ell^{\mathrm{cstnd}}}(h, x) &= \sum_{y \in \mathcal{Y}} p(x, y) \ell^{\mathrm{cstnd}}(h, x, y) \\ &= \sum_{y \in \mathcal{Y}} p(x, y) \sum_{y' \ne y} \Phi(-h(x, y')) \\ &= \sum_{y \in \mathcal{Y}} \Phi(-h(x, y)) \sum_{y' \ne y} p(x, y') \\ &= \sum_{y \in \mathcal{Y}} \Phi(-h(x, y))(1 - p(x, y)) \\ &= \Phi(-h(x, y_{\max}))(1 - p(x, y_{\max})) + \Phi(-h(x, \mathsf{h}(x)))(1 - p(x, \mathsf{h}(x))) \\ &\quad + \sum_{y \notin \{y_{\max}, \mathsf{h}(x)\}} \Phi(-h(x, y))(1 - p(x, y)). \end{aligned}$$

For any $h \in \mathcal{H}$ and $x \in \mathcal{X}$, by the symmetry and completeness of $\mathcal{H}$, we can always find a family of hypotheses $\{h_\mu : \mu \in \mathbb{R}\} \subset \mathcal{H}$ such that $h_\mu(x, \cdot)$ take the following values:

$$h_\mu(x, y) = \begin{cases} h(x, y) & \text{if } y \notin \{y_{\max}, \mathsf{h}(x)\} \\ h(x, y_{\max}) + \mu & \text{if } y = \mathsf{h}(x) \\ h(x, \mathsf{h}(x)) - \mu & \text{if } y = y_{\max}. \end{cases}$$

Note that the hypotheses $h_\mu$ satisfies the constraint:

$$\sum_{y \in \mathcal{Y}} h_\mu(x, y) = \sum_{y \in \mathcal{Y}} h(x, y) = 0, \ \forall \mu \in \mathbb{R}.$$

Let $p_1 = p(x, y_{\max})$, $p_2 = p(x, \mathsf{h}(x))$, $\tau_1 = h(x, \mathsf{h}(x))$ and $\tau_2 = h(x, y_{\max})$ to simplify the notation. Then, by the definition of $h_\mu$, we have for any $h \in \mathcal{H}$ and $x \in \mathcal{X}$,

$$\mathcal{C}_{\ell^{\mathrm{cstnd}}}(h, x) - \inf_{\mu \in \mathbb{R}} \mathcal{C}_{\ell^{\mathrm{cstnd}}}(h_\mu, x)$$

$$= \sup_{\mu \in \mathbb{R}} \Big\{ (1 - p_1)[\Phi(-\tau_2) - \Phi(-\tau_1 + \mu)] + (1 - p_2)[\Phi(-\tau_1) - \Phi(-\tau_2 - \mu)] \Big\}$$

$$= \sup_{\mu \in \mathbb{R}} \Big\{ \frac{2 - P - p_1 + p_2}{2}[\Phi(-\tau_2) - \Phi(-\tau_1 + \mu)] + \frac{2 - P + p_1 - p_2}{2}[\Phi(-\tau_1) - \Phi(-\tau_2 - \mu)] \Big\}$$

$$\hspace{10cm} (P = p_1 + p_2 \in \big[\tfrac{1}{n-1}, 1\big])$$

$$= \inf_{P \in [\frac{1}{n-1}, 1]} \inf_{\tau_1 \ge \max\{\tau_2, 0\}} \sup_{\mu \in \mathbb{R}} \Big\{ \frac{2 - P - p_1 + p_2}{2}[\Phi(-\tau_2) - \Phi(-\tau_1 + \mu)]$$

$$\qquad + \frac{2 - P + p_1 - p_2}{2}[\Phi(-\tau_1) - \Phi(-\tau_2 - \mu)] \Big\} \hspace{3cm} (\tau_1 \ge 0, \ \tau_2 \le \tau_1)$$

$$= \mathcal{T}^{\mathrm{cstnd}}(p_1 - p_2)$$

$$= \mathcal{T}^{\mathrm{cstnd}}(\Delta \mathcal{C}_{\ell_{0-1}, \mathcal{H}}(h, x)). \hspace{8cm} \text{(by Lemma 1)}$$

where for $n = 2$, an additional constraint $\tau_1 + \tau_2 = 0$ is imposed and the expression of $\mathcal{T}^{\text{comp}}$ is simplified. Since $\mathcal{T}^{\text{cstnd}}$ is convex, by Jensen's inequality, we obtain for any hypothesis $h \in \mathcal{H}$ and any distribution,

$$
\begin{aligned}
\mathcal{T}^{\text{cstnd}}&\big(\mathcal{R}_{\ell_{0-1}}(h) - \mathcal{R}^*_{\ell_{0-1}}(\mathcal{H}) + \mathcal{M}_{\ell_{0-1}}(\mathcal{H})\big) \\
&= \mathcal{T}^{\text{cstnd}}\Big(\mathbb{E}_X[\Delta\mathcal{C}_{\ell_{0-1},\mathcal{H}}(h,x)]\Big) \\
&\leq \mathbb{E}_X\Big[\mathcal{T}^{\text{cstnd}}(\Delta\mathcal{C}_{\ell_{0-1},\mathcal{H}}(h,x))\Big] \\
&\leq \mathbb{E}_X\Big[\Delta\mathcal{C}_{\ell^{\text{cstnd}},\mathcal{H}}(h,x)\Big] \\
&= \mathcal{R}_{\ell^{\text{cstnd}}}(h) - \mathcal{R}^*_{\ell^{\text{cstnd}}}(\mathcal{H}) + \mathcal{M}_{\ell^{\text{cstnd}}}(\mathcal{H}).
\end{aligned}
$$

For the second part, we first consider $n = 2$. For any $t \in [0, 1]$, we consider the distribution that concentrates on a singleton $\{x\}$ and satisfies $p(x, 1) = \frac{1+t}{2}$, $p(x, 2) = \frac{1-t}{2}$. For any $\epsilon > 0$, by the definition of infimum, we can take $h \in \mathcal{H}$ such that $h(x, 2) = \tau_\epsilon \geq 0$ and satisfies

$$
\sup_{\mu \in \mathbb{R}}\left\{\frac{1-t}{2}[\Phi(\tau_\epsilon) - \Phi(-\tau_\epsilon + \mu)] + \frac{1+t}{2}[\Phi(-\tau_\epsilon) - \Phi(\tau_\epsilon - \mu)]\right\} < \mathcal{T}^{\text{cstnd}}(t) + \epsilon.
$$

Then,

$$
\begin{aligned}
\mathcal{R}_{\ell_{0-1}}(h) - \mathcal{R}^*_{\ell_{0-1}}(\mathcal{H}) + \mathcal{M}_{\ell_{0-1}}(\mathcal{H}) &= \mathcal{R}_{\ell_{0-1}}(h) - \mathbb{E}_X\big[\mathcal{C}^*_{\ell_{0-1}}(\mathcal{H},x)\big] \\
&= \mathcal{C}_{\ell_{0-1}}(h,x) - \mathcal{C}^*_{\ell_{0-1}}(\mathcal{H},x) \\
&= t
\end{aligned}
$$

and

$$
\begin{aligned}
\mathcal{T}^{\text{cstnd}}(t) &\leq \mathcal{R}_{\ell^{\text{cstnd}}}(h) - \mathcal{R}^*_{\ell^{\text{cstnd}}}(\mathcal{H}) + \mathcal{M}_{\ell^{\text{cstnd}}}(\mathcal{H}) \\
&= \mathcal{R}_{\ell^{\text{cstnd}}}(h) - \mathbb{E}_X\big[\mathcal{C}^*_{\ell^{\text{cstnd}}}(\mathcal{H},x)\big] \\
&= \mathcal{C}_{\ell^{\text{cstnd}}}(h,x) - \mathcal{C}^*_{\ell^{\text{cstnd}}}(\mathcal{H},x) \\
&= \sup_{\mu \in \mathbb{R}}\left\{\frac{1-t}{2}[\Phi(\tau_\epsilon) - \Phi(-\tau_\epsilon + \mu)] + \frac{1+t}{2}[\Phi(-\tau_\epsilon) - \Phi(\tau_\epsilon - \mu)]\right\} \\
&< \mathcal{T}^{\text{cstnd}}(t) + \epsilon.
\end{aligned}
$$

By letting $\epsilon \to 0$, we conclude the proof. The proof for $n > 2$ directly extends from the case when $n = 2$. Indeed, For any $t \in [0, 1]$, we consider the distribution that concentrates on a singleton $\{x\}$ and satisfies $p(x, 1) = \frac{1+t}{2}$, $p(x, 2) = \frac{1-t}{2}$, $p(x, y) = 0$, $3 \leq y \leq n$. For any $\epsilon > 0$, by the definition of infimum, we can take $h \in \mathcal{H}$ such that $h(x, 1) = \tau_{1,\epsilon}$, $h(x, 2) = \tau_{2,\epsilon}$, $h(x, y) = 0$, $3 \leq y \leq n$ and satisfies $\tau_{1,\epsilon} + \tau_{2,\epsilon} = 0$, and

$$
\begin{aligned}
\inf_{P \in \left[\frac{1}{n-1}, 1\right]} &\sup_{\mu \in \mathbb{R}}\left\{\frac{2 - P - t}{2}[\Phi(-\tau_{2,\epsilon}) - \Phi(-\tau_{1,\epsilon} + \mu)] + \frac{2 - P + t}{2}[\Phi(-\tau_{1,\epsilon}) - \Phi(-\tau_{2,\epsilon} - \mu)]\right\} \\
&= \sup_{\mu \in \mathbb{R}}\left\{\frac{1-t}{2}[\Phi(-\tau_{2,\epsilon}) - \Phi(-\tau_{1,\epsilon} + \mu)] + \frac{1+t}{2}[\Phi(-\tau_{1,\epsilon}) - \Phi(-\tau_{2,\epsilon} - \mu)]\right\} \\
&< \mathcal{T}^{\text{cstnd}}(t) + \epsilon.
\end{aligned}
$$

Then,

$$
\mathcal{R}_{\ell_{0-1}}(h) - \mathcal{R}^*_{\ell_{0-1}}(\mathcal{H}) + \mathcal{M}_{\ell_{0-1}}(\mathcal{H}) = t
$$

and

$$
\mathcal{T}^{\text{cstnd}}(t) \leq \mathcal{R}_{\ell^{\text{cstnd}}}(h) - \mathcal{R}^*_{\ell^{\text{cstnd}}}(\mathcal{H}) + \mathcal{M}_{\ell^{\text{cstnd}}}(\mathcal{H}) < \mathcal{T}^{\text{cstnd}}(t) + \epsilon.
$$

$\square$

## D.2 Characterization of $\mathcal{T}^{\mathrm{cstnd}}$ (Theorem 11)

**Theorem 11** (**characterization of** $\mathcal{T}^{\mathrm{cstnd}}$). *Assume that $\Phi$ is convex, differentiable at zero and $\Phi'(0) < 0$. Then, $\mathcal{T}^{\mathrm{cstnd}}$ can be expressed as follows:*

$$
\mathcal{T}^{\mathrm{cstnd}}(t) = \begin{cases} \Phi(0) - \inf_{\mu\in\mathbb{R}}\left\{\frac{1-t}{2}\Phi(\mu) + \frac{1+t}{2}\Phi(-\mu)\right\} & n = 2 \\ \inf_{\tau\ge 0}\left\{\left(2 - \frac{1}{n-1}\right)\Phi(-\tau) - \inf_{\mu\in\mathbb{R}}\left\{\frac{2-t-\frac{1}{n-1}}{2}\Phi(-\tau+\mu) + \frac{2+t-\frac{1}{n-1}}{2}\Phi(-\tau-\mu)\right\}\right\} & n > 2 \end{cases}
$$

$$
\ge \begin{cases} \Phi(0) - \inf_{\mu\in\mathbb{R}}\left\{\frac{1-t}{2}\Phi(\mu) + \frac{1+t}{2}\Phi(-\mu)\right\} & n = 2 \\ \inf_{\tau\ge 0}\left\{2\Phi(-\tau) - \inf_{\mu\in\mathbb{R}}\left\{\frac{2-t}{2}\Phi(-\tau+\mu) + \frac{2+t}{2}\Phi(-\tau-\mu)\right\}\right\} & n > 2. \end{cases}
$$

*Proof.* For $n = 2$, we have

$$
\mathcal{T}^{\mathrm{cstnd}}(t) = \inf_{\tau\ge 0}\sup_{\mu\in\mathbb{R}}\left\{\frac{1-t}{2}[\Phi(\tau) - \Phi(-\tau+\mu)] + \frac{1+t}{2}[\Phi(-\tau) - \Phi(\tau-\mu)]\right\}
$$

$$
= \inf_{\tau\ge 0}\left(\frac{1-t}{2}\Phi(\tau) + \frac{1+t}{2}[\Phi(-\tau)]\right) - \inf_{\mu\in\mathbb{R}}\left\{\frac{1-t}{2}\Phi(-\tau+\mu) + \frac{1+t}{2}\Phi(\tau-\mu)\right\}
$$

$$
= \inf_{\tau\ge 0}\left(\frac{1-t}{2}\Phi(\tau) + \frac{1+t}{2}[\Phi(-\tau)]\right) - \inf_{\mu\in\mathbb{R}}\left\{\frac{1-t}{2}\Phi(\mu) + \frac{1+t}{2}\Phi(-\mu)\right\}
$$

$$
\ge \inf_{\tau\ge 0}(\Phi(0) - \Phi'(0)t\tau) - \inf_{\mu\in\mathbb{R}}\left\{\frac{1-t}{2}\Phi(\mu) + \frac{1+t}{2}\Phi(-\mu)\right\} \qquad (\Phi \text{ is convex})
$$

$$
= \Phi(0) - \inf_{\mu\in\mathbb{R}}\left\{\frac{1-t}{2}\Phi(\mu) + \frac{1+t}{2}\Phi(-\mu)\right\} \qquad (\Phi'(0) < 0,\ t\tau \ge 0)
$$

where the equality can be achieved by $\tau = 0$.

For $n > 2$, we have

$$
\mathcal{T}^{\mathrm{cstnd}}(t) = \inf_{P\in\left[\frac{1}{n-1}, 1\right]}\inf_{\tau_1 \ge \max\{\tau_2, 0\}}\sup_{\mu\in\mathbb{R}} F(P, \tau_1, \tau_2, \mu)
$$

where we let $F(P, \tau_1, \tau_2, \mu) = \frac{2-P-t}{2}[\Phi(-\tau_2) - \Phi(-\tau_1+\mu)] + \frac{2-P+t}{2}[\Phi(-\tau_1) - \Phi(-\tau_2-\mu)]$. For simplicity, we assume that $\Phi$ is differentiable. For general convex $\Phi$, we can proceed by using left and right derivatives, which are non-decreasing. Differentiate $F$ with respect to $\mu$, we have

$$
\frac{\partial F}{\partial \mu} = \frac{P+t-2}{2}\Phi'(-\tau_1+\mu) + \frac{2-P+t}{2}\Phi'(-\tau_2-\mu).
$$

Using the fact that $P \in \left[\frac{1}{n-1}, 1\right], t \in [0, 1]$ and $\Phi'$ is non-decreasing, we obtain that $\frac{\partial F}{\partial \mu}$ is non-increasing. Furthermore, $\Phi'$ is non-decreasing and non-positive, $\Phi$ is non-negative, we obtain that $\Phi'(+\infty) = 0$. This implies that $\frac{\partial F}{\partial \mu}(+\infty) \le 0$ and $\frac{\partial F}{\partial \mu}(-\infty) \ge 0$. Therefore, there exists $\mu_0 \in \mathbb{R}$ such that

$$
\frac{\partial F}{\partial \mu}(\mu_0) = \frac{P+t-2}{2}\Phi'(-\tau_1+\mu_0) + \frac{2-P+t}{2}\Phi'(-\tau_2-\mu_0) = 0
$$

By taking $\mu = \tau_1 - \tau_2$ and using the fact that $\Phi'(0) < 0$, we have

$$
\frac{\partial F}{\partial \mu}(\tau_1 - \tau_2) = \frac{P+t-2}{2}\Phi'(-\tau_2) + \frac{2-P+t}{2}\Phi'(-\tau_1) < 0.
$$

Thus, since $\frac{\partial F}{\partial \mu}$ is non-increasing, we obtain $\mu_0 < \tau_1 - \tau_2$. Differentiate $F$ with respect to $\tau_2$ at $\mu_0$, we have

$$
\frac{\partial F}{\partial \tau_2} = \frac{P+t-2}{2}\Phi'(-\tau_2) + \frac{2-P+t}{2}\Phi'(-\tau_2-\mu_0).
$$

Since $\Phi'$ is non-decreasing, we obtain

$$
\frac{\partial F}{\partial \tau_2} \le \frac{P+t-2}{2}\Phi'(-\tau_2) + \frac{2-P+t}{2}\Phi'(-\tau_2-\tau_1+\tau_2) = \frac{\partial F}{\partial \mu}(\tau_1 - \tau_2) < 0,
$$

which implies that the infimum $\inf_{\tau_1 \ge \max\{\tau_2, 0\}}$ is achieved when $\tau_2 = \tau_1$. Differentiate $F$ with respect to $P$ at $\mu_0$ and $\tau_1 = \tau_2$, by the convexity of $\Phi$, we obtain

$$
\frac{\partial F}{\partial P} = \Phi(-\tau_1+\mu_0) - \Phi(-\tau_1) - \Phi(-\tau_1) + \Phi(-\tau_1-\mu_0) \ge 0,
$$

which implies that the infimum $\inf_{P\in[\frac{1}{n-1},1]}$ is achieved when $P = \frac{1}{n-1}$. Above all, we obtain

$$\mathcal{T}^{\mathrm{cstnd}}(t) = \inf_{\tau\geq 0}\sup_{\mu\in\mathbb{R}} F\left(\frac{1}{n-1},\tau,\tau,\mu\right)$$

$$= \inf_{\tau\geq 0}\left\{\left(2 - \frac{1}{n-1}\right)\Phi(-\tau) - \inf_{\mu\in\mathbb{R}}\left\{\frac{2-t-\frac{1}{n-1}}{2}\Phi(-\tau+\mu) + \frac{2+t-\frac{1}{n-1}}{2}\Phi(-\tau-\mu)\right\}\right\}$$

$$\geq \inf_{\tau\geq 0}\sup_{\mu\in\mathbb{R}} F(0,\tau,\tau,\mu)$$

$$= \inf_{\tau\geq 0}\left\{2\Phi(-\tau) - \inf_{\mu\in\mathbb{R}}\left\{\frac{2-t}{2}\Phi(-\tau+\mu) + \frac{2+t}{2}\Phi(-\tau-\mu)\right\}\right\}.$$

$\square$

### D.3 Computation of examples

**Example:** $\Phi(t) = \Phi_{\exp}(t) = e^{-t}$. For $n = 2$, plugging in $\Phi(t) = e^{-t}$ in Theorem 11, gives

$$\mathcal{T}^{\mathrm{comp}} = 1 - \inf_{\mu\in\mathbb{R}}\left\{\frac{1-t}{2}e^{-\mu} + \frac{1+t}{2}e^{\mu}\right\}$$

$$= 1 - \sqrt{1-t^2}. \qquad\text{(minimum achieved at } \mu = \tfrac{1}{2}\log\tfrac{1-t}{1+t})$$

For $n > 2$, plugging in $\Phi(t) = e^{-t}$ in Theorem 11 yields

$$\mathcal{T}^{\mathrm{comp}} \geq \inf_{\tau\geq 0}\left\{2e^{\tau} - \inf_{\mu\in\mathbb{R}}\left\{\frac{2-t}{2}e^{\tau-\mu} + \frac{2+t}{2}e^{\tau+\mu}\right\}\right\}$$

$$\geq 2 - \inf_{\mu\in\mathbb{R}}\left\{\frac{2-t}{2}e^{-\mu} + \frac{2+t}{2}e^{\mu}\right\} \qquad\text{(minimum achieved at } \tau = 0)$$

$$= 2 - \sqrt{4-t^2}. \qquad\text{(minimum achieved at } \mu = \tfrac{1}{2}\log\tfrac{2-t}{2+t})$$

**Example:** $\Phi(t) = \Phi_{\mathrm{hinge}}(t) = \max\{0, 1-t\}$. For $n = 2$, plugging in $\Phi(t) = \max\{0, 1-t\}$ in Theorem 11, gives

$$\mathcal{T}^{\mathrm{comp}} = 1 - \inf_{\mu\in\mathbb{R}}\left\{\frac{1-t}{2}\max\{0, 1-\mu\} + \frac{1+t}{2}\max\{0, 1+\mu\}\right\}$$

$$= t. \qquad\text{(minimum achieved at } \mu = -1)$$

For $n > 2$, plugging in $\Phi(t) = \max\{0, 1-t\}$ in Theorem 11 yields

$$\mathcal{T}^{\mathrm{comp}} \geq \inf_{\tau\geq 0}\left\{2\max\{0, 1+\tau\} - \inf_{\mu\in\mathbb{R}}\left\{\frac{2-t}{2}\max\{0, 1+\tau-\mu\} + \frac{2+t}{2}\max\{0, 1+\tau+\mu\}\right\}\right\}$$

$$= 2 - \inf_{\mu\in\mathbb{R}}\left\{\frac{2-t}{2}\max\{0, 1-\mu\} + \frac{2+t}{2}\max\{0, 1+\mu\}\right\} \qquad\text{(minimum achieved at } \tau = 0)$$

$$= t. \qquad\text{(minimum achieved at } \mu = -1)$$

**Example:** $\Phi(t) = \Phi_{\mathrm{sq-hinge}}(t) = (1-t)^2\mathbb{1}_{t\leq 1}$. For $n = 2$, plugging in $\Phi(t) = (1-t)^2\mathbb{1}_{t\leq 1}$ in Theorem 11, gives

$$\mathcal{T}^{\mathrm{comp}} = 1 - \inf_{\mu\in\mathbb{R}}\left\{\frac{1-t}{2}(1-\mu)^2\mathbb{1}_{\mu\leq 1} + \frac{1+t}{2}(1+\mu)^2\mathbb{1}_{\mu\geq -1}\right\}$$

$$= t^2. \qquad\text{(minimum achieved at } \mu = -t)$$

For $n > 2$, plugging in $\Phi(t) = (1-t)^2\mathbb{1}_{t\leq 1}$ in Theorem 11 yields

$$\mathcal{T}^{\mathrm{comp}} \geq \inf_{\tau\geq 0}\left\{2(1+\tau)^2\mathbb{1}_{\tau\geq -1} - \inf_{\mu\in\mathbb{R}}\left\{\frac{2-t}{2}(1+\tau-\mu)^2\mathbb{1}_{-\tau+\mu\leq 1} + \frac{2+t}{2}(1+\tau+\mu)^2\mathbb{1}_{\tau+\mu\geq -1}\right\}\right\}$$

$$\geq 2 - \inf_{\mu\in\mathbb{R}}\left\{\frac{2-t}{2}(1-\mu)^2\mathbb{1}_{\mu\leq 1} + \frac{2+t}{2}(1+\mu)^2\mathbb{1}_{\mu\geq -1}\right\} \qquad\text{(minimum achieved at } \tau = 0)$$

$$= \frac{t^2}{2}. \qquad\text{(minimum achieved at } \mu = -\tfrac{t}{2})$$

**Example:** $\Phi(t) = \Phi_{\text{sq}}(t) = (1-t)^2$. For $n = 2$, plugging in $\Phi(t) = (1-t)^2$ in Theorem 11, gives

$$\mathcal{T}^{\text{comp}} = 1 - \inf_{\mu \in \mathbb{R}} \left\{ \frac{1-t}{2}(1-\mu)^2 + \frac{1+t}{2}(1+\mu)^2 \right\}$$

$$= t^2. \qquad \text{(minimum achieved at } \mu = -t)$$

For $n > 2$, plugging in $\Phi(t) = (1-t)^2$ in Theorem 11 yields

$$\mathcal{T}^{\text{comp}} \geq \inf_{\tau \geq 0} \left\{ 2(1+\tau)^2 - \inf_{\mu \in \mathbb{R}} \left\{ \frac{2-t}{2}(1+\tau-\mu)^2 + \frac{2+t}{2}(1+\tau+\mu)^2 \right\} \right\}$$

$$\geq 2 - \inf_{\mu \in \mathbb{R}} \left\{ \frac{2-t}{2}(1-\mu)^2 + \frac{2+t}{2}(1+\mu)^2 \right\} \qquad \text{(minimum achieved at } \tau = 0)$$

$$= \frac{t^2}{2}. \qquad \text{(minimum achieved at } \mu = -\frac{t}{2})$$

# E    Extensions of comp-sum losses

## E.1    Proof of $\overline{\mathcal{H}}$-consistency bounds with $\overline{\mathcal{T}}^{\text{comp}}$ (Theorem 5)

**Theorem 5** ($\overline{\mathcal{H}}$-**consistency bound for comp-sum losses**). *Assume that* $\overline{\mathcal{T}}^{\text{comp}}$ *is convex. Then, the following inequality holds for any hypothesis* $h \in \overline{\mathcal{H}}$ *and any distribution:*

$$\overline{\mathcal{T}}^{\text{comp}}\left( \mathcal{R}_{\ell_{0-1}}(h) - \mathcal{R}^*_{\ell_{0-1}}(\overline{\mathcal{H}}) + \mathcal{M}_{\ell_{0-1}}(\overline{\mathcal{H}}) \right) \leq \mathcal{R}_{\ell^{\text{comp}}}(h) - \mathcal{R}^*_{\ell^{\text{comp}}}(\overline{\mathcal{H}}) + \mathcal{M}_{\ell^{\text{comp}}}(\overline{\mathcal{H}})$$

*with* $\overline{\mathcal{T}}^{\text{comp}}$ *the* $\overline{\mathcal{H}}$*-estimation error transformation for comp-sum losses defined for all* $t \in [0,1]$ *by* $\overline{\mathcal{T}}^{\text{comp}}(t) =$

$$\begin{cases} \inf_{\tau \in [0, \frac{1}{2}]} \sup_{\mu \in [s_{\min} - \tau, 1 - \tau - s_{\min}]} \left\{ \frac{1+t}{2}[\Phi(\tau) - \Phi(1-\tau-\mu)] + \frac{1-t}{2}[\Phi(1-\tau) - \Phi(\tau+\mu)] \right\} & n = 2 \\ \inf_{P \in [\frac{1}{n-1} \vee t, 1]} \inf_{\substack{S_{\min} \leq \tau_2 \leq \tau_1 \leq S_{\max} \\ \tau_1 + \tau_2 \leq 1}} \sup_{\mu \in C} \left\{ \frac{P+t}{2}[\Phi(\tau_2) - \Phi(\tau_1 - \mu)] + \frac{P-t}{2}[\Phi(\tau_1) - \Phi(\tau_2 + \mu)] \right\} & n > 2, \end{cases}$$

*where* $C = \left[ \max\{s_{\min} - \tau_2, \tau_1 - s_{\max}\}, \min\{s_{\max} - \tau_2, \tau_1 - s_{\min}\} \right]$, $s_{\max} = \frac{1}{1+(n-1)e^{-2\inf_x \Lambda(x)}}$ *and* $s_{\min} = \frac{1}{1+(n-1)e^{2\inf_x \Lambda(x)}}$. *Furthermore, for any* $t \in [0,1]$, *there exist a distribution* $\mathcal{D}$ *and* $h \in \mathcal{H}$ *such that* $\mathcal{R}_{\ell_{0-1}}(h) - \mathcal{R}^*_{\ell_{0-1}}(\mathcal{H}) + \mathcal{M}_{\ell_{0-1}}(\mathcal{H}) = t$ *and* $\mathcal{R}_{\ell^{\text{comp}}}(h) - \mathcal{R}^*_{\ell^{\text{comp}}}(\mathcal{H}) + \mathcal{M}_{\ell^{\text{comp}}}(\mathcal{H}) = \mathcal{T}^{\text{comp}}(t)$.

*Proof.* For the comp-sum loss $\ell^{\text{comp}}$, the conditional $\ell^{\text{comp}}$-risk can be expressed as follows:

$$\mathcal{C}_{\ell^{\text{comp}}}(h, x)$$
$$= \sum_{y \in \mathcal{Y}} p(x, y) \ell^{\text{comp}}(h, x, y)$$
$$= \sum_{y \in \mathcal{Y}} p(x, y) \Phi\left( \frac{e^{h(x,y)}}{\sum_{y' \in \mathcal{Y}} e^{h(x,y')}} \right)$$
$$= \sum_{y \in \mathcal{Y}} p(x, y) \Phi(S_h(x, y))$$
$$= p(x, y_{\max}) \Phi(S_h(x, y_{\max})) + p(x, \mathsf{h}(x)) \Phi(S_h(x, \mathsf{h}(x))) + \sum_{y \notin \{y_{\max}, \mathsf{h}(x)\}} p(x, y) \Phi(S_h(x, y))$$

where we let $S_h(x, y) = \frac{e^{h(x,y)}}{\sum_{y' \in \mathcal{Y}} e^{h(x,y')}}$ for any $y \in \mathcal{Y}$ with the constraint that $\sum_{y \in \mathcal{Y}} S_h(x, y) = 1$. Note that for any $h \in \mathcal{H}$,

$$\frac{1}{1+(n-1)e^{2\Lambda(x)}} = \frac{e^{-\Lambda(x)}}{e^{-\Lambda(x)} + (n-1)e^{\Lambda(x)}} \leq S_h(x, y) \leq \frac{e^{\Lambda(x)}}{e^{\Lambda(x)} + (n-1)e^{-\Lambda(x)}} = \frac{1}{1+(n-1)e^{-2\Lambda(x)}}$$

Therefore for any $(x, y) \in \mathcal{X} \times \mathcal{Y}$, $S_h(x, y) \in [S_{\min}, S_{\max}]$, where we let $S_{\max} = \frac{1}{1+(n-1)e^{-2\Lambda(x)}}$ and $S_{\min} = \frac{1}{1+(n-1)e^{2\Lambda(x)}}$. Furthermore, all values in $[S_{\min}, S_{\max}]$ of $S_h$ can be reached for some $h \in \mathcal{H}$.

Observe that $0 \le S_{\max} + S_{\min} \le 1$. Let $y_{\max} = \mathrm{argmax}_{y \in \mathcal{Y}} \, p(x, y)$, where we choose the label with the highest index under the natural ordering of labels as the tie-breaking strategy. For any $h \in \mathcal{H}$ such that $\mathsf{h}(x) \ne y_{\max}$ and $x \in \mathcal{X}$, we can always find a family of hypotheses $\{h_\mu\} \subset \mathcal{H}$ such that $S_{h,\mu}(x, \cdot) = \frac{e^{h_\mu(x,\cdot)}}{\sum_{y' \in \mathcal{Y}} e^{h_\mu(x,y')}}$ take the following values:

$$S_{h,\mu}(x, y) = \begin{cases} S_h(x, y) & \text{if } y \notin \{y_{\max}, \mathsf{h}(x)\} \\ S_h(x, y_{\max}) + \mu & \text{if } y = \mathsf{h}(x) \\ S_h(x, \mathsf{h}(x)) - \mu & \text{if } y = y_{\max}. \end{cases}$$

Note that $S_{h,\mu}$ satisfies the constraint:

$$\sum_{y \in \mathcal{Y}} S_{h,\mu}(x, y) = \sum_{y \in \mathcal{Y}} S_h(x, y) = 1.$$

Since $S_{h,\mu}(x, y) \in [S_{\min}, S_{\max}]$, we have the following constraints on $\mu$:

$$\begin{aligned} S_{\min} - S_h(x, y_{\max}) \le \mu \le S_{\max} - S_h(x, y_{\max}) \\ S_h(x, \mathsf{h}(x)) - S_{\max} \le \mu \le S_h(x, \mathsf{h}(x)) - S_{\min}. \end{aligned} \tag{7}$$

Let $p_1 = p(x, y_{\max})$, $p_2 = p(x, \mathsf{h}(x))$, $\tau_1 = S_h(x, \mathsf{h}(x))$ and $\tau_2 = S_h(x, y_{\max})$ to simplify the notation. Let $\overline{C} = \{\mu \in \mathbb{R} : \mu \text{ verify constraint (7)}\}$. Since $S_h(x, \mathsf{h}(x)) - S_{\max} \le S_{\max} - S_h(x, y_{\max})$ and $S_{\min} - S_h(x, y_{\max}) \le S_h(x, \mathsf{h}(x)) - S_{\min}$, $\overline{C}$ is not an empty set and can be expressed as $\overline{C} = [\max\{S_{\min} - \tau_2, \tau_1 - S_{\max}\}, \min\{S_{\max} - \tau_2, \tau_1 - S_{\min}\}]$.

Then, by the definition of $S_{h,\mu}$, we have for any $h \in \mathcal{H}$ and $x \in \mathcal{X}$,

$$\mathcal{C}_{\ell^{\mathrm{comp}}}(h, x) - \inf_{\mu \in \overline{C}} \mathcal{C}_{\ell^{\mathrm{comp}}}(h_\mu, x)$$

$$= \sup_{\mu \in \overline{C}} \left\{ p_1[\Phi(\tau_2) - \Phi(\tau_1 - \mu)] + p_2[\Phi(\tau_1) - \Phi(\tau_2 + \mu)] \right\}$$

$$= \sup_{\mu \in \overline{C}} \left\{ \frac{P + p_1 - p_2}{2}[\Phi(\tau_2) - \Phi(\tau_1 - \mu)] + \frac{P - p_1 + p_2}{2}[\Phi(\tau_1) - \Phi(\tau_2 + \mu)] \right\}$$

$$\hspace{6cm} (P = p_1 + p_2 \in \left[\tfrac{1}{n-1} \vee p_1 - p_2, 1\right])$$

$$\ge \inf_{P \in \left[\frac{1}{n-1} \vee p_1 - p_2, 1\right]} \inf_{\substack{S_{\min} \le \tau_2 \le \tau_1 \le S_{\max} \\ \tau_1 + \tau_2 \le 1}} \sup_{\mu \in \overline{C}} \left\{ \frac{P + p_1 - p_2}{2}[\Phi(\tau_2) - \Phi(\tau_1 - \mu)] \right.$$

$$\left. + \frac{P - p_1 + p_2}{2}[\Phi(\tau_1) - \Phi(\tau_2 + \mu)] \right\} \hspace{1.5cm} (S_{\min} \le \tau_2 \le \tau_1 \le S_{\max}, \tau_1 + \tau_2 \le 1)$$

$$\ge \inf_{P \in \left[\frac{1}{n-1} \vee p_1 - p_2, 1\right]} \inf_{\substack{S_{\min} \le \tau_2 \le \tau_1 \le S_{\max} \\ \tau_1 + \tau_2 \le 1}} \sup_{\mu \in C} \left\{ \frac{P + p_1 - p_2}{2}[\Phi(\tau_2) - \Phi(\tau_1 - \mu)] \right.$$

$$\left. + \frac{P - p_1 + p_2}{2}[\Phi(\tau_1) - \Phi(\tau_2 + \mu)] \right\} \hspace{1.5cm} (S_{\min} \le s_{\min} \le s_{\max} \le S_{\max})$$

$$= \mathcal{T}^{\mathrm{comp}}(p_1 - p_2)$$

$$= \mathcal{T}^{\mathrm{comp}}(\Delta \mathcal{C}_{\ell_{0-1}, \mathcal{H}}(h, x)), \hspace{4cm} \text{(by Lemma 1)}$$

where $C = [\max\{s_{\min} - \tau_2, \tau_1 - s_{\max}\}, \min\{s_{\max} - \tau_2, \tau_1 - s_{\min}\}] \subset \overline{C}$, $s_{\max} = \frac{1}{1 + (n-1)e^{-2 \inf_x \Lambda(x)}}$ and $s_{\min} = \frac{1}{1 + (n-1)e^{2 \inf_x \Lambda(x)}}$. Note that for $n = 2$, an additional constraint $\tau_1 + \tau_2 = 1$ is imposed and the expression can be simplified as

$$\mathcal{C}_{\ell^{\mathrm{comp}}}(h, x) - \inf_{\mu \in \overline{C}} \mathcal{C}_{\ell^{\mathrm{comp}}}(h_\mu, x)$$

$$\ge \inf_{\tau \in [0, \frac{1}{2}]} \sup_{\mu \in [s_{\min} - \tau, 1 - \tau - s_{\min}]} \left\{ \frac{1 + p_1 - p_2}{2}[\Phi(\tau) - \Phi(1 - \tau - \mu)] + \frac{1 - p_1 + p_2}{2}[\Phi(1 - \tau) - \Phi(\tau + \mu)] \right\}$$

$$= \mathcal{T}^{\mathrm{comp}}(p_1 - p_2)$$

$$= \mathcal{T}^{\mathrm{comp}}(\Delta \mathcal{C}_{\ell_{0-1}, \mathcal{H}}(h, x)), \hspace{4cm} \text{(by Lemma 1)}$$

where we use the fact that $s_{\max} + s_{\min} = 1$ and $P = 1$ when $n = 2$. Since $\mathcal{T}^{\mathrm{comp}}$ is convex, by Jensen's inequality, we obtain for any hypothesis $h \in \mathcal{H}$ and any distribution,

$$\mathcal{T}^{\mathrm{comp}}\big(\mathcal{R}_{\ell_{0-1}}(h) - \mathcal{R}^*_{\ell_{0-1}}(\mathcal{H}) + \mathcal{M}_{\ell_{0-1}}(\mathcal{H})\big)$$

$$= \mathcal{T}^{\mathrm{comp}}\Big(\mathbb{E}_X[\Delta\mathcal{C}_{\ell_{0-1},\mathcal{H}}(h,x)]\Big)$$

$$\leq \mathbb{E}_X[\mathcal{T}^{\mathrm{comp}}(\Delta\mathcal{C}_{\ell_{0-1},\mathcal{H}}(h,x))]$$

$$\leq \mathbb{E}_X[\Delta\mathcal{C}_{\ell^{\mathrm{comp}},\mathcal{H}}(h,x)]$$

$$= \mathcal{R}_{\ell^{\mathrm{comp}}}(h) - \mathcal{R}^*_{\ell^{\mathrm{comp}}}(\mathcal{H}) + \mathcal{M}_{\ell^{\mathrm{comp}}}(\mathcal{H}).$$

For the second part, we first consider $n = 2$. For any $t \in [0,1]$, we consider the distribution that concentrates on a singleton $\{x\}$ and satisfies $p(x,1) = \frac{1+t}{2}$, $p(x,2) = \frac{1-t}{2}$. For any $\epsilon > 0$, by the definition of infimum, we can take $h \in \mathcal{H}$ such that $S_h(x,1) = \tau_\epsilon \in [0, \frac{1}{2}]$ and satisfies

$$\sup_{\mu \in [s_{\min} - \tau_\epsilon, 1 - \tau_\epsilon - s_{\min}]} \left\{ \frac{1+t}{2}[\Phi(\tau_\epsilon) - \Phi(1 - \tau_\epsilon - \mu)] + \frac{1-t}{2}[\Phi(1 - \tau_\epsilon) - \Phi(\tau_\epsilon + \mu)] \right\} < \mathcal{T}^{\mathrm{comp}}(t) + \epsilon.$$

Then,

$$\mathcal{R}_{\ell_{0-1}}(h) - \mathcal{R}^*_{\ell_{0-1}}(\mathcal{H}) + \mathcal{M}_{\ell_{0-1}}(\mathcal{H}) = \mathcal{R}_{\ell_{0-1}}(h) - \mathbb{E}_X[\mathcal{C}^*_{\ell_{0-1}}(\mathcal{H},x)]$$

$$= \mathcal{C}_{\ell_{0-1}}(h,x) - \mathcal{C}^*_{\ell_{0-1}}(\mathcal{H},x)$$

$$= t$$

and

$$\mathcal{T}^{\mathrm{comp}}(t) \leq \mathcal{R}_{\ell^{\mathrm{comp}}}(h) - \mathcal{R}^*_{\ell^{\mathrm{comp}}}(\mathcal{H}) + \mathcal{M}_{\ell^{\mathrm{comp}}}(\mathcal{H})$$

$$= \mathcal{R}_{\ell^{\mathrm{comp}}}(h) - \mathbb{E}_X[\mathcal{C}^*_{\ell^{\mathrm{comp}}}(\mathcal{H},x)]$$

$$= \mathcal{C}_{\ell^{\mathrm{comp}}}(h,x) - \mathcal{C}^*_{\ell^{\mathrm{comp}}}(\mathcal{H},x)$$

$$= \sup_{\mu \in [s_{\min} - \tau_\epsilon, 1 - \tau_\epsilon - s_{\min}]} \left\{ \frac{1+t}{2}[\Phi(\tau_\epsilon) - \Phi(1 - \tau_\epsilon - \mu)] + \frac{1-t}{2}[\Phi(1 - \tau_\epsilon) - \Phi(\tau_\epsilon + \mu)] \right\}$$

$$< \mathcal{T}^{\mathrm{comp}}(t) + \epsilon.$$

By letting $\epsilon \to 0$, we conclude the proof. The proof for $n > 2$ directly extends from the case when $n = 2$. Indeed, For any $t \in [0,1]$, we consider the distribution that concentrates on a singleton $\{x\}$ and satisfies $p(x,1) = \frac{1+t}{2}$, $p(x,2) = \frac{1-t}{2}$, $p(x,y) = 0$, $3 \leq y \leq n$. For any $\epsilon > 0$, by the definition of infimum, we can take $h \in \mathcal{H}$ such that $S_h(x,1) = \tau_{1,\epsilon}$, $S_h(x,2) = \tau_{2,\epsilon}$ and $S_h(x,y) = 0$, $3 \leq y \leq n$ and satisfies $\tau_{1,\epsilon} + \tau_{2,\epsilon} = 1$, and

$$\inf_{P \in [\frac{1}{n-1} \vee t, 1]} \sup_{\mu \in C} \left\{ \frac{P+t}{2}[\Phi(\tau_{2,\epsilon}) - \Phi(\tau_{1,\epsilon} - \mu)] + \frac{P-t}{2}[\Phi(\tau_{1,\epsilon}) - \Phi(\tau_{2,\epsilon} + \mu)] \right\}$$

$$= \sup_{\mu \in C} \left\{ \frac{1+t}{2}[\Phi(\tau_{2,\epsilon}) - \Phi(\tau_{1,\epsilon} - \mu)] + \frac{1-t}{2}[\Phi(\tau_{1,\epsilon}) - \Phi(\tau_{2,\epsilon} + \mu)] \right\}$$

$$< \mathcal{T}^{\mathrm{comp}}(t) + \epsilon.$$

Then,

$$\mathcal{R}_{\ell_{0-1}}(h) - \mathcal{R}^*_{\ell_{0-1}}(\mathcal{H}) + \mathcal{M}_{\ell_{0-1}}(\mathcal{H}) = t$$

and

$$\mathcal{T}^{\mathrm{comp}}(t) \leq \mathcal{R}_{\ell^{\mathrm{comp}}}(h) - \mathcal{R}^*_{\ell^{\mathrm{comp}}}(\mathcal{H}) + \mathcal{M}_{\ell^{\mathrm{comp}}}(\mathcal{H}) < \mathcal{T}^{\mathrm{comp}}(t) + \epsilon.$$

By letting $\epsilon \to 0$, we conclude the proof. $\qquad\square$

## E.2  Logistic loss

**Theorem 6 ($\overline{\mathcal{H}}$-consistency bounds for logistic loss).** *For any $h \in \overline{\mathcal{H}}$ and any distribution, we have*

$$\mathcal{R}_{\ell_{0-1}}(h) - \mathcal{R}^*_{\ell_{0-1}}(\overline{\mathcal{H}}) + \mathcal{M}_{\ell_{0-1}}(\overline{\mathcal{H}}) \leq \Psi^{-1}\Big(\mathcal{R}_{\ell_{\log}}(h) - \mathcal{R}^*_{\ell_{\log}}(\overline{\mathcal{H}}) + \mathcal{M}_{\ell_{\log}}(\overline{\mathcal{H}})\Big),$$

*where $\ell_{\log} = -\log\Big(\frac{e^{h(x,y)}}{\sum_{y' \in \mathcal{Y}} e^{h(x,y')}}\Big)$ and $\Psi(t) = \begin{cases} \frac{1+t}{2}\log(1+t) + \frac{1-t}{2}\log(1-t) & t \leq \frac{s_{\max} - s_{\min}}{s_{\min} + s_{\max}} \\ \frac{t}{2}\log\Big(\frac{s_{\max}}{s_{\min}}\Big) + \log\Big(\frac{2\sqrt{s_{\max}s_{\min}}}{s_{\max} + s_{\min}}\Big) & \text{otherwise.} \end{cases}$*

*Proof.* For the multinomial logistic loss $\ell_{\log}$, plugging in $\Phi(t) = -\log(t)$ in Theorem 5, gives $\overline{\mathcal{T}}^{\mathrm{comp}}$

$$\geq \inf_{P \in [\frac{1}{n-1} \vee t, 1]} \inf_{\substack{S_{\min} \leq \tau_2 \leq \tau_1 \leq S_{\max} \\ \tau_1 + \tau_2 \leq 1}} \sup_{\mu \in C} \left\{ \frac{P+t}{2}[-\log(\tau_2) + \log(\tau_1 - \mu)] + \frac{P-t}{2}[-\log(\tau_1) + \log(\tau_2 + \mu)] \right\}$$

where $C = [\max\{s_{\min} - \tau_2, \tau_1 - s_{\max}\}, \min\{s_{\max} - \tau_2, \tau_1 - s_{\min}\}]$. Here, we only compute the expression for $n > 2$. The expression for $n = 2$ will lead to the same result since it can be viewed as a special case of the expression for $n > 2$. By differentiating with respect to $\tau_2$ and $P$, we can see that the infimum is achieved when $\tau_1 = \tau_2 = \frac{s_{\min} + s_{\max}}{2}$ and $P = 1$ modulo some elementary analysis. Thus, $\overline{\mathcal{T}}^{\mathrm{comp}}$ can be reformulated as

$$\overline{\mathcal{T}}^{\mathrm{comp}} = \sup_{\mu \in C} \left\{ \frac{1+t}{2}\left[-\log\left(\frac{s_{\min} + s_{\max}}{2}\right) + \log\left(\frac{s_{\min} + s_{\max}}{2} - \mu\right)\right] \right.$$
$$\left. + \frac{1-t}{2}\left[-\log\left(\frac{s_{\min} + s_{\max}}{2}\right) + \log\left(\frac{s_{\min} + s_{\max}}{2} + \mu\right)\right] \right\}$$
$$= -\log\left(\frac{s_{\min} + s_{\max}}{2}\right) + \sup_{\mu \in C} g(\mu)$$

where $C = \left[\frac{s_{\min} - s_{\max}}{2}, \frac{s_{\max} - s_{\min}}{2}\right]$ and $g(\mu) = \frac{1+t}{2}\log\left(\frac{s_{\min} + s_{\max}}{2} - \mu\right) + \frac{1-t}{2}\log\left(\frac{s_{\min} + s_{\max}}{2} + \mu\right)$. Since $g$ is continuous, it attains its supremum over a compact set. Note that $g$ is concave and differentiable. In view of that, the maximum over the open set $(-\infty, +\infty)$ can be obtained by setting its gradient to zero. Differentiate $g(\mu)$ to optimize, we obtain

$$g(\mu^*) = 0, \quad \mu^* = -\frac{t(s_{\min} + s_{\max})}{2}.$$

Moreover, by the concavity, $g(\mu)$ is non-increasing when $\mu \geq \mu^*$. Since $s_{\max} - s_{\min} \geq 0$, we have

$$\mu^* \leq 0 \leq \frac{s_{\max} - s_{\min}}{2}$$

In view of the constraint $C$, if $\mu^* \geq \frac{s_{\min} - s_{\max}}{2}$, the maximum is achieved by $\mu = \mu^*$. Otherwise, if $\mu^* < \frac{s_{\min} - s_{\max}}{2}$, since $g(\mu)$ is non-increasing when $\mu \geq \mu^*$, the maximum is achieved by $\mu = \frac{s_{\min} - s_{\max}}{2}$. Since $\mu^* \geq \frac{s_{\min} - s_{\max}}{2}$ is equivalent to $t \leq \frac{s_{\max} - s_{\min}}{s_{\min} + s_{\max}}$, the maximum can be expressed as

$$\max_{\mu \in C} g(\mu) = \begin{cases} g(\mu^*) & t \leq \frac{s_{\max} - s_{\min}}{s_{\min} + s_{\max}} \\ g\left(\frac{s_{\min} - s_{\max}}{2}\right) & \text{otherwise} \end{cases}$$

Computing the value of $g$ at these points yields:

$$g(\mu^*) = \frac{1+t}{2}\log\frac{(1+t)(s_{\min} + s_{\max})}{2} + \frac{1-t}{2}\log\frac{(1-t)(s_{\min} + s_{\max})}{2}$$
$$g\left(\frac{s_{\min} - s_{\max}}{2}\right) = \frac{1+t}{2}\log(s_{\max}) + \frac{1-t}{2}\log(s_{\min})$$

Then, if $t \leq \frac{s_{\max} - s_{\min}}{s_{\min} + s_{\max}}$, we obtain

$$\overline{\mathcal{T}}^{\mathrm{comp}} = -\log\left(\frac{s_{\min} + s_{\max}}{2}\right) + \frac{1+t}{2}\log\frac{(1+t)(s_{\min} + s_{\max})}{2} + \frac{1-t}{2}\log\frac{(1-t)(s_{\min} + s_{\max})}{2}$$
$$= \frac{1+t}{2}\log(1+t) + \frac{1-t}{2}\log(1-t).$$

Otherwise, we obtain

$$\overline{\mathcal{T}}^{\mathrm{comp}} = -\log\left(\frac{s_{\min} + s_{\max}}{2}\right) + \frac{1+t}{2}\log(s_{\max}) + \frac{1-t}{2}\log(s_{\min})$$
$$= \frac{t}{2}\log\left(\frac{s_{\max}}{s_{\min}}\right) + \log\left(\frac{2\sqrt{s_{\max}s_{\min}}}{s_{\max} + s_{\min}}\right).$$

Since $\overline{\mathcal{T}}^{\mathrm{comp}}$ is convex, by Theorem 5, for any $h \in \overline{\mathcal{H}}$ and any distribution,

$$\mathcal{R}_{\ell_{0-1}}(h) - \mathcal{R}^*_{\ell_{0-1}}(\overline{\mathcal{H}}) + \mathcal{M}_{\ell_{0-1}}(\overline{\mathcal{H}}) \leq \Psi^{-1}\left(\mathcal{R}_{\ell_{\log}}(h) - \mathcal{R}^*_{\ell_{\log}}(\overline{\mathcal{H}}) + \mathcal{M}_{\ell_{\log}}(\overline{\mathcal{H}})\right),$$

where

$$\Psi(t) = \begin{cases} \frac{1+t}{2}\log(1+t) + \frac{1-t}{2}\log(1-t) & t \leq \frac{s_{\max} - s_{\min}}{s_{\min} + s_{\max}} \\ \frac{t}{2}\log\left(\frac{s_{\max}}{s_{\min}}\right) + \log\left(\frac{2\sqrt{s_{\max}s_{\min}}}{s_{\max} + s_{\min}}\right) & \text{otherwise.} \end{cases}$$

$\square$

### E.3 Sum exponential loss

**Theorem 9 ($\overline{\mathcal{H}}$-consistency bounds for sum exponential loss).** *For any $h \in \mathcal{H}$ and any distribution,*

$$\mathcal{R}_{\ell_{0-1}}(h) - \mathcal{R}^*_{\ell_{0-1}}(\mathcal{H}) + \mathcal{M}_{\ell_{0-1}}(\mathcal{H}) \le \Psi^{-1}\left(\mathcal{R}_{\ell_{\exp}}(h) - \mathcal{R}^*_{\ell_{\exp}}(\mathcal{H}) + \mathcal{M}_{\ell_{\exp}}(\mathcal{H})\right)$$

*where $\ell_{\exp} = \sum_{y' \ne y} e^{h(x,y') - h(x,y)}$ and $\Psi(t) = \begin{cases} 1 - \sqrt{1 - t^2} & t \le \frac{s_{\max}^2 - s_{\min}^2}{s_{\min}^2 + s_{\max}^2} \\ \frac{s_{\max} - s_{\min}}{2 s_{\max} s_{\min}} t - \frac{(s_{\max} - s_{\min})^2}{2 s_{\max} s_{\min} (s_{\max} + s_{\min})} & \text{otherwise.} \end{cases}$*

*Proof.* For the sum exponential loss $\ell_{\exp}$, plugging in $\Phi(t) = \frac{1}{t} - 1$ in Theorem 5, gives $\overline{\mathcal{T}}^{\text{comp}}$

$$\ge \inf_{P \in \left[\frac{1}{n-1} \vee t, 1\right]} \inf_{\substack{S_{\min} \le \tau_2 \le \tau_1 \le S_{\max} \\ \tau_1 + \tau_2 \le 1}} \sup_{\mu \in C} \left\{ \frac{P+t}{2}\left[\frac{1}{\tau_2} - \frac{1}{\tau_1 - \mu}\right] + \frac{P-t}{2}\left[\frac{1}{\tau_1} - \frac{1}{\tau_2 + \mu}\right] \right\}$$

where $C = \left[\max\{s_{\min} - \tau_2, \tau_1 - s_{\max}\}, \min\{s_{\max} - \tau_2, \tau_1 - s_{\min}\}\right]$. Here, we only compute the expression for $n > 2$. The expression for $n = 2$ will lead to the same result since it can be viewed as a special case of the expression for $n > 2$. By differentiating with respect to $\tau_2$ and $P$, we can see that the infimum is achieved when $\tau_1 = \tau_2 = \frac{s_{\min} + s_{\max}}{2}$ and $P = 1$ modulo some elementary analysis. Thus, $\overline{\mathcal{T}}^{\text{comp}}$ can be reformulated as

$$\overline{\mathcal{T}}^{\text{comp}} = \sup_{\mu \in C} \left\{ \frac{1+t}{2}\left[\frac{2}{s_{\min} + s_{\max}} - \frac{2}{s_{\min} + s_{\max} - 2\mu}\right] \right.$$
$$\left. + \frac{1-t}{2}\left[\frac{2}{s_{\min} + s_{\max}} - \frac{2}{s_{\min} + s_{\max} + 2\mu}\right] \right\}$$
$$= \frac{2}{s_{\min} + s_{\max}} + \sup_{\mu \in C} g(\mu)$$

where $C = \left[\frac{s_{\min} - s_{\max}}{2}, \frac{s_{\max} - s_{\min}}{2}\right]$ and $g(\mu) = -\frac{1+t}{s_{\min} + s_{\max} - 2\mu} - \frac{1-t}{s_{\min} + s_{\max} + 2\mu}$. Since $g$ is continuous, it attains its supremum over a compact set. Note that $g$ is concave and differentiable. In view of that, the maximum over the open set $(-\infty, +\infty)$ can be obtained by setting its gradient to zero. Differentiate $g(\mu)$ to optimize, we obtain

$$g(\mu^*) = 0, \quad \mu^* = \frac{s_{\min} + s_{\max}}{2} \frac{\sqrt{1-t} - \sqrt{1+t}}{\sqrt{1+t} + \sqrt{1-t}}$$

Moreover, by the concavity, $g(\mu)$ is non-increasing when $\mu \ge \mu^*$. Since $s_{\max} - s_{\min} \ge 0$, we have

$$\mu^* \le 0 \le \frac{s_{\max} - s_{\min}}{2}$$

In view of the constraint $C$, if $\mu^* \ge \frac{s_{\min} - s_{\max}}{2}$, the maximum is achieved by $\mu = \mu^*$. Otherwise, if $\mu^* < \frac{s_{\min} - s_{\max}}{2}$, since $g(\mu)$ is non-increasing when $\mu \ge \mu^*$, the maximum is achieved by $\mu = \frac{s_{\min} - s_{\max}}{2}$. Since $\mu^* \ge \frac{s_{\min} - s_{\max}}{2}$ is equivalent to $t \le \frac{s_{\max}^2 - s_{\min}^2}{s_{\min}^2 + s_{\max}^2}$, the maximum can be expressed as

$$\max_{\mu \in C} g(\mu) = \begin{cases} g(\mu^*) & t \le \frac{s_{\max}^2 - s_{\min}^2}{s_{\min}^2 + s_{\max}^2} \\ g\left(\frac{s_{\min} - s_{\max}}{2}\right) & \text{otherwise} \end{cases}$$

Computing the value of $g$ at these points yields:

$$g(\mu^*) = 1 - \sqrt{1 - t^2} - \frac{2}{s_{\min} + s_{\max}}$$
$$g\left(\frac{s_{\min} - s_{\max}}{2}\right) = -\frac{1+t}{2 s_{\max}} - \frac{1-t}{2 s_{\min}}$$

Then, if $t \le \frac{s_{\max}^2 - s_{\min}^2}{s_{\min}^2 + s_{\max}^2}$, we obtain

$$\overline{\mathcal{T}}^{\text{comp}} = \frac{2}{s_{\min} + s_{\max}} + 1 - \sqrt{1 - t^2} - \frac{2}{s_{\min} + s_{\max}}$$
$$= 1 - \sqrt{1 - t^2}.$$

Otherwise, we obtain

$$\overline{\mathfrak{T}}^{\text{comp}} = \frac{2}{s_{\min} + s_{\max}} - \frac{1+t}{2s_{\max}} - \frac{1-t}{2s_{\min}}$$

$$= \frac{s_{\max} - s_{\min}}{2s_{\max}s_{\min}}t - \frac{(s_{\max} - s_{\min})^2}{2s_{\max}s_{\min}(s_{\max} + s_{\min})}.$$

Since $\overline{\mathfrak{T}}^{\text{comp}}$ is convex, by Theorem 5, for any $h \in \overline{\mathcal{H}}$ and any distribution,

$$\mathcal{R}_{\ell_{0-1}}(h) - \mathcal{R}^*_{\ell_{0-1}}(\overline{\mathcal{H}}) + \mathcal{M}_{\ell_{0-1}}(\overline{\mathcal{H}}) \le \Psi^{-1}\big(\mathcal{R}_{\ell_{\exp}}(h) - \mathcal{R}^*_{\ell_{\exp}}(\overline{\mathcal{H}}) + \mathcal{M}_{\ell_{\exp}}(\overline{\mathcal{H}})\big),$$

where

$$\Psi(t) = \begin{cases} 1 - \sqrt{1 - t^2} & t \le \frac{s_{\max}^2 - s_{\min}^2}{s_{\min}^2 + s_{\max}^2} \\ \frac{s_{\max} - s_{\min}}{2s_{\max}s_{\min}}t - \frac{(s_{\max} - s_{\min})^2}{2s_{\max}s_{\min}(s_{\max} + s_{\min})} & \text{otherwise.} \end{cases}$$

$\square$

### E.4 Generalized cross-entropy loss

**Theorem 16** ($\overline{\mathcal{H}}$-**consistency bounds for generalized cross-entropy loss**). *For any $h \in \overline{\mathcal{H}}$ and any distribution, we have*

$$\mathcal{R}_{\ell_{0-1}}(h) - \mathcal{R}^*_{\ell_{0-1}}(\overline{\mathcal{H}}) + \mathcal{M}_{\ell_{0-1}}(\overline{\mathcal{H}}) \le \Psi^{-1}\big(\mathcal{R}_{\ell_{\text{gce}}}(h) - \mathcal{R}^*_{\ell_{\text{gce}}}(\overline{\mathcal{H}}) + \mathcal{M}_{\ell_{\text{gce}}}(\overline{\mathcal{H}})\big),$$

*where* $\Psi(t) = \begin{cases} \frac{1}{q}\left(\frac{s_{\min} + s_{\max}}{2}\right)^q\left[\left(\frac{(1+t)^{\frac{1}{1-q}} + (1-t)^{\frac{1}{1-q}}}{2}\right)^{1-q} - 1\right] & t \le \frac{s_{\max}^{1-q} - s_{\min}^{1-q}}{s_{\min}^{1-q} + s_{\max}^{1-q}} \\ \frac{t}{2q}\big(s_{\max}^q - s_{\min}^q\big) + \frac{1}{q}\left(\frac{s_{\min}^q + s_{\max}^q}{2} - \left(\frac{s_{\min} + s_{\max}}{2}\right)^q\right) & \text{otherwise.} \end{cases}$ *and* $\ell_{\text{gce}} = \frac{1}{q}\left[1 - \left(\frac{e^{h(x,y)}}{\sum_{y' \in \mathcal{Y}} e^{h(x,y')}}\right)^q\right].$

*Proof.* For generalized cross-entropy loss $\ell_{\text{gce}}$, plugging $\Phi(t) = \frac{1}{q}(1 - t^q)$ in Theorem 5, gives $\overline{\mathfrak{T}}^{\text{comp}}$

$$\ge \inf_{P \in [\frac{1}{n-1} \vee t, 1]} \inf_{\substack{S_{\min} \le \tau_2 \le \tau_1 \le S_{\max} \\ \tau_1 + \tau_2 \le 1}} \sup_{\mu \in C} \left\{ \frac{P+t}{2}\left[-\frac{1}{q}(\tau_2)^q + \frac{1}{q}(\tau_1 - \mu)^q\right] + \frac{P-t}{2}\left[-\frac{1}{q}(\tau_1)^q + \frac{1}{q}(\tau_2 + \mu)^q\right] \right\}$$

where $C = [\max\{s_{\min} - \tau_2, \tau_1 - s_{\max}\}, \min\{s_{\max} - \tau_2, \tau_1 - s_{\min}\}]$. Here, we only compute the expression for $n > 2$. The expression for $n = 2$ will lead to the same result since it can be viewed as a special case of the expression for $n > 2$. By differentiating with respect to $\tau_2$ and $P$, we can see that the infimum is achieved when $\tau_1 = \tau_2 = \frac{s_{\min} + s_{\max}}{2}$ and $P = 1$ modulo some elementary analysis. Thus, $\overline{\mathfrak{T}}^{\text{comp}}$ can be reformulated as

$$\overline{\mathfrak{T}}^{\text{comp}} = \sup_{\mu \in C} \left\{ \frac{1+t}{2q}\left[-\left(\frac{s_{\min} + s_{\max}}{2}\right)^q + \left(\frac{s_{\min} + s_{\max}}{2} - \mu\right)^q\right] \right.$$

$$\left. + \frac{1-t}{2q}\left[-\left(\frac{s_{\min} + s_{\max}}{2}\right)^q + \left(\frac{s_{\min} + s_{\max}}{2} + \mu\right)^q\right] \right\}$$

$$= -\frac{1}{q}\left(\frac{s_{\min} + s_{\max}}{2}\right)^q + \sup_{\mu \in C} g(\mu)$$

where $C = \left[\frac{s_{\min} - s_{\max}}{2}, \frac{s_{\max} - s_{\min}}{2}\right]$ and $g(\mu) = \frac{1+t}{2q}\left(\frac{s_{\min} + s_{\max}}{2} - \mu\right)^q + \frac{1-t}{2q}\left(\frac{s_{\min} + s_{\max}}{2} + \mu\right)^q$. Since $g$ is continuous, it attains its supremum over a compact set. Note that $g$ is concave and differentiable. In view of that, the maximum over the open set $(-\infty, +\infty)$ can be obtained by setting its gradient to zero. Differentiate $g(\mu)$ to optimize, we obtain

$$g(\mu^*) = 0, \quad \mu^* = \frac{(1-t)^{\frac{1}{1-q}} - (1+t)^{\frac{1}{1-q}}}{(1+t)^{\frac{1}{1-q}} + (1-t)^{\frac{1}{1-q}}} \frac{s_{\min} + s_{\max}}{2}.$$

Moreover, by the concavity, $g(\mu)$ is non-increasing when $\mu \geq \mu^*$. Since $s_{\max} - s_{\min} \geq 0$, we have

$$\mu^* \leq 0 \leq \frac{s_{\max} - s_{\min}}{2}$$

In view of the constraint $C$, if $\mu^* \geq \frac{s_{\min} - s_{\max}}{2}$, the maximum is achieved by $\mu = \mu^*$. Otherwise, if $\mu^* < \frac{s_{\min} - s_{\max}}{2}$, since $g(\mu)$ is non-increasing when $\mu \geq \mu^*$, the maximum is achieved by $\mu = \frac{s_{\min} - s_{\max}}{2}$. Since $\mu^* \geq \frac{s_{\min} - s_{\max}}{2}$ is equivalent to $t \leq \frac{s_{\max}^{1-q} - s_{\min}^{1-q}}{s_{\min}^{1-q} + s_{\max}^{1-q}}$, the maximum can be expressed as

$$\max_{\mu \in C} g(\mu) = \begin{cases} g(\mu^*) & t \leq \frac{s_{\max}^{1-q} - s_{\min}^{1-q}}{s_{\min}^{1-q} + s_{\max}^{1-q}} \\ g\left(\frac{s_{\min} - s_{\max}}{2}\right) & \text{otherwise} \end{cases}$$

Computing the value of $g$ at these points yields:

$$g(\mu^*) = \frac{1}{q}\left(\frac{s_{\min} + s_{\max}}{2}\right)^q \left(\frac{(1+t)^{\frac{1}{1-q}} + (1-t)^{\frac{1}{1-q}}}{2}\right)^{1-q}$$

$$g\left(\frac{s_{\min} - s_{\max}}{2}\right) = \frac{1+t}{2q}(s_{\max})^q + \frac{1-t}{2q}(s_{\min})^q$$

Then, if $t \leq \frac{s_{\max}^{1-q} - s_{\min}^{1-q}}{s_{\min}^{1-q} + s_{\max}^{1-q}}$, we obtain

$$\overline{\mathcal{T}}^{\text{comp}} = \frac{1}{q}\left(\frac{s_{\min} + s_{\max}}{2}\right)^q \left(\frac{(1+t)^{\frac{1}{1-q}} + (1-t)^{\frac{1}{1-q}}}{2}\right)^{1-q} - \frac{1}{q}\left(\frac{s_{\min} + s_{\max}}{2}\right)^q$$

$$= \frac{1}{q}\left(\frac{s_{\min} + s_{\max}}{2}\right)^q \left[\left(\frac{(1+t)^{\frac{1}{1-q}} + (1-t)^{\frac{1}{1-q}}}{2}\right)^{1-q} - 1\right]$$

Otherwise, we obtain

$$\overline{\mathcal{T}}^{\text{comp}} = -\frac{1}{q}\left(\frac{s_{\min} + s_{\max}}{2}\right)^q + \frac{1+t}{2q}(s_{\max})^q + \frac{1-t}{2q}(s_{\min})^q$$

$$= \frac{t}{2q}\left(s_{\max}^q - s_{\min}^q\right) + \frac{1}{q}\left(\frac{s_{\min}^q + s_{\max}^q}{2} - \left(\frac{s_{\min} + s_{\max}}{2}\right)^q\right)$$

Since $\overline{\mathcal{T}}^{\text{comp}}$ is convex, by Theorem 5, for any $h \in \overline{\mathcal{H}}$ and any distribution,

$$\mathcal{R}_{\ell_{0-1}}(h) - \mathcal{R}_{\ell_{0-1}}^*\left(\overline{\mathcal{H}}\right) + \mathcal{M}_{\ell_{0-1}}\left(\overline{\mathcal{H}}\right) \leq \Psi^{-1}\left(\mathcal{R}_{\ell_{\text{gce}}}(h) - \mathcal{R}_{\ell_{\text{gce}}}^*\left(\overline{\mathcal{H}}\right) + \mathcal{M}_{\ell_{\text{gce}}}\left(\overline{\mathcal{H}}\right)\right),$$

where

$$\Psi(t) = \begin{cases} \frac{1}{q}\left(\frac{s_{\min} + s_{\max}}{2}\right)^q \left[\left(\frac{(1+t)^{\frac{1}{1-q}} + (1-t)^{\frac{1}{1-q}}}{2}\right)^{1-q} - 1\right] & t \leq \frac{s_{\max}^{1-q} - s_{\min}^{1-q}}{s_{\min}^{1-q} + s_{\max}^{1-q}} \\ \frac{t}{2q}\left(s_{\max}^q - s_{\min}^q\right) + \frac{1}{q}\left(\frac{s_{\min}^q + s_{\max}^q}{2} - \left(\frac{s_{\min} + s_{\max}}{2}\right)^q\right) & \text{otherwise.} \end{cases}$$

$\square$

### E.5 Mean absolute error loss

**Theorem 17** ($\overline{\mathcal{H}}$-consistency bounds for mean absolute error loss). *For any $h \in \overline{\mathcal{H}}$ and any distribution, we have*

$$\mathcal{R}_{\ell_{0-1}}(h) - \mathcal{R}_{\ell_{0-1}}^*\left(\overline{\mathcal{H}}\right) + \mathcal{M}_{\ell_{0-1}}\left(\overline{\mathcal{H}}\right) \leq \frac{2\left(\mathcal{R}_{\ell_{\text{mae}}}(h) - \mathcal{R}_{\ell_{\text{mae}}}^*\left(\overline{\mathcal{H}}\right) + \mathcal{M}_{\ell_{\text{mae}}}\left(\overline{\mathcal{H}}\right)\right)}{s_{\max} - s_{\min}}.$$

*Proof.* For mean absolute error loss $\ell_{\text{mae}}$, plugging $\Phi(t) = 1 - t$ in Theorem 5, gives $\overline{\mathcal{T}}^{\text{comp}}$

$$\geq \inf_{P \in \left[\frac{1}{n-1} \vee t, 1\right]} \inf_{\substack{S_{\min} \leq \tau_2 \leq \tau_1 \leq S_{\max} \\ \tau_1 + \tau_2 \leq 1}} \sup_{\mu \in C} \left\{\frac{P+t}{2}\left[-(\tau_2) + (\tau_1 - \mu)\right] + \frac{P-t}{2}\left[-(\tau_1) + (\tau_2 + \mu)\right]\right\}$$

where $C = [\max\{s_{\min} - \tau_2, \tau_1 - s_{\max}\}, \min\{s_{\max} - \tau_2, \tau_1 - s_{\min}\}]$. Here, we only compute the expression for $n > 2$. The expression for $n = 2$ will lead to the same result since it can be viewed as a special case of the expression for $n > 2$. By differentiating with respect to $\tau_2$ and $P$, we can see that the infimum is achieved when $\tau_1 = \tau_2 = \frac{s_{\min} + s_{\max}}{2}$ and $P = 1$ modulo some elementary analysis. Thus, $\overline{\mathcal{T}}^{\mathrm{comp}}$ can be reformulated as

$$\overline{\mathcal{T}}^{\mathrm{comp}} = \sup_{\mu \in C} \left\{ \frac{1+t}{2}\left[-\left(\frac{s_{\min} + s_{\max}}{2}\right) + \left(\frac{s_{\min} + s_{\max}}{2} - \mu\right)\right] \right.$$
$$\left. + \frac{1-t}{2}\left[-\left(\frac{s_{\min} + s_{\max}}{2}\right) + \left(\frac{s_{\min} + s_{\max}}{2} + \mu\right)\right]\right\}$$
$$= \sup_{\mu \in C} -t\mu$$

where $C = \left[\frac{s_{\min} - s_{\max}}{2}, \frac{s_{\max} - s_{\min}}{2}\right]$. Since $-t\mu$ is monotonically non-increasing, the maximum over $C$ can be achieved by

$$\mu^* = \frac{s_{\min} - s_{\max}}{2}, \quad \overline{\mathcal{T}}^{\mathrm{comp}} = \frac{s_{\max} - s_{\min}}{2}\, t.$$

Since $\overline{\mathcal{T}}^{\mathrm{comp}}$ is convex, by Theorem 5, for any $h \in \overline{\mathcal{H}}$ and any distribution,

$$\mathcal{R}_{\ell_{0-1}}(h) - \mathcal{R}^*_{\ell_{0-1}}(\overline{\mathcal{H}}) + \mathcal{M}_{\ell_{0-1}}(\overline{\mathcal{H}}) \leq \frac{2\big(\mathcal{R}_{\ell_{\mathrm{mae}}}(h) - \mathcal{R}^*_{\ell_{\mathrm{mae}}}(\overline{\mathcal{H}}) + \mathcal{M}_{\ell_{\mathrm{mae}}}(\overline{\mathcal{H}})\big)}{s_{\max} - s_{\min}}.$$

$\square$

# F  Extensions of constrained losses

## F.1  Proof of $\overline{\mathcal{H}}$-consistency bound with $\overline{\mathcal{T}}^{\mathrm{cstnd}}$ (Theorem 12)

**Theorem 12** ($\overline{\mathcal{H}}$-consistency bound for constrained losses). *Assume that* $\overline{\mathcal{T}}^{\mathrm{cstnd}}$ *is convex. Then, the following inequality holds for any hypothesis* $h \in \overline{\mathcal{H}}$ *and any distribution:*

$$\overline{\mathcal{T}}^{\mathrm{cstnd}}\big(\mathcal{R}_{\ell_{0-1}}(h) - \mathcal{R}^*_{\ell_{0-1}}(\overline{\mathcal{H}}) + \mathcal{M}_{\ell_{0-1}}(\overline{\mathcal{H}})\big) \leq \mathcal{R}_{\ell_{\mathrm{cstnd}}}(h) - \mathcal{R}^*_{\ell_{\mathrm{cstnd}}}(\overline{\mathcal{H}}) + \mathcal{M}_{\ell_{\mathrm{cstnd}}}(\overline{\mathcal{H}}). \quad (6)$$

*with* $\overline{\mathcal{T}}^{\mathrm{cstnd}}$ *the* $\overline{\mathcal{H}}$*-estimation error transformation for constrained losses defined for all* $t \in [0,1]$ *by* $\overline{\mathcal{T}}^{\mathrm{cstnd}}(t) =$

$$\begin{cases} \inf\limits_{\tau \geq 0} \sup\limits_{\mu \in [\tau - \Lambda_{\min}, \tau + \Lambda_{\min}]} \left\{\frac{1-t}{2}[\Phi(\tau) - \Phi(-\tau + \mu)] + \frac{1+t}{2}[\Phi(-\tau) - \Phi(\tau - \mu)]\right\} & n = 2 \\ \inf\limits_{P \in [\frac{1}{n-1}, 1]} \inf\limits_{\tau_1 \geq \max\{\tau_2, 0\}} \sup\limits_{\mu \in C} \left\{\frac{2 - P - t}{2}[\Phi(-\tau_2) - \Phi(-\tau_1 + \mu)] + \frac{2 - P + t}{2}[\Phi(-\tau_1) - \Phi(-\tau_2 - \mu)]\right\} & n > 2, \end{cases}$$

*where* $C = [\max\{\tau_1, -\tau_2\} - \Lambda_{\min}, \min\{\tau_1, -\tau_2\} + \Lambda_{\min}]$ *and* $\Lambda_{\min} = \inf_{x \in \mathcal{X}} \Lambda(x)$. *Furthermore, for any* $t \in [0, 1]$*, there exist a distribution* $\mathcal{D}$ *and a hypothesis* $h \in \mathcal{H}$ *such that* $\mathcal{R}_{\ell_{0-1}}(h) - \mathcal{R}^*_{\ell_{0-1}}(\mathcal{H}) + \mathcal{M}_{\ell_{0-1}}(\mathcal{H}) = t$ *and* $\mathcal{R}_{\ell_{\mathrm{cstnd}}}(h) - \mathcal{R}^*_{\ell_{\mathrm{cstnd}}}(\mathcal{H}) + \mathcal{M}_{\ell_{\mathrm{cstnd}}}(\mathcal{H}) = \mathcal{T}^{\mathrm{cstnd}}(t)$.

*Proof.* For the constrained loss $\ell^{\mathrm{cstnd}}$, the conditional $\ell^{\mathrm{cstnd}}$-risk can be expressed as follows:

$$\begin{aligned} \mathcal{C}_{\ell^{\mathrm{cstnd}}}(h, x) &= \sum_{y \in \mathcal{Y}} p(x, y)\ell^{\mathrm{cstnd}}(h, x, y) \\ &= \sum_{y \in \mathcal{Y}} p(x, y) \sum_{y' \neq y} \Phi(-h(x, y')) \\ &= \sum_{y \in \mathcal{Y}} \Phi(-h(x, y)) \sum_{y' \neq y} p(x, y') \\ &= \sum_{y \in \mathcal{Y}} \Phi(-h(x, y))(1 - p(x, y)) \\ &= \Phi(-h(x, y_{\max}))(1 - p(x, y_{\max})) + \Phi(-h(x, \mathsf{h}(x)))(1 - p(x, \mathsf{h}(x))) \\ &\quad + \sum_{y \notin \{y_{\max}, \mathsf{h}(x)\}} \Phi(-h(x, y))(1 - p(x, y)). \end{aligned}$$

For any $h \in \overline{\mathcal{H}}$ and $x \in \mathcal{X}$, by the definition of $\overline{\mathcal{H}}$, we can always find a family of hypotheses $\{h_\mu\} \subset \mathcal{H}$ such that $h_\mu(x, \cdot)$ take the following values:

$$h_\mu(x, y) = \begin{cases} h(x, y) & \text{if } y \notin \{y_{\max}, \mathsf{h}(x)\} \\ h(x, y_{\max}) + \mu & \text{if } y = \mathsf{h}(x) \\ h(x, \mathsf{h}(x)) - \mu & \text{if } y = y_{\max}. \end{cases}$$

Note that the hypotheses $h_\mu$ satisfies the constraint:

$$\sum_{y \in \mathcal{Y}} h_\mu(x, y) = \sum_{y \in \mathcal{Y}} h(x, y) = 0, \ \forall \mu \in \mathbb{R}.$$

Since $h_\mu(x, y) \in [-\Lambda(x), \Lambda(x)]$, we have the following constraints on $\mu$:

$$- \Lambda(x) - h(x, y_{\max}) \le \mu \le \Lambda(x) - h(x, y_{\max})$$
$$- \Lambda(x) + h(x, \mathsf{h}(x)) \le \mu \le \Lambda(x) + h(x, \mathsf{h}(x)).$$

Let $p_1 = p(x, y_{\max})$, $p_2 = p(x, \mathsf{h}(x))$, $\tau_1 = h(x, \mathsf{h}(x))$ and $\tau_2 = h(x, y_{\max})$ to simplify the notation. Then, the constraint on $\mu$ can be expressed as

$$\mu \in \overline{C}, \quad \overline{C} = [\max\{\tau_1, -\tau_2\} - \Lambda(x), \min\{\tau_1, -\tau_2\} + \Lambda(x)]$$

Since $\max\{\tau_1, -\tau_2\} - \min\{\tau_1, -\tau_2\} = |\tau_1 + \tau_2| \le |\tau_1| + |\tau_2| \le 2\Lambda(x)$, $C$ is not an empty set. By the definition of $h_\mu$, we have for any $h \in \mathcal{H}$ and $x \in \mathcal{X}$,

$$\mathcal{C}_{\ell\text{cstnd}}(h, x) - \inf_{\mu \in \overline{C}} \mathcal{C}_{\ell\text{cstnd}}(h_\mu, x)$$

$$= \sup_{\mu \in \overline{C}} \left\{ (1 - p_1)[\Phi(-\tau_2) - \Phi(-\tau_1 + \mu)] + (1 - p_2)[\Phi(-\tau_1) - \Phi(-\tau_2 - \mu)] \right\}$$

$$= \sup_{\mu \in \overline{C}} \left\{ \frac{2 - P - p_1 + p_2}{2} [\Phi(-\tau_2) - \Phi(-\tau_1 + \mu)] + \frac{2 - P + p_1 - p_2}{2} [\Phi(-\tau_1) - \Phi(-\tau_2 - \mu)] \right\}$$

$$\left( P = p_1 + p_2 \in \left[ \tfrac{1}{n-1}, 1 \right] \right)$$

$$= \inf_{P \in [\frac{1}{n-1}, 1]} \inf_{\tau_1 \ge \max\{\tau_2, 0\}} \sup_{\mu \in \overline{C}} \left\{ \frac{2 - P - p_1 + p_2}{2} [\Phi(-\tau_2) - \Phi(-\tau_1 + \mu)] \right.$$

$$\left. + \frac{2 - P + p_1 - p_2}{2} [\Phi(-\tau_1) - \Phi(-\tau_2 - \mu)] \right\} \qquad (\tau_1 \ge 0, \tau_2 \le \tau_1)$$

$$\ge \inf_{P \in [\frac{1}{n-1}, 1]} \inf_{\tau_1 \ge \max\{\tau_2, 0\}} \sup_{\mu \in C} \left\{ \frac{2 - P - p_1 + p_2}{2} [\Phi(-\tau_2) - \Phi(-\tau_1 + \mu)] \right.$$

$$\left. + \frac{2 - P + p_1 - p_2}{2} [\Phi(-\tau_1) - \Phi(-\tau_2 - \mu)] \right\}$$

$$(C = [\max\{\tau_1, -\tau_2\} - \Lambda_{\min}, \min\{\tau_1, -\tau_2\} + \Lambda_{\min}] \subset \overline{C} \text{ since } \Lambda_{\min} \le \Lambda(x))$$

$$= \inf_{P \in [\frac{1}{n-1}, 1]} \inf_{\tau_1 \ge \max\{\tau_2, 0\}} \left\{ \frac{2 - P - p_1 + p_2}{2} \Phi(-\tau_2) + \frac{2 - P + p_1 - p_2}{2} \Phi(-\tau_1) \right.$$

$$\left. - \inf_{\mu \in C} \left\{ \frac{2 - P - p_1 + p_2}{2} \Phi(-\tau_1 + \mu) + \frac{2 - P + p_1 - p_2}{2} \Phi(-\tau_2 - \mu) \right\} \right\}$$

$$= \mathcal{T}^{\text{cstnd}}(p_1 - p_2)$$

$$= \mathcal{T}^{\text{cstnd}}(\Delta \mathcal{C}_{\ell_{0-1}, \mathcal{H}}(h, x)). \qquad \text{(by Lemma 1)}$$

Note that for $n = 2$, an additional constraint $\tau_1 + \tau_2 = 1$ is imposed and the expression can be simplified as

$$\mathcal{C}_{\ell\text{cstnd}}(h, x) - \inf_{\mu \in \overline{C}} \mathcal{C}_{\ell\text{cstnd}}(h_\mu, x)$$

$$\ge \inf_{\tau \ge 0} \sup_{\mu \in [\tau - \Lambda_{\min}, \tau + \Lambda_{\min}]} \left\{ \frac{1 - p_1 + p_2}{2} [\Phi(\tau) - \Phi(-\tau + \mu)] + \frac{1 + p_1 - p_2}{2} [\Phi(-\tau) - \Phi(\tau - \mu)] \right\}$$

$$= \mathcal{T}^{\text{cstnd}}(p_1 - p_2)$$

$$= \mathcal{T}^{\text{cstnd}}(\Delta \mathcal{C}_{\ell_{0-1}, \mathcal{H}}(h, x)). \qquad \text{(by Lemma 1)}$$

Since $\mathcal{T}^{\text{cstnd}}$ is convex, by Jensen's inequality, we obtain for any hypothesis $h \in \mathcal{H}$ and any distribution,

$$\mathcal{T}^{\text{cstnd}}\big(\mathcal{R}_{\ell_{0-1}}(h) - \mathcal{R}^*_{\ell_{0-1}}(\mathcal{H}) + \mathcal{M}_{\ell_{0-1}}(\mathcal{H})\big)$$

$$= \mathcal{T}^{\text{cstnd}}\Big(\underset{X}{\mathbb{E}}[\Delta\mathcal{C}_{\ell_{0-1},\mathcal{H}}(h,x)]\Big)$$

$$\leq \underset{X}{\mathbb{E}}\big[\mathcal{T}^{\text{cstnd}}(\Delta\mathcal{C}_{\ell_{0-1},\mathcal{H}}(h,x))\big]$$

$$\leq \underset{X}{\mathbb{E}}\big[\Delta\mathcal{C}_{\ell^{\text{cstnd}},\mathcal{H}}(h,x)\big]$$

$$= \mathcal{R}_{\ell^{\text{cstnd}}}(h) - \mathcal{R}^*_{\ell^{\text{cstnd}}}(\mathcal{H}) + \mathcal{M}_{\ell^{\text{cstnd}}}(\mathcal{H}).$$

Let $n = 2$. For any $t \in [0,1]$, we consider the distribution that concentrates on a singleton $\{x\}$ and satisfies $p(x,1) = \frac{1+t}{2}$, $p(x,2) = \frac{1-t}{2}$. For any $\epsilon > 0$, by the definition of infimum, we can take $h \in \mathcal{H}$ such that $h(x,2) = \tau_\epsilon \geq 0$ and satisfies

$$\underset{\mu \in [\tau_\epsilon - \Lambda_{\min}, \tau_\epsilon + \Lambda_{\min}]}{\sup} \left\{ \frac{1-t}{2}[\Phi(\tau_\epsilon) - \Phi(-\tau_\epsilon + \mu)] + \frac{1+t}{2}[\Phi(-\tau_\epsilon) - \Phi(\tau_\epsilon - \mu)] \right\} < \mathcal{T}^{\text{cstnd}}(t) + \epsilon.$$

Then,

$$\mathcal{R}_{\ell_{0-1}}(h) - \mathcal{R}^*_{\ell_{0-1}}(\mathcal{H}) + \mathcal{M}_{\ell_{0-1}}(\mathcal{H}) = \mathcal{R}_{\ell_{0-1}}(h) - \mathbb{E}_X\big[\mathcal{C}^*_{\ell_{0-1}}(\mathcal{H},x)\big]$$

$$= \mathcal{C}_{\ell_{0-1}}(h,x) - \mathcal{C}^*_{\ell_{0-1}}(\mathcal{H},x)$$

$$= t$$

and

$$\mathcal{T}^{\text{cstnd}}(t) \leq \mathcal{R}_{\ell^{\text{cstnd}}}(h) - \mathcal{R}^*_{\ell^{\text{cstnd}}}(\mathcal{H}) + \mathcal{M}_{\ell^{\text{cstnd}}}(\mathcal{H})$$

$$= \mathcal{R}_{\ell^{\text{cstnd}}}(h) - \mathbb{E}_X\big[\mathcal{C}^*_{\ell^{\text{cstnd}}}(\mathcal{H},x)\big]$$

$$= \mathcal{C}_{\ell^{\text{cstnd}}}(h,x) - \mathcal{C}^*_{\ell^{\text{cstnd}}}(\mathcal{H},x)$$

$$= \underset{\mu \in [\tau_\epsilon - \Lambda_{\min}, \tau_\epsilon + \Lambda_{\min}]}{\sup} \left\{ \frac{1-t}{2}[\Phi(\tau_\epsilon) - \Phi(-\tau_\epsilon + \mu)] + \frac{1+t}{2}[\Phi(-\tau_\epsilon) - \Phi(\tau_\epsilon - \mu)] \right\}$$

$$< \mathcal{T}^{\text{cstnd}}(t) + \epsilon.$$

By letting $\epsilon \to 0$, we conclude the proof. The proof for $n > 2$ directly extends from the case when $n = 2$. Indeed, for any $t \in [0,1]$, we consider the distribution that concentrates on a singleton $\{x\}$ and satisfies $p(x,1) = \frac{1+t}{2}$, $p(x,2) = \frac{1-t}{2}$, $p(x,y) = 0, 3 \leq y \leq n$. For any $\epsilon > 0$, by the definition of infimum, we can take $h \in \mathcal{H}$ such that $h(x,1) = \tau_{1,\epsilon}$, $h(x,2) = \tau_{2,\epsilon}$, $h(x,3) = 0$, $3 \leq y \leq n$ and satisfies $\tau_{1,\epsilon} + \tau_{2,\epsilon} = 0$, and

$$\underset{P \in \left[\frac{1}{n-1}, 1\right]}{\inf} \underset{\mu \in C}{\sup} \left\{ \frac{2-P-t}{2}[\Phi(-\tau_{2,\epsilon}) - \Phi(-\tau_{1,\epsilon} + \mu)] + \frac{2-P+t}{2}[\Phi(-\tau_{1,\epsilon}) - \Phi(-\tau_{2,\epsilon} - \mu)] \right\}$$

$$= \underset{\mu \in C}{\sup} \left\{ \frac{1-t}{2}[\Phi(-\tau_{2,\epsilon}) - \Phi(-\tau_{1,\epsilon} + \mu)] + \frac{1+t}{2}[\Phi(-\tau_{1,\epsilon}) - \Phi(-\tau_{2,\epsilon} - \mu)] \right\}$$

$$< \mathcal{T}^{\text{cstnd}}(t) + \epsilon.$$

Then,

$$\mathcal{R}_{\ell_{0-1}}(h) - \mathcal{R}^*_{\ell_{0-1}}(\mathcal{H}) + \mathcal{M}_{\ell_{0-1}}(\mathcal{H}) = t$$

and

$$\mathcal{T}^{\text{cstnd}}(t) \leq \mathcal{R}_{\ell^{\text{cstnd}}}(h) - \mathcal{R}^*_{\ell^{\text{cstnd}}}(\mathcal{H}) + \mathcal{M}_{\ell^{\text{cstnd}}}(\mathcal{H}) < \mathcal{T}^{\text{cstnd}}(t) + \epsilon.$$

By letting $\epsilon \to 0$, we conclude the proof. $\qquad\square$

### F.2 Constrained exponential loss

**Theorem 13 ($\overline{\mathcal{H}}$-consistency bounds for constrained exponential loss).** *Let $\Phi(t) = e^{-t}$. For any $h \in \overline{\mathcal{H}}$ and any distribution,*

$$\mathcal{R}_{\ell_{0-1}}(h) - \mathcal{R}^*_{\ell_{0-1}}(\overline{\mathcal{H}}) + \mathcal{M}_{\ell_{0-1}}(\overline{\mathcal{H}}) \leq \Psi^{-1}\big(\mathcal{R}_{\ell^{\text{cstnd}}}(h) - \mathcal{R}^*_{\ell^{\text{cstnd}}}(\overline{\mathcal{H}}) + \mathcal{M}_{\ell^{\text{cstnd}}}(\overline{\mathcal{H}})\big)$$

*where* $\Psi(t) = \begin{cases} 1 - \sqrt{1 - t^2} & t \leq \frac{e^{2\Lambda_{\min}} - 1}{e^{2\Lambda_{\min}} + 1} \\ \frac{t}{2}\left(e^{\Lambda_{\min}} - e^{-\Lambda_{\min}}\right) + \frac{2 - e^{\Lambda_{\min}} - e^{-\Lambda_{\min}}}{2} & \text{otherwise.} \end{cases}$

*Proof.* For $n = 2$, plugging in $\Phi(t) = e^{-t}$ in Theorem 12, gives

$$\overline{\mathcal{J}}^{\text{cstnd}}(t) = \inf_{\tau \geq 0} \sup_{\mu \in [\tau - \Lambda_{\min}, \tau + \Lambda_{\min}]} \left\{ \frac{1-t}{2} [e^{-\tau} - e^{\tau - \mu}] + \frac{1+t}{2} [e^{\tau} - e^{-\tau + \mu}] \right\}.$$

By differentiating with respect to $\tau$, we can see that the infimum is achieved when $\tau = 0$ modulo some elementary analysis. Thus, $\overline{\mathcal{J}}^{\text{cstnd}}$ can be reformulated as

$$\overline{\mathcal{J}}^{\text{cstnd}} = \sup_{\mu \in [-\Lambda_{\min}, \Lambda_{\min}]} \left\{ \frac{1-t}{2} [1 - e^{-\mu}] + \frac{1+t}{2} [1 - e^{\mu}] \right\}$$

$$= 1 + \sup_{\mu \in [-\Lambda_{\min}, \Lambda_{\min}]} g(\mu).$$

where $g(\mu) = -\frac{1-t}{2} e^{-\mu} - \frac{1+t}{2} e^{\mu}$. Since $g$ is continuous, it attains its supremum over a compact set. Note that $g$ is concave and differentiable. In view of that, the maximum over the open set $(-\infty, +\infty)$ can be obtained by setting its gradient to zero. Differentiate $g(\mu)$ to optimize, we obtain

$$g(\mu^*) = 0, \quad \mu^* = \frac{1}{2} \log \frac{1-t}{1+t}$$

Moreover, by the concavity, $g(\mu)$ is non-increasing when $\mu \geq \mu^*$. Since $\mu^* \leq 0$ and $\Lambda_{\min} \geq 0$, we have

$$\mu^* \leq 0 \leq \Lambda_{\min}$$

In view of the constraint, if $\mu^* \geq -\Lambda_{\min}$, the maximum is achieved by $\mu = \mu^*$. Otherwise, if $\mu^* < -\Lambda_{\min}$, since $g(\mu)$ is non-increasing when $\mu \geq \mu^*$, the maximum is achieved by $\mu = -\Lambda_{\min}$. Since $\mu^* \geq -\Lambda_{\min}$ is equivalent to $t \leq \frac{e^{2\Lambda_{\min}} - 1}{e^{2\Lambda_{\min}} + 1}$, the maximum can be expressed as

$$\max_{\mu \in [-\Lambda_{\min}, \Lambda_{\min}]} g(\mu) = \begin{cases} g(\mu^*) & t \leq \frac{e^{2\Lambda_{\min}} - 1}{e^{2\Lambda_{\min}} + 1} \\ g(-\Lambda_{\min}) & \text{otherwise} \end{cases}$$

Computing the value of $g$ at these points yields:

$$g(\mu^*) = -\sqrt{1 - t^2}$$

$$g(-\Lambda_{\min}) = -\frac{1-t}{2} e^{\Lambda_{\min}} - \frac{1+t}{2} e^{-\Lambda_{\min}}.$$

Then, if $t \leq \frac{e^{2\Lambda_{\min}} - 1}{e^{2\Lambda_{\min}} + 1}$, we obtain

$$\overline{\mathcal{J}}^{\text{cstnd}} = 1 - \sqrt{1 - t^2}.$$

Otherwise, we obtain

$$\overline{\mathcal{J}}^{\text{cstnd}} = 1 - \frac{1-t}{2} e^{\Lambda_{\min}} - \frac{1+t}{2} e^{-\Lambda_{\min}}$$

$$= \frac{t}{2} \left( e^{\Lambda_{\min}} - e^{-\Lambda_{\min}} \right) + \frac{2 - e^{\Lambda_{\min}} - e^{-\Lambda_{\min}}}{2}.$$

For $n > 2$, plugging in $\Phi(t) = e^{-t}$ in Theorem 12, gives

$$\overline{\mathcal{J}}^{\text{cstnd}}(t) = \inf_{P \in [\frac{1}{n-1}, 1]} \inf_{\tau_1 \geq \max\{\tau_2, 0\}} \sup_{\mu \in C} \left\{ \frac{2 - P - t}{2} [e^{\tau_2} - e^{\tau_1 - \mu}] + \frac{2 - P + t}{2} [e^{\tau_1} - e^{\tau_2 + \mu}] \right\}.$$

where $C = [\max\{\tau_1, -\tau_2\} - \Lambda_{\min}, \min\{\tau_1, -\tau_2\} + \Lambda_{\min}]$. By differentiating with respect to $\tau_2$ and $P$, we can see that the infimum is achieved when $\tau_2 = \tau_1 = 0$ and $P = 1$ modulo some elementary analysis. Thus, $\overline{\mathcal{J}}^{\text{cstnd}}$ can be reformulated as

$$\overline{\mathcal{J}}^{\text{cstnd}} = \sup_{\mu \in C} \left\{ \frac{1-t}{2} [1 - e^{-\mu}] + \frac{1+t}{2} [1 - e^{\mu}] \right\}$$

$$= 1 + \sup_{\mu \in C} g(\mu).$$

where $C = [-\Lambda_{\min}, \Lambda_{\min}]$ and $g(\mu) = -\frac{1-t}{2}e^{-\mu} - \frac{1+t}{2}e^{\mu}$. Since $g$ is continuous, it attains its supremum over a compact set. Note that $g$ is concave and differentiable. In view of that, the maximum over the open set $(-\infty, +\infty)$ can be obtained by setting its gradient to zero. Differentiate $g(\mu)$ to optimize, we obtain

$$g(\mu^*) = 0, \quad \mu^* = \frac{1}{2}\log\frac{1-t}{1+t}$$

Moreover, by the concavity, $g(\mu)$ is non-increasing when $\mu \geq \mu^*$. Since $\mu^* \leq 0$ and $\Lambda_{\min} \geq 0$, we have

$$\mu^* \leq 0 \leq \Lambda_{\min}$$

In view of the constraint, if $\mu^* \geq -\Lambda_{\min}$, the maximum is achieved by $\mu = \mu^*$. Otherwise, if $\mu^* < -\Lambda_{\min}$, since $g(\mu)$ is non-increasing when $\mu \geq \mu^*$, the maximum is achieved by $\mu = -\Lambda_{\min}$. Since $\mu^* \geq -\Lambda_{\min}$ is equivalent to $t \leq \frac{e^{2\Lambda_{\min}}-1}{e^{2\Lambda_{\min}}+1}$, the maximum can be expressed as

$$\max_{\mu\in[-\Lambda_{\min},\Lambda_{\min}]} g(\mu) = \begin{cases} g(\mu^*) & t \leq \frac{e^{2\Lambda_{\min}}-1}{e^{2\Lambda_{\min}}+1} \\ g(-\Lambda_{\min}) & \text{otherwise} \end{cases}$$

Computing the value of $g$ at these points yields:

$$g(\mu^*) = -\sqrt{1-t^2}$$
$$g(-\Lambda_{\min}) = -\frac{1-t}{2}e^{\Lambda_{\min}} - \frac{1+t}{2}e^{-\Lambda_{\min}}.$$

Then, if $t \leq \frac{e^{2\Lambda_{\min}}-1}{e^{2\Lambda_{\min}}+1}$, we obtain

$$\overline{\mathcal{J}}^{\text{cstnd}} = 1 - \sqrt{1-t^2}.$$

Otherwise, we obtain

$$\overline{\mathcal{J}}^{\text{cstnd}} = 1 - \frac{1-t}{2}e^{\Lambda_{\min}} - \frac{1+t}{2}e^{-\Lambda_{\min}}$$
$$= \frac{t}{2}\left(e^{\Lambda_{\min}} - e^{-\Lambda_{\min}}\right) + \frac{2 - e^{\Lambda_{\min}} - e^{-\Lambda_{\min}}}{2}.$$

Since $\overline{\mathcal{J}}^{\text{cstnd}}$ is convex, by Theorem 12, for any $h \in \overline{\mathcal{H}}$ and any distribution,

$$\mathcal{R}_{\ell_{0-1}}(h) - \mathcal{R}^*_{\ell_{0-1}}(\overline{\mathcal{H}}) + \mathcal{M}_{\ell_{0-1}}(\overline{\mathcal{H}}) \leq \Psi^{-1}\left(\mathcal{R}_{\ell^{\text{cstnd}}}(h) - \mathcal{R}^*_{\ell^{\text{cstnd}}}(\overline{\mathcal{H}}) + \mathcal{M}_{\ell^{\text{cstnd}}}(\overline{\mathcal{H}})\right)$$

where

$$\Psi(t) = \begin{cases} 1 - \sqrt{1-t^2} & t \leq \frac{e^{2\Lambda_{\min}}-1}{e^{2\Lambda_{\min}}+1} \\ \frac{t}{2}\left(e^{\Lambda_{\min}} - e^{-\Lambda_{\min}}\right) + \frac{2-e^{\Lambda_{\min}}-e^{-\Lambda_{\min}}}{2} & \text{otherwise}. \end{cases}$$

$\square$

