# OpenReview forum: "$H$-Consistency Bounds: Characterization and Extensions"
_NeurIPS.cc/2023/Conference — NeurIPS 2023 poster_

### Official Review · Reviewer_AHgy · 2023-07-01

**Soundness:** 3 good
**Presentation:** 3 good
**Contribution:** 3 good
**Rating:** 7
**Confidence:** 3

**Summary:**

This work proposes a general characterization and an extension of H-consistency bounds for multiclass classification. By introducing an error transformation function, the paper provides a general tool for deriving H-consistency bounds with tightness guarantees. The paper demonstrates that the proposed tools and bounds can recover or even improve results from various recent works.

**Strengths:**

This is good work in the field, original with innovative methodologies and new insights. The paper provides a general characterization and extension of H-consistency bounds for multiclass classification. Focusing on two widely studied types of loss functions in the literature, comp-sum losses and constrained losses, a new tool is introduced to derive H-consistency bounds with tightness guarantees. Before this work, deriving such bounds often required proofs for each instance. The tool proposed in this work, based on error transformations (Theorem 2, 3, 5, 10, 11, 12), is general and can be applied to derive H-consistency bounds (Theorem 4, 6, 9, Corollary 7, 8) for various loss functions and hypothesis sets. The paper demonstrates that the proposed tools and bounds can recover or even improve results from various recent works (Line 201-206, 246-248, 253-255, 261-272, 301-306, 318-322).

The quality of research is good, with meticulous comparisons with several recent works and detailed analysis leading to convincing conclusions. Well-organized and well-written, the paper adeptly communicates complex concepts in a relatively understandable way. The work is significant, with findings that resonate beyond the specific area of study and are likely to stimulate further research in the field of H-consistency bounds for multiclass classification.

Overall, this paper is a nice addition to the literature.

**Weaknesses:**

As far as I am concerned, I do not see any significant weaknesses.

**Questions:**

Could you provide further explanation on the connection between the results presented in this study and the theory of calibration functions as described by Steinwart in 2007?

**Limitations:**

I could not find the location where the authors explicitly described the limitations of the work.

---

> ### Author Rebuttal · Authors · 2023-08-08
>
> Thank you for your appreciation of our work. Below please find responses to specific questions.
>
> **Could you provide further explanation on the connection between the results presented in this study and the theory of calibration functions as described by Steinwart in 2007?**
>
> **Response:** The calibration function is a general tool tailored to the family of all measurable functions, and is designed to provide excess error bounds between a surrogate loss and the target loss. The calibration function does not take into account the specific hypothesis set. On the other hand, our error transformation function is a general tool designed to provide $H$-consistency bounds between a surrogate loss and the target loss. These functions are tailored to the hypothesis set $H$ adopted. In the special case where $H$ is the family of all measurable functions, the error transformation function coincides with the calibration function, and the derived $H$-consistency bound coincides with an excess error bound. We will further clarify and detail these explanations in the final version.
>
> **Limitations:**
> **I could not find the location where the authors explicitly described the limitations of the work.**
>
> **Response:** Thank you for pointing it out. We will add an explicit discussion in the final version.

---

> > ### Comment · Reviewer_AHgy · 2023-08-16
> >
> > Thanks for the clarifications. I have read all the reviews and responses. I appreciate the authors' effort. I retain my rating of 7.

---

### Official Review · Reviewer_PRWE · 2023-07-04

**Soundness:** 3 good
**Presentation:** 2 fair
**Contribution:** 3 good
**Rating:** 6
**Confidence:** 3

**Summary:**

Unlike previous work on H-consistency bounds for surrogate loss functions, this paper attempts to analyze them from a more unified perspective.
For this purpose, the paper proposed a new error transformation method to present new tight H-consistency bounds for comp-sum losses and constrained losses in multi-class classification. With this method, the authors derived the
H-consistency bounds in a unified manner for various loss functions, and a new theoretical analysis have developed for non-complete hypothesis sets.

**Strengths:**

# Writing and Organization
- The paper is well organized such that it is easy to understand the contribution of this paper, especially the technical difference from existing work.
-  The extensive comparison in Appendix A is very helpful in understanding the recent progress in this field, and I thought this Appendix A should be placed in the main paper if there would be any spaces.

# Technical contribution
- This paper presents a significant improvement over the existing H-consistency bounds by developing a unified and consistent approach based on error transformation, which seems novel.

**Weaknesses:**

- I think the argument of section 2 should be improved to understand more easily.
 1. The notation introduced in Sec 2 is hard to understand at first read. It would be helpful to add the summary of notations in the Appendix so that readers can easily follow the importance of this setting
 2. I think the current argument in paragraph 106 is hard to follow. I finally understood the paragraph after reading Appendix B.2. It might be better to introduce  $I_l(\mathcal{H})$ in the main paper, which only appears in the Appendix, better to understand the difference between $M_l$ and $A_l$.
 3.  I like the explanation in Appendix B.2 and B.3 more than the current main paper since the importance of each concept is discussed mathematically and clearly.

- The statement, especially Theorem 2, is very general and hard to understand. I understand that this is unavoidable because the developed analysis is a kind of unified viewpoint, but it would be better to add a little more discussion of the statements, for example,  under Theorem 2.

Typos
- In line 107, H_all is used before its formal definition. Currently, it is defined in Line 114, so it should be corrected.
- Some parentheses are missing in equations from lines 532 to 533 in the Appendix.

**Questions:**

- In line 59, the author explained that the bound in Awasthi 2022b may not be tight due to its ad-hoc analysis. I would like to know what it means by ad-hoc analysis and when it is not tight to clarify the limitation of existing work.

- In Appendix C.1, I could not follow the equation between lines 574 and 575. Could you explain how the condition is derived from the second equality to the third equality, where $\inf_{\tau_1 \leq \max(\dots...)}$ appeared.

- I found it hard to understand what is the intuition behind J^{comp} for $n>2$ in Theorem 2 since the constraint is very complex. Is there any clear explanation for that?

- In theorem 2 and 5, the statement says that there exists a distribution D and h such that the equality holds. Could you present an explicit example of that ?

**Limitations:**

The limitation and assumption of the analysis is presented in detail.

---

> ### Author Rebuttal · Authors · 2023-08-08
>
> Thank you for your encouraging review. We have carefully addressed all the questions raised. Please find our responses below.
>
> **Weaknesses:**
>
> **1. I think the argument of section 2 should be improved to understand more easily.**
>
> **Response:** We thank the reviewer for their suggestions. We will certainly incorporate them and will further improve our notation and presentation. In particular, we will include a notation summary in the appendix and a comprehensive discussion of each concept in the main body, by leveraging the extra page available in the final version.
>
> **2. The statement, especially Theorem 2, is very general and hard to understand. I understand that this is unavoidable because the developed analysis is a kind of unified viewpoint, but it would be better to add a little more discussion of the statements, for example, under Theorem 2.**
>
> **Response:** Thank you for your suggestions. We will definitely provide a more comprehensive discussion of the general results, particularly focusing on Theorem 2, in the final version.
>
> **3. Typos.**
>
> Response: Thank you, we will correct them.
>
> **Questions:**
>
> **1. In line 59, the author explained that the bound in Awasthi 2022b may not be tight due to its ad-hoc analysis. I would like to know what it means by ad-hoc analysis and when it is not tight to clarify the limitation of existing work.**
>
> **Response:** When referring to their ''ad-hoc analysis'', we mean that their method of upper-bounding the estimation error of the target loss by that of the surrogate loss heavily depends on the specific loss function and does not extend to a new loss function. Their upper bounds may not be tight, as the ad-hoc inequality used in their derivation could, in some instances, be further improved. For instance, their bounds for constrained exponential losses are found not to be tight. In contrast, we offer a general tool to derive $H$-consistency bounds in a more systematic manner. Moreover, by using the error transformation functions, we succeed in obtaining bounds that are tighter than those provided by Awasthi et al. [2022b].
> We will further clarify this in the final version.
>
> **2. In Appendix C.1, I could not follow the equation between lines 574 and 575. Could you explain how the condition is derived from the second equality to the third equality, where $\inf_{\gamma_1\leq \max(\ldots)}$ appeared.**
>
> **Response:** Sorry for the confusion. There is a typo; the third equality should actually be an inequality, derived by directly taking the infimum. We will correct that in the final version.
>
> **3. I found it hard to understand what is the intuition behind J^{comp} for $n>2$ in Theorem 2 since the constraint is very complex. Is there any clear explanation for that?**
>
> **Response:** We acknowledge the complexity of the formulation for $n>2$, which is a result of its general and unifying nature, encompassing various loss functions. The conditional probability vector and scoring functions take on more flexible forms when $n>2$, leading to intricate constraints. We will provide a thorough explanation and clarification in the final version to make it more understandable.
>
> **4. In theorem 2 and 5, the statement says that there exists a distribution D and h such that the equality holds. Could you present an explicit example of that?**
>
> **Response:** The example of a particular distribution $\mathcal{D}$ and a hypothesis $h$ will be contingent on the specific loss function and hypothesis set. This is because the conditional probability vector of $\mathcal{D}$ and the scoring functions of $h$ must closely achieve the infimum in the transformation. In general, they are challenging to fully characterize, and often only existence can be shown. Nevertheless, we will seek to include in the final version an explicit example in simple cases for illustration.

---

### Official Review · Reviewer_x2ot · 2023-07-04

**Soundness:** 3 good
**Presentation:** 3 good
**Contribution:** 3 good
**Rating:** 7
**Confidence:** 2

**Summary:**

The authors propose a general characterization and extension of H-consistency bounds for multi-class classification. They introduce an error transformation function that serves as a general tool for deriving these guarantees with tightness guarantees. The paper demonstrates that calculating the error transformation function enables the derivation of H-consistency bounds for various loss functions and hypothesis sets.

The general tools and tight bounds presented in the paper offer several advantages. They improve existing bounds for complete hypothesis sets, encompass a wide range of previously studied and new loss functions, extend beyond the completeness assumption, provide guarantees for bounded hypothesis sets, and offer a stronger guarantee for logistic loss with linear hypothesis sets compared to previous work.

Overall, this paper contributes to the understanding and derivation of H-consistency bounds, providing a more general framework and tools for analyzing surrogate loss functions in machine learning.

**Strengths:**

This paper is well presented. It introduced a new general tool for deriving these H-consistency bounds with tightness guarantees. The results are carefully articulated and compared with existing work. I did not have enough time to verify all the math details, but the results look solid.

**Weaknesses:**

See next section

**Questions:**

Can the theory be extended to derive tight bounds for more complicated models such as simple neural networks?

**Limitations:**

No potential negative societal impact

---

> ### Author Rebuttal · Authors · 2023-08-08
>
> Thank you for your appreciation of our work. Below please find responses to specific questions.
>
> **Can the theory be extended to derive tight bounds for more complicated models such as simple neural networks?**
>
> **Response:** Good question! Our $H$-consistency bounds, as demonstrated in Theorem 2, Theorem 4, Theorem 5, Theorem 6, Theorem 9, Theorem 10, and Theorem 12, are not limited to specific hypothesis set forms. They are directly applicable to various types of hypothesis sets, including linear functions and complex neural networks. For instance, Corollary 8, derived from Theorem 6, provides $H$-consistency bounds for linear hypothesis sets by explicitly computing and incorporating the term $\Lambda(x)$ into the general formulation of $\Psi$.
>
> The same derivation can be extended to neural networks studied in [Awasthi et al., 2022a] and their multi-class generalization, where we calculate and substitute the corresponding $\Lambda(x)$ value. As a result, we obtain novel and tight $H$-consistency bounds for bounded neural network hypothesis sets in multi-class classification, highlighting the remarkable versatility of our general tools. We will further elaborate on that and include specific results related to the hypothesis set of neural networks in the final version.

---

> > ### Comment · Reviewer_x2ot · 2023-08-16
> >
> > Thank you for the clarification.

---

### Official Review · Reviewer_C2H4 · 2023-07-06

**Soundness:** 3 good
**Presentation:** 3 good
**Contribution:** 3 good
**Rating:** 6
**Confidence:** 4

**Summary:**

H-consistency bounds for surrogate loss function plays an important part in learning theory.
This manuscript provides a comprehensive characterization and extension of H-consistency
bounds for multi-class classification. The authors introduce a novel error transformation function
that enables the derivation of tighter bounds, even under weaker assumptions compared
to existing related works. Technically, this research presents a systematic approach to H-consistency bounds. Moreover,
it contributes to the refinement of results concerning the consistency of multi-class
classification on a theoretical level.

**Strengths:**

The authors present a systematic technique for H-consistency analysis and apply it to
various types of loss functions.

The authors compare the results in the manuscript with those in related works in detail
and show a significant advance.

**Weaknesses:**

This manuscript includes a wealth of work but only limited technical innovations compared
to [1].

[1] P. Awasthi, A. Mao, M. Mohri, and Y. Zhong. Multi-class H-consistency bounds.
NeurIPS 2022.

**Questions:**

• As stated in the manuscript, H-consistency is a generalization of Bayes consistency,
and new techniques have been developed in the analysis of H-consistency. Is it possible
to illustrate the advantages of H-consistency over Bayes consistency with more specific
examples?

• The H-consistency analysis for not only the linear hypothesis class but also the neural
network class has been done in [2]. Can the results in the manuscript be extended to
the neural network class as well? As far as I understand, the analysis in the manuscript
relies heavily on the explicit form of the hypothesis, does it mean that it is difficult to
extend these inequalities to the deep neural network class?

• Between Line 256 and 257: Φ−1 → Ψ−1

[2] P. Awasthi, A. Mao, M. Mohri, and Y. Zhong. H-consistency bounds for surrogate loss
minimizers. ICML 2022.

**Limitations:**

I suggest the authors illustrating more specifically the advantages of H-consistency over Bayes
consistency.

---

> ### Author Rebuttal · Authors · 2023-08-08
>
> Thank you for your encouraging review. We have carefully addressed all the questions raised. Please find our responses below.
>
> **Weaknesses:**
> **This manuscript includes a wealth of work but only limited technical innovations compared to [1].**
>
> **Response:** Our work introduces a key technical innovation: error transformation functions. These functions enable a more systematic and straightforward derivation of $H$-consistency bounds. The versatility of this novel tool offers significant advantages, including tighter bounds surpassing those in [1] for complete hypothesis sets and, importantly, tailored guarantees for bounded hypothesis sets, going beyond completeness assumptions.
>
> We are confident that our innovative approach will pave the way for advances in multi-class classification consistency research and benefit the analysis of consistency in various other scenarios.
>
> **Questions:**
>
> **1. As stated in the manuscript, H-consistency is a generalization of Bayes consistency, and new techniques have been developed in the analysis of H-consistency. Is it possible to illustrate the advantages of H-consistency over Bayes consistency with more specific examples?**
>
> **Limitations:**
> **I suggest the authors illustrating more specifically the advantages of H-consistency over Bayes consistency.**
>
> **Response:** Certainly, in the final version, we will provide illustrative examples. A critical issue with Bayes consistency lies in its assumption of access to the entire family of measurable functions, which contrasts with the limited hypothesis set $H$ a learning algorithm can rely on. The concept of $H$-consistency (and $H$-consistency bound) specifically aims to capture the properties of the particular hypothesis set $H$ used.
>
> For instance, Long and Servedio (2013) presented a compelling case where they demonstrated both theoretically and empirically that the expected error of an algorithm minimizing a Bayes-consistent loss remains bounded by a positive constant. In contrast, the expected error of an algorithm minimizing an inconsistent but realizable $H$-consistent loss approaches zero. This example highlights the significance of considering the characteristics of the chosen hypothesis set $H$ in practical learning scenarios.
>
>
> **2. The H-consistency analysis for not only the linear hypothesis class but also the neural network class has been done in [2]. Can the results in the manuscript be extended to the neural network class as well? As far as I understand, the analysis in the manuscript relies heavily on the explicit form of the hypothesis, does it mean that it is difficult to extend these inequalities to the deep neural network class?**
>
> **Response:** Good question! Our $H$-consistency bounds, as demonstrated in Theorem 2, Theorem 4, Theorem 5, Theorem 6, Theorem 9, Theorem 10, and Theorem 12, are not limited to specific hypothesis set forms. They are directly applicable to various types of hypothesis sets, including linear functions and complex neural networks. For instance, Corollary 8, derived from Theorem 6, provides $H$-consistency bounds for linear hypothesis sets by explicitly computing and incorporating the term $\Lambda(x)$ into the general formulation of $\Psi$.
>
> The same derivation can be extended to neural networks studied in [2] and their multi-class generalization, where we calculate and substitute the corresponding $\Lambda(x)$ value. As a result, we obtain novel and tight $H$-consistency bounds for bounded neural network hypothesis sets in multi-class classification, highlighting the remarkable versatility of our general tools. We will further elaborate on that and include specific results related to the hypothesis set of neural networks in the final version.
>
> **3. Between Line 256 and 257: $\Phi^{-1} \to \Psi^{-1}$**
>
> **Response:** Thank you, we will correct the typo.

---

### Official Review · Reviewer_VFQY · 2023-07-06

**Soundness:** 3 good
**Presentation:** 1 poor
**Contribution:** 2 fair
**Rating:** 4
**Confidence:** 3

**Summary:**

This paper presents an extension of previous works on $\mathcal{H}$-consistency bounds for multi-classification using a novel method. The strength of the paper lies in proposing a new method and tool that guarantees tight $\mathcal{H}$-consistency bounds for multi-classification.

**Strengths:**

The paper proposes a general characterisation $\mathcal{H}$-consistency bounds for multi-classification via new tools.

**Weaknesses:**

One weakness of the paper is the lack of clarity in the discussion of tightness guarantees. While the statement in lines 191-192 suggests the existence of a distribution $\mathcal{D}$ and a hypothesis for which the upper bound in Theorem 2 is tight, it is not explicitly clarified if this bound holds tight for all data distributions. Therefore, it is recommended to provide further clarification regarding the tightness statement.

To enhance the comprehension of the paper, it would be beneficial to include an experimental design demonstrating the tightness of the proposed bounds in comparison to previous results.

The authors mention the application of a tool to derive new results, but it would be helpful to explicitly mention and explain the techniques employed in the main body of the paper. Currently, it is unclear what these tools entail.

In the proof of Theorem 3, it is advised to provide additional elaboration on the equalities following line 587. It should be noted that in the second equality, the infimum with respect to $\tau$ is taken over all terms.

In line 318-319, the authors mentioned "Next, we illustrate the application of our theory through an example of constrained exponential losses,". However, the paper is finished.

The proof of Theorems 2 and 5 for the tightness is limited to n=2. For n>2, there is no discussion.

**Questions:**

Please refer to the weaknesses section for the identified limitations.


**Limitations:**

Please refer to the weaknesses section for the identified questions.

---

> ### Author Rebuttal · Authors · 2023-08-08
>
> Thank you for your useful comments. We have carefully addressed all the questions raised. Please find our responses below.
>
> **1. One weakness of the paper is the lack of clarity in the discussion of tightness guarantees. While the statement in lines 191-192 suggests the existence of a distribution $\mathcal{D}$ and a hypothesis for which the upper bound in Theorem 2 is tight, it is not explicitly clarified if this bound holds tight for all data distributions. Therefore, it is recommended to provide further clarification regarding the tightness statement.**
>
> **Response:** We will further clarify that in the final version. In short, our $H$-consistency bounds are distribution-independent and we do not claim tightness across all distributions. Our analysis can be extended, however, to derive finer distribution-dependent bounds under assumptions such as Massart’s noise condition, as in (Awasthi et al., 2022).
>
> **2. To enhance the comprehension of the paper, it would be beneficial to include an experimental design demonstrating the tightness of the proposed bounds in comparison to previous results.**
>
> **Response:** This is certainly a natural suggestion. We will seek to add such experiments in the final version to empirically illustrate the tightness of our proposed bounds.
>
> **3. The authors mention the application of a tool to derive new results, but it would be helpful to explicitly mention and explain the techniques employed in the main body of the paper. Currently, it is unclear what these tools entail.**
>
> **Response:** We will further clarify that in the final version. Our error transformation function serves as a very general tool for deriving $H$-consistency bounds with tightness guarantees. These functions are defined within each class of loss functions including comp-sum losses and constrained losses, and their formulation depends on the structure of the individual loss function class, the range of the hypothesis set and the number of classes. To derive explicit bounds, all that is needed is to calculate these error transformation functions. Under some broad assumptions on the auxiliary function within a loss function, these error transformation functions can be further distilled into more simplified forms, making them straightforward to compute.
>
> **4. In the proof of Theorem 3, it is advised to provide additional elaboration on the equalities following line 587. It should be noted that in the second equality, the infimum with respect to $\tau$ is taken over all terms.**
>
> **Response:** Thank you for the suggestion. You are absolutely correct, the infimum with respect to $\tau$ is taken over all terms in the second equality. We will definitely add more explanation and clarify the derivation further in the final version.
>
> **5. In line 318-319, the authors mentioned "Next, we illustrate the application of our theory through an example of constrained exponential losses,". However, the paper is finished.**
>
> **Response:** Sorry for the confusion. The example of constrained exponential losses is deferred to Appendix F.2 due to space limitations. We will further improve our presentation for readers and use the extra page in the final version to add more discussions in the main body.
>
> **6. The proof of Theorems 2 and 5 for the tightness is limited to n=2. For n>2, there is no discussion.**
>
> **Response:** Sorry for the confusion. The proof for n>2 closely follows and directly extends from the case when n= 2, by considering the distribution that concentrates on a singleton. This was inadvertently omitted, but we will definitely include a thorough discussion in the final version.

---

> > ### Comment · Reviewer_VFQY · 2023-08-15
> > **Response**
> >
> > I thank the authors for their response.
> >
> > It seems that the paper is not well-organised (item 6 and 5) and some parts are omitted.
> >
> > The authors had the chance to provide a simple experiments and discuss their results in rebuttal one-page.
> >
> > Regarding the proof of Theorem 3, in line 587, the second equality is not clear. could you please clarify this equality.

---

> > > ### Author Response · Authors · 2023-08-15
> > > **Thank you**
> > >
> > > We appreciate the reviewer’s feedback and comments.
> > >
> > > **1. It seems that the paper is not well-organised (item 6 and 5) and some parts are omitted.**
> > >
> > > **Response:** We appreciate the positive feedback we have received from multiple reviewers regarding our paper's organization. It is unfortunate that your perspective differs but we value your input. Due to space limitations, we had to relocate a portion of the content to the appendix shortly before the deadline. Furthermore, a minor section was inadvertently commented out, as you previously noted. As we have highlighted before, rest assured that these issues will be addressed and easily rectified in the final version.
> > >
> > > **2. The authors had the chance to provide simple experiments and discuss their results in rebuttal one-page.**
> > >
> > > **Response:** As indicated in our previous response, we already *prove* the tightness of our bounds. As already promised, we will seek to present an experiment illustrating the tightness of our bounds in the final version. But, we should emphasize that presenting a specific empirical example showcasing this tightness is not straightforward.
> > >
> > > **3. Regarding the proof of Theorem 3, in line 587, the second equality is not clear. Could you please clarify this equality.**
> > >
> > > **Response:** The equality simply corresponds to the fact that the supremum of a negative term can be equivalently written as the infimum of that term. Since the terms depending on $\mu$ are both negative, they can be first grouped together and next the equivalence mentioned is applied.
> > >
> > > In detail, the second equality repositions the supremum from the preceding equality within the curly brackets by using the fact that $\sup_{\mu \in [-\tau, 1-\tau]} \bigg \\{ -\frac{1+t}{2} \Phi(1-\tau-\mu) - \frac{1-t}{2} \Phi (\tau + \mu) \bigg \\} = -\inf_{\mu \in [-\tau, 1-\tau]} \bigg \\{ \frac{1+t}{2} \Phi(1-\tau-\mu) + \frac{1-t}{2} \Phi (\tau + \mu) \bigg \\}$. As you rightly noted, the infimum with respect to $\tau$ in the second equality should encompass all terms. We will correct that typo in the final version.
> > >
> > > Further, the subsequent equality is grounded on the observation that for any $\mu$ within the interval $[-\tau, 1-\tau]$, the values $(1-\tau-\mu)$, $(\tau + \mu)$ are confined to $[0,1]$ and $(1-\tau-\mu) + (\tau + \mu) =1$. This leads us to express $\inf_{\mu \in [-\tau, 1-\tau]} \bigg \\{ \frac{1+t}{2} \Phi(1-\tau-\mu) + \frac{1-t}{2} \Phi (\tau + \mu) \bigg \\}$ in an equivalent form in the third equality as $\inf_{\mu \in [-\frac12, \frac12]} \bigg \\{ \frac{1-t}{2} \Phi(\frac12+\mu) + \frac{1+t}{2} \Phi (\frac12 - \mu) \bigg \\}$. Consequently, the infimum with respect to $\tau$ pertains exclusively to the first part of the third equality.

---

> > > > ### Comment · Reviewer_VFQY · 2023-08-17
> > > > **Reposne.**
> > > >
> > > > I thank the authors for their response.
> > > >
> > > > Due the experiments, organisation and presentation of the paper, I decrease my evaluation score to 4.

---

### Official Review · Reviewer_3BV1 · 2023-07-08

**Soundness:** 2 fair
**Presentation:** 2 fair
**Contribution:** 2 fair
**Rating:** 5
**Confidence:** 1

**Summary:**

The authors present some extension study on the H-consistency bounds for various loss functions. They also introduce an error transformation function that the authors claim could be a general tool in deriving H-consistency bound.

**Strengths:**

The authors seem very well aware of the related work and have solid comparison with them.

**Weaknesses:**

Maybe the authors can simplify the notation and presentation so as the paper can be consumed by more general audience.

**Questions:**

1. I understand the importance of introducing a more general mathematical tool in deriving H-consistence bound for a wide range of function (but I did not check the correctness of this part). I don't know what's the value of a tigher H-consistent bound. This bound does not seem really useful (or maybe I am wrong) in pratise to guide people's design decision. Whereas Awasthi et al introduced the H-consistent bound.

2. Essentailly following 1 above, while we pushed the theoretical guranttee forward a little bit, application wise, what are the add-on values that this paper brings compared to the related work?

**Limitations:**

A bit hard for general audience to understand.

---

> ### Author Rebuttal · Authors · 2023-08-08
>
> Thank you for your useful comments. We have carefully addressed all the questions raised. Please find our responses below.
>
> **Weaknesses:**
> **Maybe the authors can simplify the notation and presentation so as the paper can be consumed by more general audience.**
>
> **Limitations:**
> **A bit hard for general audience to understand.**
>
> **Response:** We will definitely follow your suggestions and will seek to simplify and improve the notation and presentation for readers. The extra page in the final version will also allow us to expand the discussion in the main body.
>
> **Questions:**
>
> **1. I understand the importance of introducing a more general mathematical tool in deriving H-consistence bound for a wide range of functions (but I did not check the correctness of this part). I don't know what's the value of a tighter H-consistent bound. This bound does not seem really useful (or maybe I am wrong) in practice to guide people's design decisions. Whereas Awasthi et al introduced the H-consistent bound.**
>
> **Response:** Indeed, one of our principal contributions lies in a more general and convenient mathematical tool for proving $H$-consistency bounds. However, the derivation of tighter $H$-consistent bounds is equally significant.
>
> As mentioned by Awasthi et al, given a hypothesis set $H$, $H$-consistency bounds can be used to compare different surrogate loss functions and select the most favorable one, which depends on 1) The functional form of the $H$-consistency bound; 2) The smoothness of the loss and, more generally, its optimization properties; 3) Approximation properties of the surrogate loss function: for instance, given a choice of $H$, the minimizability gap for a surrogate loss may be more or less favorable; 4) The dependency of the multiplicative constant on the number of classes. Consequently, a tighter $H$-consistency bound provides a more accurate comparison, as a loose bound might not adequately capture the full advantage of using one surrogate loss. In contrast, Bayes-consistency does not take into account the hypothesis set and is an asymptotic property, thereby failing to guide the comparison of different surrogate losses.
>
> Another application of our $H$-consistency bounds involves deriving generalization bounds for surrogate loss minimizers, expressed in terms of the same quantities previously discussed. Therefore, when dealing with finite samples, a tighter $H$-consistency bound could also result in a corresponding tighter generalization bound.
>
> We will further elaborate on these aspects in the relevant section of our paper, underscoring the importance and the value of tighter $H$-consistency bounds.
>
> **2. Essentially following 1 above, while we pushed the theoretical guarantee forward a little bit, application wise, what are the add-on values that this paper brings compared to the related work?**
>
> **Response:** As already mentioned, our significant contributions include a more general and convenient mathematical tool for proving $H$-consistency bounds, along with tighter bounds that enable a better comparison of surrogate loss functions. These improved bounds also lead to tighter finite sample generalization bounds compared to previous work. Moreover, our novel results extend beyond previous completeness assumptions, offering guarantees applicable to bounded hypothesis sets commonly used with regularization. This enhancement provides meaningful learning guarantees. Further details will be expanded in the final version.

---

### Decision · Program_Chairs · 2023-09-21

**Decision:**

Accept (poster)

**Comment:**

This meta review is based on the reviews, the authors rebuttal and the discussions with the reviewers, discussions with the SAC, and ultimately my own judgement on the paper. There was a consensus that the paper contributes sound and interesting contributions. I feel this work deserves to be featured at NeurIPS and will attract interest from the community. I would like to personally invite the authors to carefully revise their manuscript to take into account the remarks and suggestions made by reviewers. Congratulations!